# Improved Regret Analysis in Gaussian Process Bandits:
# Optimality for Noiseless Reward, RKHS norm, and Non-Stationary Variance

**Shogo Iwazaki** [1]  **Shion Takeno** [2]

## Abstract

We study the Gaussian process (GP) bandit problem, whose goal is to minimize regret under an unknown reward function lying in some reproducing kernel Hilbert space (RKHS). The maximum posterior variance analysis is vital in analyzing near-optimal GP bandit algorithms such as maximum variance reduction (MVR) and phased elimination (PE). Therefore, we first show the new upper bound of the maximum posterior variance, which improves the dependence of the noise variance parameters of the GP. By leveraging this result, we refine the MVR and PE to obtain (i) a nearly optimal regret upper bound in the noiseless setting and (ii) regret upper bounds that are optimal with respect to the RKHS norm of the reward function. Furthermore, as another application of our proposed bound, we analyze the GP bandit under the time-varying noise variance setting, which is the kernelized extension of the linear bandit with heteroscedastic noise. For this problem, we show that MVR and PE-based algorithms achieve noise variance-dependent regret upper bounds, which match our regret lower bound.

## 1. Introduction

The Gaussian process (GP) bandits (Srinivas et al., 2010) is a powerful framework for sequential decision-making tasks to minimize regret defined by a black-box reward function, which belongs to known reproducing kernel Hilbert space (RKHS). The applications include many fields such as robotics (Berkenkamp et al., 2021), experimental design (Lei et al., 2021), and hyperparameter tuning task (Snoek et al., 2012).

Many existing studies have been conducted to obtain the

[1]MI-6 Ltd. Tokyo, Japan [2]Department of Mechanical Engineering, Nagoya University, Aichi, Japan. Correspondence to: Shogo Iwazaki <shogo.iwazaki@gmail.com>.

*Proceedings of the 42$^{nd}$ International Conference on Machine Learning*, Vancouver, Canada. PMLR 267, 2025. Copyright 2025 by the author(s).

theoretical guarantee for the regret. Established work by Srinivas et al. (2010) has shown the upper bounds of the cumulative regret for the GP upper confidence bound (GP-UCB) algorithm. Furthermore, Valko et al. (2013) have shown a tighter regret upper bound for the SupKernelUCB algorithm. Scarlett et al. (2017) have shown the lower bound of the regret, which implies that the regret upper bound from (Valko et al., 2013) is near-optimal; that is, the regret upper bound matches the lower bound except for the poly-logarithmic factor. Then, several studies further tackled obtaining a near-optimal GP-bandit algorithm. Vakili et al. (2021a) have proposed maximum variance reduction (MVR), which is shown to be near-optimal for the simple regret incurred by the last recommended action. Furthermore, Li & Scarlett (2022) have shown that phased elimination (PE) is near-optimal for the cumulative regret. The regret analysis of MVR and PE heavily depends on the upper bound for the maximum posterior variance.

We derive the upper bound of the maximum posterior variance in Section 3, by which we tackle tightening the regret upper bound in the settings where room for improvement remains. Our contributions are summarized as follows:

1. In Section 3, we obtain the upper bound of the maximum posterior variance (Lemma 3.1 and Corollary 3.2). Our proposed bound is tighter than the existing bound when the noise variances approach zero.

2. In Section 4, we analyze the GP bandit under the noiseless setting. We show a novel result that PE achieves the cumulative regret upper bound that matches the conjectured lower bound shown by Vakili (2022) under common assumptions in the GP bandit literature. Furthermore, we prove that MVR achieves the exponentially converging and near-optimal simple regret upper bounds for squared exponential (SE) and Matérn kernels, respectively. These results are summarized in Tables 1–2.

3. In Section 5, we show that the modified PE- and MVR-style algorithms achieve the near-optimal cumulative and simple regret upper bounds with respect to the RKHS norm upper bound of the reward function under

several conditions. These results are summarized in Tables 3–5.

4. In Section 6, we analyze the GP-bandit problem with the non-stationary noise variance, which is the kernelized extension of the linear bandit with heteroscedastic noise (Zhou et al., 2021). We first study the regret lower bound. Then, we show that the modified PE- and MVR-style algorithms achieve the near-optimal cumulative and simple regret upper bounds, respectively. To our knowledge, our analyses are the first for this setting, though the non-stationary noise is a frequently faced problem.

### 1.1. Related Works

The theoretical assumption of the GP bandit is twofold: Bayesian setting (Srinivas et al., 2010; De Freitas et al., 2012; Russo & Van Roy, 2014; Scarlett, 2018; Takeno et al., 2023; 2024) where the reward function follows GPs, and the frequentist setting, where the reward function lies in a known RKHS (Srinivas et al., 2010; Chowdhury & Gopalan, 2017; Vakili et al., 2021a; Li & Scarlett, 2022). Although this paper concentrates on deriving the regret upper bound for the frequentist setting, our Lemma 3.1 and Corollary 3.2 are versatile and can be applied to the Bayesian setting.

Many GP bandit algorithms have been proposed in the frequentist setting (for example, Srinivas et al., 2010; Valko et al., 2013; Chowdhury & Gopalan, 2017; Janz et al., 2020; Vakili et al., 2021a; Li & Scarlett, 2022). Although several existing methods (Valko et al., 2013; Janz et al., 2020; Camilleri et al., 2021; Salgia et al., 2021; Li & Scarlett, 2022) achieve near-optimal regret upper bounds for the ordinary GP bandit setting as summarized in (Li & Scarlett, 2022), we develop PE- and MVR-style algorithms due to their simplicity. On the other hand, although these existing methods are near-optimal regarding the time horizons, the optimality regarding the RKHS norm of the reward function has not been shown as summarized in Tables 3–5.

The regret analyses are also conducted on the noiseless setting (Bull, 2011; Lyu et al., 2019; Vakili, 2022; Salgia et al., 2024; Kim & Sanz-Alonso, 2024; Flynn & Reeb, 2025). Regarding the cumulative regret, we obtained a tighter upper bound for both SE and Matérn kernels than existing results without the additional assumption for the reward function like Assumption 4.2 in (Salgia et al., 2021). Regarding the simple regret, Kim & Sanz-Alonso (2024) have shown that the random sampling-based algorithm achieves the known-best regret upper bound in terms of the expectation regarding the algorithm's randomness. Compared with this result, we show the regret upper bounds that always hold with the deterministic MVR-style algorithm. In particular, the regret upper bound is tighter for the Matérn kernel than that from (Kim & Sanz-Alonso, 2024). Tables 1–2 summarize the

comparison.

Compared with the regret upper bound, the analysis for the regret lower bound is limited (Bull, 2011; Scarlett et al., 2017; Cai & Scarlett, 2021; Vakili, 2022). From these results, we will confirm the optimality of the GP bandit algorithms in Sections 4 and 5. In Section 6, our regret lower bound for the non-stationary noise variance setting is directly obtained from the proofs of (Bull, 2011; Scarlett et al., 2017).

The linear bandit with heteroscedastic noise, where the noise variance is non-stationary with respect to the time horizons, has been studied (Zhou et al., 2021; Zhang et al., 2021; Kim et al., 2022; Zhou & Gu, 2022; Zhao et al., 2023). These studies aim to obtain the noise variance-dependent regret upper bound, characterized by the sum of noise variances. To our knowledge, the kernelized extension of this setting has not been investigated. Furthermore, as discussed in Section 6, the direct extension from the linear bandit methods is not near-optimal.

## 2. Preliminaries

**Problem Setting.** Let $f : \mathcal{X} \to \mathbb{R}$ be an unknown reward function with compact input domain $\mathcal{X} \subset \mathbb{R}^d$. At each step $t$, a learner chooses the query point $\boldsymbol{x}_t \in \mathcal{X}$; after that, the learner observes the corresponding reward $y_t := f(\boldsymbol{x}_t) + \epsilon_t$, where $\epsilon_t$ is a mean-zero random variable. Under this setup, the learner's goal is to minimize the following cumulative regret $R_T$ or the simple regret $r_T$:

$$R_T = \sum_{t \in [T]} f(\boldsymbol{x}^*) - f(\boldsymbol{x}_t), \tag{1}$$

$$r_T = f(\boldsymbol{x}^*) - f(\widehat{\boldsymbol{x}}_T), \tag{2}$$

where $[T] = \{1, \dots, T\}$ and $\boldsymbol{x}^* \in \arg \max_{\boldsymbol{x} \in \mathcal{X}} f(\boldsymbol{x})$. Furthermore, $\widehat{\boldsymbol{x}}_T \in \mathcal{X}$ is the estimated maximizer, which is returned by the algorithm at the end of step $T$.

**Regularity Assumptions.** To construct an algorithm, we leverage the following assumptions.

**Assumption 2.1** (Smoothness of $f$)**.** Assume that $f$ be an element of RKHS $\mathcal{H}_k$ with bounded RKHS norm $\|f\|_{\mathcal{H}_k} \leq B < \infty$. Here, $\mathcal{H}_k$ and $\|f\|_{\mathcal{H}_k}$ respectively denote RKHS and its norm endowed with a known positive definite kernel $k : \mathcal{X} \times \mathcal{X} \to \mathbb{R}$. Furthermore, we assume $k(\boldsymbol{x}, \boldsymbol{x}) \leq 1$ holds for all $\boldsymbol{x} \in \mathcal{X}$.

**Assumption 2.2** (Assumption for noise)**.** The noise sequence $(\epsilon_t)_{t \in \mathbb{N}_+}$ is mutually independent. Furthermore, assume that $\epsilon_t$ is a sub-Gaussian random variable with variance proxy $\rho_t \geq 0$; namely, $\mathbb{E}[\exp(\lambda \epsilon_t)] \leq \exp(\lambda^2 \rho_t^2 / 2)$ holds for all $\lambda \in \mathbb{R}$.

In existing works (e.g., Srinivas et al., 2010), Assumption 2.1 is the standard assumption for encoding the smooth-

*Table 1.* Comparison between existing noiseless algorithms' guarantees for cumulative regret and our result. In all algorithms, the smoothness parameter of the Matérn kernel is assumed to be $\nu > 1/2$. Furthermore, $d$, $\ell$, $\nu$, and $B$ are supposed to be $\Theta(1)$ here. "Type" column shows that the regret guarantee is (D)eterministic or (P)robabilistic. Throughout this paper, the notation $\widetilde{O}(\cdot)$ represents the order notation whose poly-logarithmic dependence is ignored.

| Algorithm | Regret (SE) | Regret (Matérn) | | | Type | Remark |
|---|---|---|---|---|---|---|
| | | $\nu < d$ | $\nu = d$ | $\nu > d$ | | |
| GP-UCB (Lyu et al., 2019) (Kim & Sanz-Alonso, 2024) | $O\left(\sqrt{T \ln^d T}\right)$ | $\widetilde{O}\left(T^{\frac{\nu+d}{2\nu+d}}\right)$ | | | D | |
| Explore-then-Commit (Vakili, 2022) | N/A | $\widetilde{O}\left(T^{\frac{d}{\nu+d}}\right)$ | | | P | |
| Kernel-AMM-UCB (Flynn & Reeb, 2025) | $O\left(\ln^{d+1} T\right)$ | $\widetilde{O}\left(T^{\frac{\nu d + d^2}{2\nu^2 + 2\nu d + d^2}}\right)$ | | | D | |
| REDS (Salgia et al., 2024) | N/A | $\widetilde{O}\left(T^{\frac{d-\nu}{d}}\right)$ | $O\left(\ln^{\frac{5}{2}} T\right)$ | $O\left(\ln^{\frac{3}{2}} T\right)$ | P | Assumption for level-set is required. |
| **PE** **(our analysis)** | $O\left(\ln T\right)$ | $\widetilde{O}\left(T^{\frac{d-\nu}{d}}\right)$ | $O\left(\ln^{2+\alpha} T\right)$ | $O\left(\ln T\right)$ | D | $\alpha > 0$ is an arbitrarily fixed constant. |
| Conjectured Lower Bound (Vakili, 2022) | N/A | $\Omega\left(T^{\frac{d-\nu}{d}}\right)$ | $\Omega(\ln T)$ | $\Omega(1)$ | N/A | |

ness of the underlying reward function depending on the kernel. We focus on the following SE kernel $k_{\text{SE}}$ and Matérn kernel $k_{\text{Matérn}}$ that are commonly used in the GP bandit:

$$k_{\text{SE}}(\boldsymbol{x}, \widetilde{\boldsymbol{x}}) = \exp\left(-\frac{\|\boldsymbol{x} - \widetilde{\boldsymbol{x}}\|_2^2}{2\ell^2}\right),$$

$$k_{\text{Matérn}}(\boldsymbol{x}, \widetilde{\boldsymbol{x}}) =$$
$$\frac{2^{1-\nu}}{\Gamma(\nu)}\left(\frac{\sqrt{2\nu}\|\boldsymbol{x} - \widetilde{\boldsymbol{x}}\|_2}{\ell}\right)^\nu J_\nu\left(\frac{\sqrt{2\nu}\|\boldsymbol{x} - \widetilde{\boldsymbol{x}}\|_2}{\ell}\right),$$

where $\ell > 0$ and $\nu > 0$ are the lengthscale and smoothness parameter, respectively. Furthermore, $\Gamma(\cdot)$ and $J_\nu$ denote Gamma and modified Bessel function, respectively. Assumption 2.2 is also common in existing work (e.g., Vakili et al., 2021a; Li & Scarlett, 2022). In Sections 4–5, we consider the stationary noise variance setting whose variance proxy $\rho_t$ is fixed over time, while the non-stationary noise variance setting that allows the time-dependent variance proxies $\rho_t$ in Section 6.

**Gaussian Process.** GP is a fundamental kernel-based model that gives both the prediction and its uncertainty quantification of the underlying function. Let $\mathcal{GP}(0, k)$ be a mean-zero GP whose covariance is characterized by the kernel function $k$. In addition, let $\mathbf{X} := (\boldsymbol{x}_1, \ldots, \boldsymbol{x}_t)$ and $\boldsymbol{y} := (y_1, \ldots, y_t)^\top$ be training input and output data of GP, respectively. Then, under the Bayesian assumption that $f$ follows the GP prior $\mathcal{GP}(0, k)$, the posterior distribution of $f(\boldsymbol{x})$ given $\mathbf{X}$ and $\boldsymbol{y}$ is defined as Gaussian distribution,

whose mean $\mu_\Sigma(\boldsymbol{x}; \mathbf{X}, \boldsymbol{y})$ and variance $\sigma_\Sigma^2(\boldsymbol{x}; \mathbf{X})$ are

$$\mu_\Sigma(\boldsymbol{x}; \mathbf{X}, \boldsymbol{y}) = \boldsymbol{k}(\boldsymbol{x}, \mathbf{X})^\top (\mathbf{K}(\mathbf{X}, \mathbf{X}) + \Sigma)^{-1}\boldsymbol{y},$$
$$\sigma_\Sigma^2(\boldsymbol{x}; \mathbf{X}) = k(\boldsymbol{x}, \boldsymbol{x})$$
$$- \boldsymbol{k}(\boldsymbol{x}, \mathbf{X})^\top (\mathbf{K}(\mathbf{X}, \mathbf{X}) + \Sigma)^{-1}\boldsymbol{k}(\boldsymbol{x}, \mathbf{X}),$$

where $\mathbf{K}(\mathbf{X}, \mathbf{X}) := [k(\boldsymbol{x}, \widetilde{\boldsymbol{x}})]_{\boldsymbol{x}, \widetilde{\boldsymbol{x}} \in \mathbf{X}} \in \mathbb{R}^{t \times t}$ and $\mathbf{K}(\boldsymbol{x}, \mathbf{X}) = [k(\boldsymbol{x}, \widetilde{\boldsymbol{x}})]_{\widetilde{\boldsymbol{x}} \in \mathcal{X}} \in \mathbb{R}^t$ respectively represent the kernel matrix and vector defined by $\boldsymbol{x}$ and $\mathbf{X}$. Furthermore, $\Sigma \in \mathbb{R}^{t \times t}$ is the positive definite variance parameter matrix that defines the noise structure of the observation of GP. Namely, given the input data $\mathbf{X}$, the corresponding outputs $\boldsymbol{y}$ is assumed to be given as $\boldsymbol{y} = \boldsymbol{f}(\mathbf{X}) + \boldsymbol{\epsilon}$ under the GP modeling, where $\boldsymbol{\epsilon} \sim \mathcal{N}(\mathbf{0}, \Sigma)$ and $\boldsymbol{f}(\mathbf{X}) \sim \mathcal{N}(\mathbf{0}, K(\mathbf{X}, \mathbf{X}))$. We would like to emphasize that the above modeling assumptions ($f \sim \mathcal{GP}(0, k)$ and $\boldsymbol{\epsilon} \sim \mathcal{N}(\mathbf{0}, \Sigma)$) are "fictional" assumptions that are used only for GP modeling, and are distinct from Assumptions 2.1 and 2.2.

**Maximum Information Gain.** The maximum information gain (MIG) is the kernel-dependent complexity parameter used to characterize the regret and confidence bounds in the GP bandits. Given the variance parameter matrix $\Sigma_T := \text{diag}(\lambda_1^2, \ldots, \lambda_T^2)$, the MIG $\gamma_T(\Sigma_T)$ is defined as

$$\gamma_T(\Sigma_T) = \max_{\mathbf{X} \subset \mathcal{X}^T} I_{\Sigma_T}(\boldsymbol{f}(\mathbf{X}), \boldsymbol{y}), \tag{3}$$

where $I_{\Sigma_T}(\boldsymbol{f}(\mathbf{X}), \boldsymbol{y}) := \frac{1}{2}\ln \frac{\det(\Sigma_T + \mathbf{K}(\mathbf{X}, \mathbf{X}))}{\det(\Sigma_T)}$ denote the mutual information between $\boldsymbol{y}$ and $\boldsymbol{f}(\mathbf{X})$, under the GP modeling assumptions $\boldsymbol{f}(\mathbf{X}) \sim \mathcal{N}(\mathbf{0}, K(\mathbf{X}, \mathbf{X}))$, $\boldsymbol{y} = \boldsymbol{f}(\mathbf{X}) + \boldsymbol{\epsilon}$, and $\boldsymbol{\epsilon} \sim \mathcal{N}(\mathbf{0}, \Sigma_T)$. When $\Sigma_T = \lambda^2 \boldsymbol{I}_T$ with some fixed $\lambda > 0$ and the identity matrix $\boldsymbol{I}_T \in \mathbb{R}^{T \times T}$, the upper bound of MIG $\overline{\gamma}(T, \lambda^2)$

is known in several commonly used kernels. For example, $\gamma_T(\lambda^2 \boldsymbol{I}_T) \leq \overline{\gamma}(T, \lambda^2) = O(\ln^{d+1}(T/\lambda^2))$ and $\gamma_T(\lambda^2 \boldsymbol{I}_T) \leq \overline{\gamma}(T, \lambda^2) = O((T/\lambda^2)^{\frac{d}{2\nu+d}}(\ln(T/\lambda^2))^{\frac{2\nu}{2\nu+d}})$ in SE and Matérn kernels with $\nu > 1/2$, respectively (Vakili et al., 2021b). Furthermore, for general $\Sigma_T$, we can see $\gamma_T(\Sigma_T) \leq \gamma_T(\underline{\lambda}_T^2 \boldsymbol{I}_T)$ with $\underline{\lambda}_T^2 = \min_{t \in [T]} \lambda_t^2$ from the data processing inequality (Theorem 2.8.1 of Cover & Thomas, 2006). We show the proof in Appendix B for completeness.

**Maximum Variance Reduction and Phased Elimination.** MVR is the algorithm that sequentially chooses the most uncertain action $\boldsymbol{x}_t = \operatorname{argmax}_{\boldsymbol{x} \in \mathcal{X}} \sigma_\Sigma^2(\boldsymbol{x}; \mathbf{X})$ as shown in Algorithm 2 in Appendix G. PE is the algorithm that combines MVR and candidate elimination as shown in Algorithm 1 in Appendix G. PE divides the time horizons into batches with appropriately designed lengths and performs MVR in each batch. In PE, after each batch, the inputs whose UCB is lower than the maximum of the lower CB are eliminated from the candidates. MVR and PE achieve near-optimal simple and cumulative regret upper bounds, respectively. Due to their simplicity, we analyze MVR- and PE-style algorithms.

# 3. Uniform Upper Bound of Posterior Variance for Maximum Variance Reduction

In this section, we describe the theoretical core result that gives a new upper bound of the posterior variance for the MVR algorithm. Specifically, our result improves the existing upper bound of posterior variance when decreasing noise variance parameters.

**Lemma 3.1** (General posterior variance upper bound for MVR). *Fix any compact subset $\widetilde{\mathcal{X}} \subset \mathcal{X}$. Then, the following two statements hold:*

1. ***Stationary var.:*** *Let $(\overline{\lambda}_T)_{T \in \mathbb{N}_+}$ be a non-negative sequence, and $(\widetilde{\lambda}_T)_{T \in \mathbb{N}_+}$ be a strictly positive sequence such that $\overline{\lambda}_T \leq \widetilde{\lambda}_T$. Furthermore, for any $T \in \mathbb{N}_+$, $t \in [T]$, define $\boldsymbol{x}_{T,t} \in \widetilde{\mathcal{X}}$ as $\boldsymbol{x}_{T,t} \in \arg\max_{\boldsymbol{x} \in \widetilde{\mathcal{X}}} \sigma_{\overline{\lambda}_T^2 \boldsymbol{I}_t}(\boldsymbol{x}; \mathbf{X}_{T,t-1})$, where $\mathbf{X}_{T,t-1} = (\boldsymbol{x}_{T,1}, \ldots, \boldsymbol{x}_{T,t-1})$. Then, for any $T \in \{T \in \mathbb{N}_+ \mid T/2 \geq 3\gamma_T(\widetilde{\lambda}_T^2 \boldsymbol{I}_T)\}$, the following inequality holds:*

$$\max_{\boldsymbol{x} \in \widetilde{\mathcal{X}}} \sigma_{\overline{\lambda}_T^2 \boldsymbol{I}_T}(\boldsymbol{x}; \mathbf{X}_{T,T}) \leq \frac{4}{T} \sqrt{\widetilde{\lambda}_T^2 T \gamma_T(\widetilde{\lambda}_T^2 \boldsymbol{I}_T)}. \quad (4)$$

2. ***Non-stationary var.:*** *Let $(\lambda_t)_{t \in \mathbb{N}_+}$ be a non-negative sequence, and $(\widetilde{\lambda}_t)_{t \in \mathbb{N}_+}$ be a strictly positive sequence such that $\lambda_t \leq \widetilde{\lambda}_t$. Furthermore, for any $t \in \mathbb{N}_+$, define $\boldsymbol{x}_t \in \widetilde{\mathcal{X}}$ as $\boldsymbol{x}_t \in \arg\max_{\boldsymbol{x} \in \widetilde{\mathcal{X}}} \sigma_{\Sigma_{t-1}}(\boldsymbol{x}; \mathbf{X}_{t-1})$, where $\mathbf{X}_{t-1} = (\boldsymbol{x}_1, \ldots, \boldsymbol{x}_{t-1})$ and $\Sigma_{t-1} =$*

$\operatorname{diag}(\lambda_1^2, \ldots, \lambda_{t-1}^2)$. *Then, for any $T \in \{T \in \mathbb{N}_+ \mid T/2 \geq 4\gamma_T(\widetilde{\Sigma}_T)\}$,*

$$\max_{\boldsymbol{x} \in \widetilde{\mathcal{X}}} \sigma_{\Sigma_T}(\boldsymbol{x}; \mathbf{X}_T) \leq \frac{4}{T} \sqrt{\left(\sum_{t=1}^T \widetilde{\lambda}_t^2\right) \gamma_T(\widetilde{\Sigma}_T)}, \quad (5)$$

*where $\widetilde{\Sigma}_t = \operatorname{diag}(\widetilde{\lambda}_1^2, \ldots, \widetilde{\lambda}_t^2)$.*

To make the above statements explicit, we give the following corollary for $k = k_{\mathrm{SE}}$ and $k = k_{\mathrm{Matérn}}$ with the stationary variance parameter as a special case of Lemma 3.1. The proof is in Appendix C.

**Corollary 3.2.** *Suppose the assumptions in statement 1 of Lemma 3.1. Then, the following four statements hold:*

1. *Suppose $k = k_{\mathrm{SE}}$ and fix any $\alpha > 0$. If $\overline{\lambda}_T^2 = \Omega(\exp(-T^{\frac{1}{d+1}} \ln^{-\alpha}(1 + T)))$, Eq. (4) holds with $\widetilde{\lambda}_T^2 = \overline{\lambda}_T^2$ for all $T \geq \overline{T}$, where $\overline{T} < \infty$ is the constant that depends on $\mathcal{X}$, $\alpha$, $d$, and $\ell$.*

2. *Suppose $k = k_{\mathrm{Matérn}}$ with $\nu > 1/2$ and fix any $\alpha > 0$. If $\overline{\lambda}_T^2 = \Omega(T^{-\frac{2\nu}{d}}(\ln(1 + T))^{\frac{2\nu(1+\alpha)}{d}})$, Eq. (4) holds with $\widetilde{\lambda}_T^2 = \overline{\lambda}_T^2$ for all $T \geq \overline{T}$, where $\overline{T} < \infty$ is the constant that depends on $\mathcal{X}$, $\alpha$, $d$, $\ell$, and $\nu$.*

3. *Suppose $k = k_{\mathrm{SE}}$ and fix any $\alpha > 0$ and $C > 0$. If $\forall T \in \mathbb{N}_+, \overline{\lambda}_T^2 < C \exp\left(-T^{\frac{1}{d+1}} \ln^{-\alpha}(1 + T)\right)$ (including $\overline{\lambda}_T = 0$), the following inequality holds for all $T \geq \overline{T}$:*

$$\max_{\boldsymbol{x} \in \widetilde{\mathcal{X}}} \sigma_{\overline{\lambda}_T^2 \boldsymbol{I}_T}(\boldsymbol{x}; \mathbf{X}_{T,T}) \leq O\left(\sqrt{\exp\left(-T^{\frac{1}{d+1}} \ln^{-\alpha} T\right)}\right),$$

*where $\overline{T} < \infty$ and the implied constant of the above inequality depend on $\mathcal{X}$, $\alpha$, $d$, $C$, and $\ell$.*

4. *Suppose $k = k_{\mathrm{Matérn}}$ with $\nu > 1/2$ and fix any $\alpha > 0$ and $C > 0$. If $\forall T \in \mathbb{N}_+, \overline{\lambda}_T^2 < CT^{-\frac{2\nu}{d}}(\ln(1 + T))^{\frac{2\nu(1+\alpha)}{d}}$ (including $\overline{\lambda}_T = 0$), the following inequality holds for all $T \geq \overline{T}$:*

$$\max_{\boldsymbol{x} \in \widetilde{\mathcal{X}}} \sigma_{\overline{\lambda}_T^2 \boldsymbol{I}_T}(\boldsymbol{x}; \mathbf{X}_{T,T}) \leq O\left(T^{-\frac{\nu}{d}}(\ln T)^{\frac{\nu(1+\alpha)}{d}}\right), \quad (6)$$

*where $\overline{T} < \infty$ and the implied constant of the above inequality depend on $\mathcal{X}$, $\alpha$, $d$, $C$, $\ell$, and $\nu$.*

*In all of the above statements, $\overline{T}$ increases as decreasing of $\alpha$, and $\overline{T} \to \infty$ as $\alpha \to 0$.*

**Comparison with Existing Upper Bound.** Here, we compare statement 1 in Lemma 3.1 with the existing upper bound. By considering the case $\widetilde{\lambda}_T^2 = \overline{\lambda}_T^2$, our result shows $O\left(T^{-1}\sqrt{\overline{\lambda}_T^2 T \gamma_T(\overline{\lambda}_T^2 \boldsymbol{I}_T)}\right)$ upper bound of posterior standard deviation under the sublinear increasing condition of MIG so that $T/2 \geq 3\gamma_T(\overline{\lambda}_T^2 \boldsymbol{I}_T)$ holds. On the other hand, the existing analysis of MVR[1] implies

$$
\max_{\boldsymbol{x} \in \widetilde{\mathcal{X}}} \sigma_{\overline{\lambda}_T^2 \boldsymbol{I}_T}(\boldsymbol{x}; \mathbf{X}_{T,T})
$$
$$
\leq
\begin{cases}
O\left(\frac{1}{T}\sqrt{\overline{\lambda}_T^2 T \gamma_T(\overline{\lambda}_T^2 \boldsymbol{I}_T)}\right) & \text{if } \overline{\lambda}_T^2 = \Omega(1), \\
O\left(\frac{1}{T}\sqrt{\frac{T\gamma_T(\overline{\lambda}_T^2 \boldsymbol{I}_T)}{\ln\left(1+\overline{\lambda}_T^{-2}\right)}}\right) & \text{if } \overline{\lambda}_T^2 = o(1).
\end{cases} \quad (7)
$$

Since $\overline{\lambda}_T^2 \leq 1/\ln(1+\overline{\lambda}_T^{-2})$, our analysis improves the noise parameter dependence on the decreasing regime of $\overline{\lambda}_T^2$. Furthermore, several recent noiseless GP bandit works also derive the related result to Lemma 3.1 or Corollary 3.2. Flynn & Reeb (2025) consider the noiseless setting by relying on elliptical potential count lemma (Lemma 3.3 below), and the naive adaptation of their analysis leads[2]

$$
\max_{\boldsymbol{x} \in \widetilde{\mathcal{X}}} \sigma_{\overline{\lambda}_T^2 \boldsymbol{I}_T}(\boldsymbol{x}; \mathbf{X}_{T,T})
$$
$$
\leq O\left(\frac{\gamma_T(\overline{\lambda}_T^2 \boldsymbol{I}_T)}{T} + \frac{\sqrt{\overline{\lambda}_T^2 T \gamma_T(\overline{\lambda}_T^2 \boldsymbol{I}_T)}}{T}\right).
$$

In the above equation, the decreasing speed of $\overline{\lambda}_T^2$ is more restricted than that of Corollary 3.2 to obtain the same order upper bound as Eq. (4). For example, if $k = k_{\text{Matérn}}$ and $\overline{\lambda}_T^2 = \Omega(T^{-\frac{2\nu}{d}}(\ln(1+T))^{\frac{2\nu(1+\alpha)}{d}})$ as with the condition of statement 4 in Corollary 3.2, the first term $\frac{\gamma_T(\overline{\lambda}_T^2 \boldsymbol{I}_T)}{T}$ dominates the second term since the existing upper bound of MIG implies $\frac{\gamma_T(\overline{\lambda}_T^2 \boldsymbol{I}_T)}{T} = \widetilde{O}(1)$. Furthermore, Salgia et al. (2024) derives a similar result to our statement 4 with a random sampling algorithm instead of MVR. The main theoretical advantage of our result is that the constant $\overline{T}$ depends on the entire input space $\mathcal{X}$ instead of the subset $\widetilde{\mathcal{X}}$, while that of (Salgia et al., 2024) has the dependence on $\widetilde{\mathcal{X}}$. This dependence on $\widetilde{\mathcal{X}}$ raises the requirement of the additional level-set assumption to apply the PE-style

algorithm in noiseless feedback (see Assumption 4.2 in (Salgia et al., 2024)). Furthermore, as described in the proof sketch below, our proof mainly relies on the simple extension of the well-known information gain arguments from (Srinivas et al., 2010), not on the technique of (Salgia et al., 2024) that involves the theoretical tools from function approximation literature.

**Proof sketch of Lemma 3.1.** Here, since statement 2 is derived as the extension of the proof of statement 1, we only describe the proof sketch of statement 1 for simplicity. We leave the full proof, including statement 2, in Appendix C. Our proof is based on the following two observations:

1. For any index set $\mathcal{T} \subset [T]$, $\max_{\boldsymbol{x} \in \widetilde{\mathcal{X}}} \sigma_{\widetilde{\lambda}_T^2 \boldsymbol{I}_T}(\boldsymbol{x}; \mathbf{X}_{T,T})$ can be bounded from above by the average observed posterior standard deviation on $\mathcal{T}$ from the definition of MVR. Namely, $\max_{\boldsymbol{x} \in \widetilde{\mathcal{X}}} \sigma_{\widetilde{\lambda}_T^2 \boldsymbol{I}_T}(\boldsymbol{x}; \mathbf{X}_{T,T}) \leq \frac{1}{|\mathcal{T}|}\sum_{t \in \mathcal{T}} \sigma_{\widetilde{\lambda}_T^2 \boldsymbol{I}_{t-1}}(\boldsymbol{x}_{T,t}; \mathbf{X}_{T,t-1})$ holds.

2. If we set $\mathcal{T} = \{t \in [T] \mid \widetilde{\lambda}_T^{-1}\sigma_{\widetilde{\lambda}_T^2 \boldsymbol{I}_{t-1}}(\boldsymbol{x}_{T,t}; \mathbf{X}_{T,t-1}) \leq 1\}$,

$$
\sigma_{\widetilde{\lambda}_T^2 \boldsymbol{I}_{t-1}}^2(\boldsymbol{x}_{T,t}; \mathbf{X}_{T,t-1})
$$
$$
= \widetilde{\lambda}_T^2 \min\left\{1, \widetilde{\lambda}_T^{-2}\sigma_{\widetilde{\lambda}_T^2 \boldsymbol{I}_{t-1}}^2(\boldsymbol{x}_{T,t}; \mathbf{X}_{T,t-1})\right\} \quad (8)
$$
$$
\leq 2\widetilde{\lambda}_T^2 \ln\left(1 + \widetilde{\lambda}_T^{-2}\sigma_{\widetilde{\lambda}_T^2 \boldsymbol{I}_{t-1}}^2(\boldsymbol{x}_{T,t}; \mathbf{X}_{T,t-1})\right).
$$

for all $t \in \mathcal{T}$. We use $\forall a \geq 0, \min\{1, a\} \leq 2\ln(1+a)$ in the last line. By relying on the standard MIG-based analysis (e.g., Theorem 5.3, 5.4 in (Srinivas et al., 2010)) and the assumption $\overline{\lambda}_T \leq \widetilde{\lambda}_T$, the above inequality implies

$$
\sum_{t \in \mathcal{T}} \sigma_{\widetilde{\lambda}_T^2 \boldsymbol{I}_{t-1}}(\boldsymbol{x}_{T,t}; \mathbf{X}_{T,t-1}) \leq 2\sqrt{\widetilde{\lambda}_T^2 T \gamma_T(\widetilde{\lambda}_T^2 \boldsymbol{I}_T)}. \quad (9)
$$

From the above two observations, the remaining interest is the increasing speed of $|\mathcal{T}|$. We use the following lemma, which we call *elliptical potential count* lemma in (Flynn & Reeb, 2025), as the analogy of elliptical potential arguments in linear bandits.

**Lemma 3.3** (Elliptical potential count lemma, Lemma D.9 in (Flynn & Reeb, 2025)). *Fix any $T \in \mathbb{N}_+$, any sequence $\boldsymbol{x}_1, \ldots, \boldsymbol{x}_T \in \mathcal{X}$, and any $\overline{\lambda} > 0$. Set $\mathcal{T}^c$ as $\mathcal{T}^c = \{t \in [T] \mid \overline{\lambda}^{-1}\sigma_{\overline{\lambda}^2 \boldsymbol{I}_{t-1}}(\boldsymbol{x}_t; \mathbf{X}_{t-1}) > 1\}$, where $\mathbf{X}_t = (\boldsymbol{x}_1, \ldots, \boldsymbol{x}_t)$. Then, the number of elements of $\mathcal{T}^c$ satisfies $|\mathcal{T}^c| \leq \min\left\{3\overline{\gamma}\left(3\gamma_T(\overline{\lambda}^2 \boldsymbol{I}_T), \overline{\lambda}^2\right), 3\gamma_T(\overline{\lambda}^2 \boldsymbol{I}_T)\right\}$. Furthermore, $\overline{\gamma}(\cdot, \cdot)$ is any monotonic upper bound of MIG defined on $\mathbb{R}_+ \times \mathbb{R}_+$, which satisfies $\forall T \in \mathbb{N}_+, \lambda > 0, \gamma_T(\lambda^2 \boldsymbol{I}_T) \leq \overline{\gamma}(T, \lambda^2)$ and $\forall \lambda > 0, T \geq 1, \epsilon \geq 0, \overline{\gamma}(T, \lambda^2) \leq \overline{\gamma}(T+\epsilon, \lambda^2)$.*

---

[1]Eq. (7) also holds for any algorithm by replacing $\max_{\boldsymbol{x} \in \widetilde{\mathcal{X}}} \sigma_{\overline{\lambda}_T^2 \boldsymbol{I}_T}(\boldsymbol{x}; \mathbf{X}_{T,T})$ with $\frac{1}{T}\sum_{t=1}^{T} \sigma_{\overline{\lambda}_T^2 \boldsymbol{I}_T}(\boldsymbol{x}_{T,t}; \mathbf{X}_{T,t-1})$. For example, GP-UCB (Srinivas et al., 2010; Chowdhury & Gopalan, 2017) and GP-TS (Chowdhury & Gopalan, 2017) use the upper bound of $\frac{1}{T}\sum_{t=1}^{T} \sigma_{\overline{\lambda}_T^2 \boldsymbol{I}_T}(\boldsymbol{x}_{T,t}; \mathbf{X}_{T,t-1})$.

[2]In some settings of $\overline{\lambda}_T^2$, the first term of r.h.s. $\gamma_T(\overline{\lambda}_T^2 \boldsymbol{I}_T)/T$ can be small by putting $\min\left\{\overline{\gamma}\left(\gamma_T(\overline{\lambda}_T^2 \boldsymbol{I}_T), \overline{\lambda}_T^2\right), \gamma_T(\overline{\lambda}_T^2 \boldsymbol{I}_T)\right\}$ in the first term, instead of $\gamma_T(\overline{\lambda}_T^2 \boldsymbol{I}_T)$. See Lemma 3.3.

From the above lemma, we obtain the lower bound of $\mathcal{T}$ as $|\mathcal{T}| \geq T - 3\gamma_T(\widetilde{\lambda}_T^2 \boldsymbol{I}_T)$. Finally, we obtain the desired result by noting $|\mathcal{T}| \geq T/2$ holds for any $T \in \{T \in \mathbb{N}_+ \mid T/2 \geq 3\gamma_T(\widetilde{\lambda}_T^2 \boldsymbol{I}_T)\}$.

## 4. Noiseless Setting

As a first application of our result, we study a noiseless setting; namely, we focus on the setting where $\rho_t = 0$ for all $t \in \mathbb{N}_+$ in Assumption 2.2. The following results show our cumulative and simple regret guarantees for PE and MVR.

**Theorem 4.1** (Cumulative Regret Bound for PE.). *Suppose Assumptions 2.1 and 2.2 hold with $\rho_t = 0$ for all $t \in \mathbb{N}_+$. Furthermore, assume $B$, $d$, $\ell$, and $\nu$ are $\Theta(1)$. Then, when running Algorithm 1 with $\beta^{1/2} = B$, $\lambda = 0$, and any fixed $N_1 \in \mathbb{N}_+$, the following statements hold:*

- *If $k = k_{\mathrm{SE}}$, $R_T = O(\ln T)$.*

- *If $k = k_{\mathrm{Matérn}}$ with $\nu > 1/2$,*

$$R_T = \begin{cases} \widetilde{O}(T^{\frac{d-\nu}{d}}) & \text{if } \nu < d, \\ O((\ln T)^{2+\alpha}) & \text{if } \nu = d, \\ O(\ln T) & \text{if } \nu > d. \end{cases} \quad (10)$$

*Here, $\alpha > 0$ is an arbitrarily fixed constant.*

**Theorem 4.2** (Simple Regret Bound for MVR.). *Suppose the same conditions as those of Theorem 4.1. Then, when running Algorithm 2 with $\lambda = 0$, the following statements hold:*

- *If $k = k_{\mathrm{SE}}$, $r_T = O\left(\exp\left(-\frac{1}{2}T^{\frac{1}{d+1}}\ln^{-\alpha} T\right)\right)$.*

- *If $k = k_{\mathrm{Matérn}}$ with $\nu > 1/2$, $r_T = \widetilde{O}\left(T^{-\frac{\nu}{d}}\right)$.*

*Remark* 4.3. The above theorems assume that the learner can exactly choose $\boldsymbol{x}_t$ in the algorithms, which is unreasonable for a continuous domain $\mathcal{X}$. However, a similar guarantee, which is worse by an additional $\sqrt{\ln T}$ multiplicative factor than the above results, can be obtained by the existing analysis (Li & Scarlett, 2022) under the additional Lipschitz assumption for $f$. Note that such Lipschitz assumption for $f$ automatically holds under fixed $B$ when we set $k = k_{\mathrm{SE}}$ or $k = k_{\mathrm{Matérn}}$ with $\nu > 1$ (Lee et al., 2022).

The proofs of Theorems 4.1 and 4.2 are respectively derived by directly following the standard analysis of PE and MVR with statements 3 and 4 of Corollary 3.2. We describe full proofs in Appendix D for completeness.

**Discussion.** As summarized in Tables 1–2, our results are the same as or superior to the best-known upper bounds in almost all cases. The only exception is the simple regret

with $k = k_{\mathrm{SE}}$, whose polynomial factor in exponential gets worse from $-(\alpha + 1/d)$ into $-1/(d+1)$, compared to the algorithm of (Kim & Sanz-Alonso, 2024). Roughly speaking, the numerator of the factor $-1/(d+1)$ in our analysis comes from the exponent of the upper bound of MIG $O(\ln^{d+1}(T/\lambda^2))$. We expect our simple regret has room of improvement from $\widetilde{O}(\exp(-T^{\frac{1}{d+1}}\ln^{-\alpha} T))$ into $\widetilde{O}(\exp(-T^{\frac{2}{d}}\ln^{-\alpha} T))$ in future work, since the conjectured best upper bound of MIG is $O(\ln^{d/2}(T/\lambda^2))$ from the regret lower bound (Scarlett et al., 2017).

## 5. Optimal Dependence of RKHS Norm Upper Bound

As the second application of our result, we consider improving the existing dependence of RKHS norm upper bound $B$ in the regret upper bounds.

### 5.1. Simple Regret

The following theorem shows our results for simple regret.

**Theorem 5.1** (Simple Regret Bound for MVR.). *Suppose Assumptions 2.1 and 2.2 hold with $\rho_t = \rho > 0$ for all $t \in \mathbb{N}_+$. Furthermore, assume $\rho$, $d$, $\ell$, and $\nu$ are $\Theta(1)$, and $\mathcal{X}$ is finite. Then, when running Algorithm 2 with $\lambda^2 = \Theta(B^{-2})$, the following statements hold for any fixed $\alpha > 0$ with probability at least $1 - \delta$:*

- *If $k = k_{\mathrm{SE}}$, $B = O\left(\exp\left(T^{\frac{1}{d+1}}\ln^{-\alpha}(1+T)\right)\right)$, and $T \geq \overline{T}$, then, $r_T = O\left(\sqrt{\frac{\ln^{d+1}(TB^2)}{T}}\right)$. Here, $\overline{T}$ is the constant, defined in statement 1 of Corollary 3.2.*

- *If $k = k_{\mathrm{Matérn}}$ with $\nu > 1/2$, $B = O\left(T^{\frac{\nu}{d}}\ln^{\frac{-\nu(1+\alpha)}{d}}(1+T)\right)$, and $T \geq \overline{T}$, then, $r_T = \widetilde{O}\left(B^{\frac{d}{2\nu+d}}T^{-\frac{\nu}{2\nu+d}}\right)$. Here, $\overline{T}$ is the constant defined in statement 2 of Corollary 3.2.*

The important point is that the setting of the noise parameter $\lambda^2 = \Theta(1/B^2)$ depends on $B$. We describe the full proof of the above theorem in Appendix E.

*Remark* 5.2. If we allow an additional logarithmic factor, we can eliminate the finiteness assumption of $\mathcal{X}$ in Theorem 5.1 by relying on the discretization with $1/T$-net as with Remark 4.3. The notable point we have to care about is that the Lipschitz constant is given as $B$ in the existing result (Lee et al., 2022). Therefore, the extension of Theorem 5.1 for continuous domain requires $1/(BT)$-net to maintain the order of $B$, and the resulting regret upper bound suffers from additional $\sqrt{\ln(TB)}$ factor, instead of $\sqrt{\ln T}$ factor derived from the standard discretizing argument that does not care the dependence of $B$ (Li & Scarlett, 2022).

**Discussion.** In both kernels, the polynomial dependence of $B$ matches the lower bound in (Scarlett et al., 2017), while there exists room for improvement in the logarithmic factor. On the other hand, there exist some exceptional cases that Theorem 5.1 does not cover, even though its lower bound of the simple regret is guaranteed to converge to 0. For example, when $B = \Theta(T^{\frac{\nu}{d}} \ln^{-\frac{\nu}{d}} T)$, the lower bound of (Scarlett et al., 2017) suggest that some algorithm find any $\epsilon$-optimal point for sufficiently large $T$ (namely, simple regret converges to 0), while violating our assumption. As with the discussion in Section 4, this limitation can be eliminated in the future if the upper bound of MIG matches the conjectured best upper bound.

### 5.2. Cumulative Regret

The following theorem also shows that the PE algorithm can achieve optimal dependence of $B$ up to a poly-logarithmic factor.

**Theorem 5.3** (Cumulative Regret Bound for PE.). *Suppose Assumptions 2.1 and 2.2 hold with $\rho_t = \rho > 0$ for all $t \in \mathbb{N}_+$. Furthermore, assume $\rho$, $d$, $\ell$, and $\nu$ are $\Theta(1)$, and $\mathcal{X}$ is finite. Then, when running Algorithm 1 with $\beta^{1/2} = (B + \rho\lambda^{-1})\sqrt{2 \ln \frac{2|\mathcal{X}|(1+\log_2 T)}{\delta}}$, $\lambda^2 = \Theta(B^{-2})$, and any fixed $N_1 \in \mathbb{N}_+$, the following statements hold with probability at least $1 - \delta$:*

- *If $k = k_{\mathrm{SE}}$ and $B = O(\sqrt{T})$, then, $R_T = O\left((\ln T)\sqrt{T\left(\ln^{d+1}(TB^2)\right)\left(\ln \frac{|\mathcal{X}|}{\delta}\right)}\right)$.*

- *If $k = k_{\mathrm{Matérn}}$ with $\nu > 1/2$ and $B = O\left(T^{\frac{2\nu^2+3\nu d}{4d^2+4\nu^2+6\nu d}}\right)$, then, $R_T = \widetilde{O}\left(T^{\frac{\nu+d}{2\nu+d}} B^{\frac{d}{2\nu+d}}\right)$.*

See Appendix E for the proof. In contrast to the analysis of the MVR, the analysis of PE cannot leverage Corollary 3.2 by setting $\lambda^2 = \Theta(B^{-2})$. Intuitively, this is because the existence of the common constant $\overline{T}$ over each batch is not guaranteed since $\lambda^2$ depends only on $T$, not the total step size of each batch. Due to this limitation, the above result is proved by leveraging Lemma 3.3 as with the analysis of (Flynn & Reeb, 2025), instead of using Corollary 3.2. As a result, the conditions about $B$ are more restricted than those of Theorem 5.1, due to the fundamental limitation of the analysis of (Flynn & Reeb, 2025) as previously discussed in Section 3. For example, if $k = k_{\mathrm{Matérn}}$, the increasing speed of $B = O\left(T^{\frac{2\nu^2+3\nu d}{4d^2+4\nu^2+6\nu d}}\right)$ in Theorem 5.3 is more restricted than $B = \widetilde{O}\left(T^{\frac{\nu}{d}}\right)$ in Theorem 5.1, regardless the fact that lower bound (Scarlett et al., 2017) suggest the sublinear cumulative regret is achievable when $B = o(T^{\frac{\nu}{d}})$. We leave future research to break this limitation.

## 6. Non-Stationary Variance

As the third application of our result, we consider the non-stationary variance setting, which falls between a noiseless and a noisy regime. In this setting, our goal is to quantify the regret by the cumulative variance proxy $V_T = \sum_{t \in [T]} \rho_t^2$. That is, we aim to construct an algorithm that achieves better performance than the one for the stationary noise setting if $V_T$ increases sublinearly. While the non-stationary variance setting has already been studied and motivated in the linear bandits (Zhou et al., 2021), to our knowledge, no existing GP-bandits literature exists for this problem. Therefore, in Appendix J, we describe some potential applications to motivate non-stationary variance setting in GP-bandits.

By following the existing works (Zhou et al., 2021; Zhang et al., 2021; Zhou & Gu, 2022), we suppose that the learner can access true variance proxy $\rho_t^2$ at the end of step $t$. We leave the unknown $\rho_t^2$ setting for future research. Note that, as described later, the direct extension of the existing linear bandit algorithm with non-stationary variance does not lead to the near-optimal guarantee.

**Lower bound.** Since the stationary noise problem with $\rho^2 = V_T/T$ is subsumed in the non-stationary problem with the cumulative variance proxy $V_T$, the following lower bounds are obtained as the corollary of the existing stationary variance lower bound (Scarlett et al., 2017) and noiseless lower bound (Bull, 2011).

**Corollary 6.1** (Lower bound for cumulative regret). *Let $\mathcal{X}$ be $\mathcal{X} = [0,1]^d$. Furthermore, assume $V_T = \Omega(1)$ and $V_T = O(T^2)$ with sufficiently small implied constant. Then, for any algorithm, there exists a GP bandit problem instance that satisfies Assumptions 2.1 and 2.2 with $\sum_{t \in [T]} \rho_t^2 = V_T$ and the following two statements:*

- *If $k = k_{\mathrm{SE}}$, $\mathbb{E}[R_T] = \Omega\left(\sqrt{V_T \ln^{\frac{d}{2}} \frac{T^2}{V_T}}\right)$.*

- *If $k = k_{\mathrm{Matérn}}$, $\mathbb{E}[R_T] = \Omega\left(V_T^{\frac{\nu}{2\nu+d}} T^{\frac{d}{2\nu+d}}\right)$.*

*Here, we assume $B$, $d$, $\ell$, and $\nu$ are $\Theta(1)$.*

**Corollary 6.2** (Lower bound for simple regret). *Let $\mathcal{X}$ be $\mathcal{X} = [0,1]^d$. Furthermore, assume $V_T = \Omega(1)$. If there exists an algorithm such that $\mathbb{E}[r_T] \le \epsilon$ hold for all problem instances that satisfy Assumptions 2.1 and 2.2 with $\sum_{t \in [T]} \rho_t^2 = V_T$, then:*

- *For $k = k_{\mathrm{SE}}$, the total step size $T$ needs to satisfy $T \ge \Omega\left(\sqrt{\frac{V_T}{\epsilon^2} \ln^{\frac{d}{2}} \frac{1}{\epsilon}}\right)$.*

- *For $k = k_{\mathrm{Matérn}}$, the total step size $T$ needs to satisfy*

$$T \geq \begin{cases} \Omega\left(\sqrt{\frac{V_T}{\epsilon^2}\left(\frac{1}{\epsilon}\right)^{\frac{d}{\nu}}}\right) & \text{if } d \leq 2\nu \text{ or } V_T = \Omega(T^{\frac{d-2\nu}{d}}), \\ \Omega\left(\left(\frac{1}{\epsilon}\right)^{\frac{d}{\nu}}\right) & \text{if } d > 2\nu \text{ and } V_T = O(T^{\frac{d-2\nu}{d}}). \end{cases}$$

*Here, we assume $B$, $d$, $\ell$, and $\nu$ are $\Theta(1)$. Furthermore, $\epsilon > 0$ is a sufficiently small constant.*

The lower bound $\Omega\left((1/\epsilon)^{\frac{d}{\nu}}\right)$ for $k_{\text{Matérn}}$ if $d > 2\nu$ and $V_T = O(T^{\frac{d-2\nu}{d}})$ in Corollary 6.2 come from (Bull, 2011), and others come from (Scarlett et al., 2017).

Note that the noiseless lower bound for expected regret also holds for noisy settings since an expected regret in the noisy setting can always be reduced to one in the noiseless setting, whose algorithm randomness is induced by observation noise. Interestingly, the above simple regret lower bound indicates that if $d > 2\nu$ and $V_T = O(T^{\frac{d-2\nu}{d}})$, the non-stationary variance setting may have the same level of difficulty as that of the noiseless problem. Our VA-MVR algorithm proposed below justifies this fact by providing a simple regret upper bound matching the above lower bound.

**Algorithm.** Algorithms 3 and 4 in Appendix H show PE and MVR-based algorithms for non-stationary variance problems, which we call variance-aware PE (VA-PE) and MVR (VA-MVR), respectively. The algorithms themselves are the variant of the standard PE or MVR algorithms that directly set the true variance proxy $\rho_t^2$ to the noise variance parameter $\lambda_t^2$ for the heteroscedastic GP model.

**Theoretical analysis.** The following theorems give the cumulative and simple regret guarantees for VA-PE and VA-MVR, respectively.

**Theorem 6.3** (Cumulative regret upper bound for VA-PE). *Suppose Assumptions 2.1 and 2.2, and $|\mathcal{X}| < \infty$ holds. Furthermore, assume $V_T := \sum_{t \in [T]} \rho_t^2 = \Omega(1)$. Then, when running Algorithm 3 with $\beta^{1/2} = \left(B + \sqrt{2 \ln \frac{2|\mathcal{X}|(1+\log_2 T)}{\delta}}\right)$, with probability at least $1-\delta$,*
$$R_T = O\left((\ln T)\sqrt{V_T\left(\ln^{d+1}\frac{T^2}{V_T}\right)\left(\ln\frac{|\mathcal{X}|}{\delta}\right)}\right) \text{ if } k = k_{\text{SE}}.$$
*Furthermore, if $k = k_{\text{Matérn}}$,*
$$R_T = \begin{cases} \widetilde{O}\left(V_T^{\frac{\nu}{2\nu+d}}T^{\frac{d}{2\nu+d}}\right) & \text{if } d \leq 2\nu, \\ \widetilde{O}\left(V_T^{\frac{\nu}{2\nu+d}}T^{\frac{d}{2\nu+d}}\right) & \text{if } d > 2\nu, V_T = \Omega\left(T^{\frac{d-2\nu}{d}}\right), \\ \widetilde{O}\left(T^{\frac{d-\nu}{d}}\right) & \text{if } d > 2\nu, V_T = O\left(T^{\frac{d-2\nu}{d}}\right). \end{cases}$$

**Theorem 6.4** (Simple regret upper bound for VA-MVR). *Suppose Assumptions 2.1 and 2.2, and $|\mathcal{X}| < \infty$ holds. Furthermore, assume $V_T = \Omega(1)$. Then, when running Algorithm 4, with probability at least $1 - \delta$, $r_T =$*

$O\left(\sqrt{\frac{V_T}{T^2}\left(\ln^{d+1}\frac{T^2}{V_T}\right)\left(\ln\frac{|\mathcal{X}|}{\delta}\right)}\right)$ *if $k = k_{\text{SE}}$. Furthermore, if $k = k_{\text{Matérn}}$,*

$$r_T = \begin{cases} \widetilde{O}\left(V_T^{\frac{\nu}{2\nu+d}}T^{-\frac{2\nu}{2\nu+d}}\right) & \text{if } d \leq 2\nu, \\ \widetilde{O}\left(V_T^{\frac{\nu}{2\nu+d}}T^{-\frac{2\nu}{2\nu+d}}\right) & \text{if } d > 2\nu, V_T = \Omega\left(T^{\frac{d-2\nu}{d}}\right), \\ \widetilde{O}\left(T^{-\frac{\nu}{d}}\right) & \text{if } d > 2\nu, V_T = O\left(T^{\frac{d-2\nu}{d}}\right). \end{cases}$$

In both results, the regret upper bound matches the lower bound up to the logarithmic factor, except for the cumulative regret guarantee for $d > 2\nu$, $V_T = O\left(T^{\frac{d-2\nu}{d}}\right)$ in $k = k_{\text{Matérn}}$. However, note that the resulting regret $\widetilde{O}\left(T^{\frac{d-\nu}{d}}\right)$ in this exceptional case matches the conjectured lower bound (Vakili, 2022); Therefore, as with our simple regret lower bound, $\widetilde{O}\left(T^{\frac{d-\nu}{d}}\right)$ upper bound in our analysis has no room for improvement if the conjectured lower bound in (Vakili, 2022) is true.

**Comparison with the stationary setting.** When $V_T = \Theta(T)$, our result matches the existing $\widetilde{O}(T^{\frac{\nu+d}{2\nu+d}})$ upper bound for the stationary setting. If we consider the setting that $V_T$ increases sublinearly, our algorithm achieves a smaller regret than the existing stationary lower bounds. For example, if $V_T = \Theta(1)$ in $k = k_{\text{SE}}$, the resulting regret becomes logarithmcally increasing regret $R_T = \widetilde{O}(1)$, while the regret lower bound for stationary variance setting is $\widetilde{\Omega}(\sqrt{T})$.

**Comparison with the algorithm in heteroscedastic linear bandits.** In heteroscedastic linear bandits, the weighted OFUL+ algorithm, which is known to achieve nearly optimal regret with UCB-based algorithm construction, is proposed in the known $\rho_t^2$ setting. We can also consider the extension of weighted OFUL+ to the GP bandits by constructing a UCB-based score with a heteroscedastic GP model. We call this extension variance-aware GP-UCB (VA-GP-UCB), and give the details in Appendix I. However, the regret of VA-GP-UCB becomes strictly worse than VA-PE and VA-MVR due to the following two reasons: Firstly, as with the stationary adaptive confidence bound (e.g., Lemma 3.11 in (Abbasi-Yadkori, 2013)), the existing adaptive confidence bound for heteroscedastic GP-model (Kirschner & Krause, 2018) contains $O(\sqrt{\gamma_t(\Sigma_t)})$ factor in the confidence width parameter, which leads to the sub-optimal order of the regret. Secondly, our analysis of VA-GP-UCB relies on the extension of the elliptical potential count lemma (Lemma C.1) to a heteroscedastic GP model, which could result in worse dependence of the noise variance parameters than that of Lemma 3.1 (See the discussion in Section 3). To our knowledge, the existing technical tools provide no direct way to avoid the above two issues.

Finally, we give the summary of our results for the non-stationary variance setting in Tables 6–8 in Appendix A.

## 7. Conclusion

We study the GP-bandit problem with the following three settings: (i) noiseless observation, (ii) varying RKHS norm, and (iii) non-stationary variance setting. We first propose a new uniform upper bound of the posterior standard deviation of GP in the MVR algorithm. By leveraging this upper bound, we refine the regret guarantee of the existing PE and MVR algorithms. Our derived upper bound matches the lower bound up to the logarithmic factor in the aforementioned three settings.

## Acknowledgements

This work was supported by JST ACT-X Grant Number JPMJAX23CD, JST PRESTO Grant Number JP-MJPR24J6, JSPS KAKENHI Grant Number (JP23K19967 and JP24K20847), and RIKEN Center for Advanced Intelligence Project.

## Impact Statement

This paper presents work whose goal is to advance the field of Machine Learning. There are many potential societal consequences of our work, none of which we feel must be specifically highlighted here.

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

*Table 2.* Comparison between existing noiseless algorithms' guarantees for simple regret and our result. In all algorithms except for GP-UCB+ and EXPLOIT+, the smoothness parameter of the Matérn kernel is assumed to be $\nu > 1/2$.

| Algorithm | Regret (SE) | Regret (Matérn) | Type | Remark |
|---|---|---|---|---|
| GP-EI (Bull, 2011) | N/A | $\widetilde{O}\left(T^{-\frac{\min\{1,\nu\}}{d}}\right)$ | D | |
| GP-EI with $\epsilon$-Greedy (Bull, 2011) | N/A | $\widetilde{O}\left(T^{-\frac{\nu}{d}}\right)$ | P | |
| GP-UCB (Lyu et al., 2019) (Kim & Sanz-Alonso, 2024) | $O\left(\sqrt{\frac{\ln^d T}{T}}\right)$ | $\widetilde{O}\left(T^{-\frac{\nu}{2\nu+d}}\right)$ | D | |
| Kernel-AMM-UCB (Flynn & Reeb, 2025) | $O\left(\frac{\ln^{d+1} T}{T}\right)$ | $\widetilde{O}\left(T^{-\frac{\nu d+2\nu^2}{2\nu^2+2\nu d+d^2}}\right)$ | D | |
| GP-UCB+, EXPLOIT+ (Kim & Sanz-Alonso, 2024) | $O\left(\exp\left(-CT^{\frac{1}{d}-\alpha}\right)\right)$ | $O\left(T^{-\frac{\nu}{d}+\alpha}\right)$ | P | $\alpha > 0$ is an arbitrarily fixed constant. $C > 0$ is some constant. |
| **MVR** **(our analysis)** | $O\left(\exp\left(-\frac{1}{2}T^{\frac{1}{d+1}}\ln^{-\alpha} T\right)\right)$ | $\widetilde{O}\left(T^{-\frac{\nu}{d}}\right)$ | D | $\alpha > 0$ is an arbitrarily fixed constant. |
| Lower Bound (Bull, 2011) | N/A | $\Omega\left(T^{-\frac{\nu}{d}}\right)$ | N/A | |

*Table 3.* Comparison of expected cumulative regret upper bound between existing noisy algorithms' guarantees and our result in the regime where the RKHS norm upper bound $B$ may change along with $T$. Here, $d$, $\ell$, $\nu$, and the noise level $\rho^2$ are supposed to be $\Theta(1)$. Note that the table below describes the expected regret by setting confidence level $\delta = 1/T$ and $\delta = 1/(TB)$ in existing PE and our PE, respectively. The resulting regrets for existing PE and our PE respectively suffer from additional $O(\sqrt{\ln T})$ and $O(\sqrt{\ln TB^2})$ factors in the high-probability regret upper bound of PE.

| Algorithm | Cumulative Regret (SE) | Cumulative Regret (Matérn) |
|---|---|---|
| GP-UCB (Srinivas et al., 2010) | $O\left(B\sqrt{T\ln^{d+1} T} + \sqrt{T}\ln^{d+1} T\right)$ | $\widetilde{O}\left(BT^{\frac{\nu+d}{2\nu+d}} + T^{\frac{2\nu+3d}{4\nu+2d}}\right)$ |
| Existing PE (Li & Scarlett, 2022) | $O\left(\max\{B,1\}\sqrt{T\ln^{d+4} T}\right)$ | $\widetilde{O}\left(\max\{B,1\}T^{\frac{\nu+d}{2\nu+d}}\right)$ |
| **PE** **(our analysis)** | $O\left(\sqrt{T\ln^{d+2}(TB^2)}(\ln T)\right)$ | $\widetilde{O}\left(B^{\frac{d}{2\nu+d}}T^{\frac{\nu+d}{2\nu+d}}\right)$ |
| Lower bound (Scarlett et al., 2017) | $\Omega\left(\sqrt{T\ln^{\frac{d}{2}}(TB^2)}\right)$ | $\Omega\left(B^{\frac{d}{2\nu+d}}T^{\frac{\nu+d}{2\nu+d}}\right)$ |

## A. Additional Table Summary

We describe the additional table to summarize the results of the existing analysis and ours. Table 2 summarizes the results of the simple regret upper bounds. Furthermore, Tables 3–5 and Tables 6–8 show the results for the setting of Section 5 and Section 6, respectively.

## B. Monotone Property of MIG

Fix $\mathbf{X} \subset \mathcal{X}^T$. Let $\boldsymbol{y}_1$ and $\boldsymbol{y}_2$ be $\boldsymbol{y}_1 = \boldsymbol{f}(\mathbf{X}) + \boldsymbol{\epsilon}_1$ and $\boldsymbol{y}_2 = \boldsymbol{f}(\mathbf{X}) + \boldsymbol{\epsilon}_1 + \boldsymbol{\epsilon}_2$, where $\boldsymbol{\epsilon}_1 \sim \mathcal{N}(\boldsymbol{0}, \boldsymbol{S}_1)$, $\boldsymbol{\epsilon}_2 \sim \mathcal{N}(\boldsymbol{0}, \boldsymbol{S}_2)$, and $\boldsymbol{S}_1$ and $\boldsymbol{S}_2$ are some positive semidefinite matrices. Then, from the definition, $\boldsymbol{f}(\mathbf{X})$ and $\boldsymbol{y}_2$ are conditionally independent given $\boldsymbol{y}_1$. Therefore, we can see that,

$$
\begin{aligned}
I(\boldsymbol{f}(\mathbf{X}); \boldsymbol{y}_1) &= I(\boldsymbol{f}(\mathbf{X}); \boldsymbol{y}_1) + I(\boldsymbol{f}(\mathbf{X}); \boldsymbol{y}_2 \mid \boldsymbol{y}_1) \\
&= I(\boldsymbol{f}(\mathbf{X}); \boldsymbol{y}_2) + I(\boldsymbol{f}(\mathbf{X}); \boldsymbol{y}_1 \mid \boldsymbol{y}_2) \\
&\geq I(\boldsymbol{f}(\mathbf{X}); \boldsymbol{y}_2),
\end{aligned}
$$

*Table 4.* Comparison between existing noisy algorithms' guarantees and our result in the regime where the RKHS norm upper bound $B$ may change along with $T$. Here, $d$, $\ell$, $\nu$, and the noise level $\rho^2$ are supposed to be $\Theta(1)$. As with Table 3, note that the resulting regret upper bounds for existing MVR and ours suffer from additional logarithmic factors in high-probability regret upper bound.

| Algorithm | Simple regret (SE) | Simple regret (Matérn) |
|---|---|---|
| GP-UCB (Srinivas et al., 2010) | $O\left(B\sqrt{\frac{\ln^{d+1} T}{T}} + \frac{\ln^{d+1} T}{\sqrt{T}}\right)$ | $\widetilde{O}\left(BT^{-\frac{\nu}{2\nu+d}} + T^{-\frac{2\nu-d}{4\nu+2d}}\right)$ |
| Existing MVR (Vakili et al., 2021a) | $O\left(\max\{B,1\}\sqrt{\frac{\ln^{d+2} T}{T}}\right)$ | $\widetilde{O}\left(\max\{B,1\}T^{-\frac{\nu}{2\nu+d}}\right)$ |
| **MVR** (**our analysis**) | $O\left(\sqrt{\frac{\ln^{d+2}(TB^2)}{T}}\right)$ | $\widetilde{O}\left(B^{\frac{d}{2\nu+d}}T^{-\frac{\nu}{2\nu+d}}\right)$ |

*Table 5.* Comparison of the total time step condition to find the expected $\epsilon$-optimal solution in the regime where the RKHS norm upper bound $B$ may change along with $T$. Here, $d$, $\ell$, $\nu$, and the noise level $\rho^2$ are supposed to be $\Theta(1)$.

| Algorithm | Time to simple regret $\epsilon$ (SE) | Time to simple regret $\epsilon$ (Matérn) |
|---|---|---|
| GP-UCB (Srinivas et al., 2010) | $O\left(\frac{B^2}{\epsilon^2}\ln^{d+1}\frac{B}{\epsilon} + \frac{1}{\epsilon^2}\ln^{2(d+1)}\frac{1}{\epsilon}\right)$ | $\widetilde{O}\left(\left(\frac{B}{\epsilon}\right)^{2+\frac{d}{\nu}} + \left(\frac{1}{\epsilon}\right)^{\frac{4\nu+2d}{d-2\nu}}\right)$ (if $2\nu > d$) |
| Existing MVR (Vakili et al., 2021a) | $O\left(\frac{\max\{B^2,1\}}{\epsilon^2}\ln^{d+2}\frac{\max\{B,1\}}{\epsilon}\right)$ | $\widetilde{O}\left(\left(\frac{\max\{B,1\}}{\epsilon}\right)^{2+\frac{d}{\nu}}\right)$ |
| **MVR** (**our analysis**) | $O\left(\frac{1}{\epsilon^2}\ln^{d+2}\frac{B}{\epsilon}\right)$ | $\widetilde{O}\left(\frac{1}{\epsilon^2}\left(\frac{B}{\epsilon}\right)^{\frac{d}{\nu}}\right)$ |
| Lower bound (Scarlett et al., 2017) | $\Omega\left(\frac{1}{\epsilon^2}\ln^{\frac{d}{2}}\frac{B}{\epsilon}\right)$ | $\Omega\left(\frac{1}{\epsilon^2}\left(\frac{B}{\epsilon}\right)^{\frac{d}{\nu}}\right)$ |

where $I(\cdot;\cdot)$ and $I(\cdot;\cdot\mid\cdot)$ are mutual information and conditional mutual information. The above inequality is called the data processing inequality (Theorem 2.8.1 of Cover & Thomas, 2006). Note that the first and second equalities and last inequality are obtained by $I(\boldsymbol{f}(\mathbf{X}); \boldsymbol{y}_2 \mid \boldsymbol{y}_1) = 0$ from the conditional independent property, the chain rule, and the non-negativity of the mutual information, respectively. Since this inequality holds for any $\mathbf{X} \subset \mathcal{X}^T$, by setting $\boldsymbol{S}_1 = \underline{\lambda}_T^2 \boldsymbol{I}_T$ and $\boldsymbol{S}_2 = \Sigma_T - \underline{\lambda}_T^2 \boldsymbol{I}_T$ we can obtain $\max_{\mathbf{X}\subset\mathcal{X}^T} I_{\underline{\lambda}_T^2 \boldsymbol{I}_T}(\boldsymbol{f}(\mathbf{X}); \boldsymbol{y}) \geq \max_{\mathbf{X}\subset\mathcal{X}^T} I_{\Sigma_T}(\boldsymbol{f}(\mathbf{X}); \boldsymbol{y})$.

## C. Proof of Section 3

### C.1. Proof of Lemma 3.1

*Proof.* Below, we give the proofs for stationary and non-stationary variance parameters separately.

*Table 6.* Summary of the cumulative regret upper bounds of the naive applications of existing results and our results in the non-stationary variance setting. Here, $d$, $\ell$, and $\nu$ are supposed to be $\Theta(1)$. In the table below, $\overline{\rho}_T^2 := \max_{t\in[T]}\rho_t^2$ denotes the maximum variance proxy up to step $T$. Note that the table below describes the expected regret by setting the confidence level $\delta = 1/T$ in PE and VA-PE, respectively. The resulting regrets suffer from additional $O(\sqrt{\ln T})$ factors in the high-probability regret upper bound of these algorithms.

| Algorithm | Cumulative Regret (SE) | Cumulative Regret (Matérn) | |
| --- | --- | --- | --- |
| | | $d \leq 2\nu$ or $V_T = \Omega\left(T^{\frac{d-2\nu}{d}}\right)$ | $d < 2\nu$ and $V_T = O\left(T^{\frac{d-2\nu}{d}}\right)$ |
| GP-UCB (Srinivas et al., 2010) | $O\left(\sqrt{\frac{T}{\ln(1+\overline{\rho}_T^{-2})}}\ln^{d+1}\frac{T}{\overline{\rho}_T^2}\right)$ | $\widetilde{O}\left(\sqrt{\frac{1}{\ln(1+\overline{\rho}_T^{-2})}}\overline{\rho}_T^{-\frac{2d}{2\nu+d}}T^{\frac{2\nu+3d}{4\nu+2d}}\right)$ | |
| PE (Li & Scarlett, 2022) | $O\left((\ln T)^{3/2}\sqrt{\frac{T}{\ln(1+\overline{\rho}_T^{-2})}}\ln^{d+1}\frac{T}{\overline{\rho}_T^2}\right)$ | $\widetilde{O}\left(\sqrt{\frac{1}{\ln(1+\overline{\rho}_T^{-2})}}\overline{\rho}_T^{-\frac{d}{2\nu+d}}T^{\frac{\nu+d}{2\nu+d}}\right)$ | |
| **VA-GP-UCB** **(ours)** | $O\left(\sqrt{V_T}\ln^{d+1}T\right)$ | $\widetilde{O}\left(T^{\frac{2d}{2\nu+d}}\sqrt{V_T}\right)$ | |
| **VA-PE** **(ours)** | $O\left((\ln T)^{3/2}\sqrt{V_T\left(\ln^{d+1}\frac{T^2}{V_T}\right)}\right)$ | $\widetilde{O}\left(V_T^{\frac{\nu}{2\nu+d}}T^{\frac{d}{2\nu+d}}\right)$ | $\widetilde{O}\left(T^{\frac{d-\nu}{d}}\right)$ |
| Lower bound (Corollary 6.1) | $\Omega\left(\sqrt{V_T\ln^{\frac{d}{2}}\frac{T^2}{V_T}}\right)$ | $\Omega\left(V_T^{\frac{\nu}{2\nu+d}}T^{\frac{d}{2\nu+d}}\right)$ | |

*Table 7.* Summary of the simple regret upper bounds of the naive applications of existing results and our results in non-stationary variance setting. Here, $d$, $\ell$, and $\nu$ are supposed to be $\Theta(1)$. In the table below, $\overline{\rho}_T^2 := \max_{t\in[T]}\rho_t^2$ denotes the maximum variance proxy up to step $T$. As with Table 6, note that the resulting regrets for MVR and VA-MVR suffer from additional logarithmic factors in high-probability regret upper bound.

| Algorithm | Simple Regret (SE) | Simple Regret (Matérn) | |
| --- | --- | --- | --- |
| | | $d \leq 2\nu$ or $V_T = \Omega\left(T^{\frac{d-2\nu}{d}}\right)$ | $d > 2\nu$ and $V_T = O\left(T^{\frac{d-2\nu}{d}}\right)$ |
| GP-UCB (Srinivas et al., 2010) | $O\left(\sqrt{\frac{1}{T\ln(1+\overline{\rho}_T^{-2})}}\ln^{d+1}\frac{T}{\overline{\rho}_T^2}\right)$ | $\widetilde{O}\left(\sqrt{\frac{1}{\ln(1+\overline{\rho}_T^{-2})}}\overline{\rho}_T^{-\frac{2d}{2\nu+d}}T^{-\frac{2\nu-d}{4\nu+2d}}\right)$ | |
| MVR (Vakili et al., 2021a) | $O\left(\sqrt{\frac{\ln T}{T\ln(1+\overline{\rho}_T^{-2})}}\ln^{d+1}\frac{T}{\overline{\rho}_T^2}\right)$ | $\widetilde{O}\left(\sqrt{\frac{1}{\ln(1+\overline{\rho}_T^{-2})}}\overline{\rho}_T^{-\frac{d}{2\nu+d}}T^{-\frac{\nu}{2\nu+d}}\right)$ | |
| **VA-GP-UCB** **(ours)** | $O\left(\frac{\sqrt{V_T}}{T}\ln^{d+1}T\right)$ | $\widetilde{O}\left(T^{-\frac{2\nu-d}{2\nu+d}}\sqrt{V_T}\right)$ | |
| **VA-MVR** **(ours)** | $O\left(\sqrt{\frac{V_T}{T^2}\left(\ln^{d+1}\frac{T^2}{V_T}\right)}(\ln T)\right)$ | $\widetilde{O}\left(V_T^{\frac{\nu}{2\nu+d}}T^{-\frac{2\nu}{2\nu+d}}\right)$ | $\widetilde{O}\left(T^{-\frac{\nu}{d}}\right)$ |

**Stationary variance case.** Fix any $T \in \{T \in \mathbb{N}_+ \mid T/2 \geq 3\gamma_T(\widetilde{\lambda}_T^2 \boldsymbol{I}_T)\}$ and define $\mathcal{T} = \left\{t \in [T] \mid \widetilde{\lambda}_T^{-1}\sigma_{\widetilde{\lambda}_T^2\boldsymbol{I}_{t-1}}(\boldsymbol{x}_{T,t};\mathbf{X}_{T,t-1}) \leq 1\right\}$. Then, we have the following if $\mathcal{T} \neq \emptyset$:

$$\max_{\boldsymbol{x}\in\widetilde{\mathcal{X}}}\sigma_{\overline{\lambda}_T^2\boldsymbol{I}_T}(\boldsymbol{x};\mathbf{X}_{T,T}) \leq \frac{1}{|\mathcal{T}|}\sum_{t\in\mathcal{T}}\sigma_{\overline{\lambda}_T^2\boldsymbol{I}_{t-1}}(\boldsymbol{x}_{T,t};\mathbf{X}_{T,t-1}) \tag{11}$$

$$\leq \frac{1}{|\mathcal{T}|}\sum_{t\in\mathcal{T}}\sigma_{\widetilde{\lambda}_T^2\boldsymbol{I}_{t-1}}(\boldsymbol{x}_{T,t};\mathbf{X}_{T,t-1}) \tag{12}$$

$$= \frac{1}{|\mathcal{T}|}\sum_{t\in\mathcal{T}}\widetilde{\lambda}_T\min\left\{1,\widetilde{\lambda}_T^{-1}\sigma_{\widetilde{\lambda}_T^2\boldsymbol{I}_{t-1}}(\boldsymbol{x}_{T,t};\mathbf{X}_{T,t-1})\right\} \tag{13}$$

$$\leq \frac{1}{|\mathcal{T}|}\sqrt{\widetilde{\lambda}_T^2|\mathcal{T}|\sum_{t\in\mathcal{T}}\min\left\{1,\widetilde{\lambda}_T^{-2}\sigma_{\widetilde{\lambda}_T^2\boldsymbol{I}_{t-1}}^2(\boldsymbol{x}_{T,t};\mathbf{X}_{T,t-1})\right\}} \tag{14}$$

$$\leq \frac{2}{|\mathcal{T}|}\sqrt{\widetilde{\lambda}_T^2 T\sum_{t\in[T]}\frac{1}{2}\ln\left(1+\widetilde{\lambda}_T^{-2}\sigma_{\widetilde{\lambda}_T^2\boldsymbol{I}_{t-1}}^2(\boldsymbol{x}_{T,t};\mathbf{X}_{T,t-1})\right)} \tag{15}$$

$$\leq \frac{2}{|\mathcal{T}|}\sqrt{\widetilde{\lambda}_T^2 T\gamma_T(\widetilde{\lambda}_T^2\boldsymbol{I}_T)}, \tag{16}$$

*Table 8.* Summary of the total time step condition to find $\epsilon$-optimal solution in non-stationary variance setting. We only focus on our algorithms and a lower bound here for simplicity.

| Algorithm | Time to Simple Regret $\epsilon$ (SE) | Time to Simple Regret $\epsilon$ (Matérn) | |
|---|---|---|---|
| | | $d \leq 2\nu$ or $V_T = \Omega\left(T^{\frac{d-2\nu}{d}}\right)$ | $d > 2\nu$ and $V_T = O\left(T^{\frac{d-2\nu}{d}}\right)$ |
| **VA-GP-UCB (ours)** | $O\left(\sqrt{\frac{V_T}{\epsilon^2} \ln^{d+1} \frac{V_T}{\epsilon^2}}\right)$ | $\widetilde{O}\left(\left(\frac{V_T}{\epsilon^2}\right)^{\frac{2\nu+d}{2\nu-d}}\right)$ (if $2\nu > d$) | |
| **VA-MVR (ours)** | $O\left(\sqrt{\frac{V_T}{\epsilon^2}\left(\ln^{d+1}\frac{1}{\epsilon}\right)\left(\ln\frac{V_T}{\epsilon^2}\right)}\right)$ | $\widetilde{O}\left(\left(\frac{V_T}{\epsilon^2}\right)^{\frac{1}{2}}\left(\frac{1}{\epsilon}\right)^{\frac{d}{2\nu}}\right)$ | $\widetilde{O}\left(\left(\frac{1}{\epsilon}\right)^{\frac{d}{\nu}}\right)$ |
| Lower bound (Corollary 6.2) | $\Omega\left(\sqrt{\frac{V_T}{\epsilon^2} \ln^{\frac{d}{2}} \frac{1}{\epsilon}}\right)$ | $\Omega\left(\left(\frac{V_T}{\epsilon^2}\right)^{\frac{1}{2}}\left(\frac{1}{\epsilon}\right)^{\frac{d}{2\nu}}\right)$ | $\Omega\left(\left(\frac{1}{\epsilon}\right)^{\frac{d}{\nu}}\right)$ |

where:

- Eq. (11) follows from the fact that $\sigma_{\widetilde{\lambda}_T^2 \boldsymbol{I}_T}(\boldsymbol{x}; \mathbf{X}_{T,T}) \leq \sigma_{\widetilde{\lambda}_T^2 \boldsymbol{I}_{t-1}}(\boldsymbol{x}_{T,t}; \mathbf{X}_{T,t-1})$ holds for all $t \in [T]$, $\boldsymbol{x} \in \widetilde{\mathcal{X}}$ from the MVR selection rule.

- Eq. (12) follows from the monotonicity of the posterior variance on the noise parameter. Note that $\overline{\lambda}_T \leq \widetilde{\lambda}_T$ holds from the assumption.

- Eq. (13) follows from the definition of $\mathcal{T}$.

- Eq. (14) follows from Schwartz's inequality.

- Eq. (15) follows from $|\mathcal{T}| \leq T$ and the inequality $\forall a \geq 0, \min\{1, a\} \leq 2\ln(1+a)$.

- Eq. (16) follows from $\sum_{t \in [T]} \frac{1}{2} \ln\left(1 + \widetilde{\lambda}_T^{-2} \sigma_{\widetilde{\lambda}_T^2 \boldsymbol{I}_{t-1}}^2(\boldsymbol{x}_{T,t}; \mathbf{X}_{T,t-1})\right) = I_{\widetilde{\lambda}_T^2 \boldsymbol{I}_T}(\boldsymbol{f}(\mathbf{X}_{T,T}), \boldsymbol{y}) \leq \gamma_T(\widetilde{\lambda}_T^2 \boldsymbol{I}_T)$. See, e.g., Theorem 5.3 in (Srinivas et al., 2010).

Furthermore, from the condition of $T$, we have

$$|\mathcal{T}| = T - |\mathcal{T}^c| \tag{17}$$

$$\geq T - 3\gamma_T(\widetilde{\lambda}_T^2 \boldsymbol{I}_T) \tag{18}$$

$$\geq T/2, \tag{19}$$

where the first inequality follows from Lemma 3.3. Combining Eq. (19) with Eq. (16), we obtain the desired result.

**Non-stationary variance setting.** We start by extending the elliptical potential count lemma to the non-stationary setting.

**Lemma C.1** (Elliptical potential count lemma for non-stationary variance setting). *Fix any $T \in \mathbb{N}_+$, any sequence $\boldsymbol{x}_1, \ldots, \boldsymbol{x}_T \in \mathcal{X}$, and $\lambda_1, \ldots, \lambda_T > 0$. Define $\mathcal{T}^c$ as $\mathcal{T}^c = \{t \in [T] \mid \lambda_t^{-1}\sigma_{\Sigma_{t-1}}(\boldsymbol{x}_t; \mathbf{X}_{t-1}) > 1\}$, where $\mathbf{X}_{t-1} = (\boldsymbol{x}_1, \ldots, \boldsymbol{x}_{t-1})$ and $\Sigma_{t-1} = \mathrm{diag}(\lambda_1^2, \ldots, \lambda_{t-1}^2)$. Then, the number of elements of $\mathcal{T}^c$ satisfies $|\mathcal{T}^c| \leq \min\{4\overline{\gamma}\left(4\gamma_T(\Sigma_T), \underline{\lambda}_T^2\right), 4\gamma_T(\Sigma_T)\}$, where $\underline{\lambda}_T^2 = \min_{t \in [T]} \lambda_t^2$. Furthermore, $\overline{\gamma}(\cdot, \cdot)$ is any monotonic upper bound of MIG defined on $\mathbb{R}_+ \times \mathbb{R}_+$, which satisfies $\forall T \in \mathbb{N}_+, \lambda > 0, \gamma_T(\lambda^2 \boldsymbol{I}_T) \leq \overline{\gamma}(T, \lambda^2)$ and $\forall \lambda > 0, T \geq 1, \epsilon \geq 0, \overline{\gamma}(T, \lambda^2) \leq \overline{\gamma}(T + \epsilon, \lambda^2)$.*

*Proof of Lemma C.1.* If $\mathcal{T}^c = \emptyset$, the claimed inequality is trivial, so we focus on the case where $\mathcal{T}^c \neq \emptyset$ hereafter. From

the definition of $\mathcal{T}^c$, we have

$$|\mathcal{T}^c| = \sum_{t \in \mathcal{T}^c} \min\left\{1, \lambda_t^{-1} \sigma_{\Sigma_{t-1}}(\boldsymbol{x}_t; \mathbf{X}_{t-1})\right\} \tag{20}$$

$$\leq \sum_{t \in \mathcal{T}^c} \min\left\{1, \lambda_t^{-2} \sigma_{\Sigma_{t-1}}^2(\boldsymbol{x}_t; \mathbf{X}_{t-1})\right\} \tag{21}$$

$$\leq 4 \sum_{t \in \mathcal{T}^c} \frac{1}{2} \ln\left(1 + \lambda_t^{-2} \sigma_{\Sigma_{t-1}}^2(\boldsymbol{x}_t; \mathbf{X}_{t-1})\right) \tag{22}$$

$$\leq 4 \sum_{t \in [T]} \frac{1}{2} \ln\left(1 + \lambda_t^{-2} \sigma_{\Sigma_{t-1}}^2(\boldsymbol{x}_t; \mathbf{X}_{t-1})\right) \tag{23}$$

$$\leq 4\gamma_T(\Sigma_T). \tag{24}$$

In the above inequalities:

- Eqs. (20) and (21) follows from $1 = \min\left\{1, \lambda_t^{-1} \sigma_{\Sigma_{t-1}}(\boldsymbol{x}_t; \mathbf{X}_{t-1})\right\} \leq \min\left\{1, \lambda_t^{-2} \sigma_{\Sigma_{t-1}}^2(\boldsymbol{x}_t; \mathbf{X}_{t-1})\right\}$, which holds for all $t \in \mathcal{T}^c$ from the definition of $\mathcal{T}^c$.

- Eq. (22) follows from the inequality $\forall a \geq 0, \min\{1, a\} \leq 2\ln(1 + a)$.

- Eq. (24) follows from $\sum_{t \in [T]} \frac{1}{2} \ln\left(1 + \lambda_t^{-2} \sigma_{\Sigma_{t-1}}^2(\boldsymbol{x}_t; \mathbf{X}_{t-1})\right) = I_{\Sigma_T}(\boldsymbol{f}(\mathbf{X}_T), \boldsymbol{y}) \leq \gamma_T(\Sigma_T)$. This is a direct extension of Theorem 5.3 in (Srinivas et al., 2010) and is proved explicitly in the proof of Proposition 1 in (Makarova et al., 2021).

Here, we set $t_1, \ldots, t_{|\mathcal{T}^c|} \in [T]$ as the elements of $\mathcal{T}^c$, which are indexed in the increasing order. Furthermore, for all $i \in [|\mathcal{T}^c|]$, let us respectively define $\widetilde{\boldsymbol{x}}_i, \widetilde{\mathbf{X}}_i, \widetilde{\lambda}_t$, and $\widetilde{\Sigma}_i$ as $\widetilde{\boldsymbol{x}}_i = \boldsymbol{x}_{t_i}, \widetilde{\mathbf{X}}_i = (\boldsymbol{x}_{t_1}, \ldots, \boldsymbol{x}_{t_i}), \widetilde{\lambda}_i = \lambda_{t_i}$, and $\widetilde{\Sigma}_i = \mathrm{diag}(\lambda_{t_1}^2, \ldots, \lambda_{t_i}^2)$. Then, from Eq. (22), we also have the following inequality:

$$|\mathcal{T}^c| \leq 4 \sum_{t \in \mathcal{T}^c} \frac{1}{2} \ln\left(1 + \lambda_t^{-2} \sigma_{\Sigma_{t-1}}^2(\boldsymbol{x}_t; \mathbf{X}_{t-1})\right) \tag{25}$$

$$\leq 4 \sum_{t=1}^{|\mathcal{T}^c|} \frac{1}{2} \ln\left(1 + \widetilde{\lambda}_t^{-2} \sigma_{\widetilde{\Sigma}_{t-1}}^2(\widetilde{\boldsymbol{x}}_t; \widetilde{\mathbf{X}}_{t-1})\right) \tag{26}$$

$$\leq 4\gamma_{|\mathcal{T}^c|}\left(\widetilde{\Sigma}_{|\mathcal{T}^c|}\right) \tag{27}$$

$$\leq 4\overline{\gamma}\left(|\mathcal{T}^c|, \min_{t \in [|\mathcal{T}^c|]} \widetilde{\lambda}_t^2\right) \tag{28}$$

$$\leq 4\overline{\gamma}\left(4\gamma_T(\Sigma_T), \underline{\lambda}_T^2\right). \tag{29}$$

The second inequality follows from the fact that the posterior variance has the monotonicity on data; namely, $\sigma_{\Sigma_{t_i-1}}^2(\boldsymbol{x}_{t_i}; \mathbf{X}_{t_i-1}) \leq \sigma_{\widetilde{\Sigma}_{i-1}}^2(\widetilde{\boldsymbol{x}}_i; \widetilde{\mathbf{X}}_{i-1})$ holds since the input data $\widetilde{\mathbf{X}}_{i-1}$ is included in $\mathbf{X}_{t_i-1}$. $\qquad \square$

The remaining proof is given by following the proof strategy under the stationary variance setting using Lemma C.1. Here, fix any $T \in \{T \in \mathbb{N}_+ \mid T/2 \geq 4\gamma_T(\widetilde{\Sigma}_T)\}$ and define $\mathcal{T} = \left\{t \in [T] \mid \widetilde{\lambda}_t^{-1} \sigma_{\widetilde{\Sigma}_{t-1}}(\boldsymbol{x}_t; \mathbf{X}_{t-1}) \leq 1\right\}$. Then, as with the proof

in the stationary variance setting, we have:

$$\max_{\boldsymbol{x} \in \widetilde{\mathcal{X}}} \sigma_{\Sigma_T}(\boldsymbol{x}; \mathbf{X}_T) \leq \frac{1}{|\mathcal{T}|} \sum_{t \in \mathcal{T}} \sigma_{\Sigma_{t-1}}(\boldsymbol{x}_t; \mathbf{X}_{t-1}) \tag{30}$$

$$\leq \frac{1}{|\mathcal{T}|} \sum_{t \in \mathcal{T}} \sigma_{\widetilde{\Sigma}_{t-1}}(\boldsymbol{x}_t; \mathbf{X}_{t-1}) \tag{31}$$

$$= \frac{1}{|\mathcal{T}|} \sum_{t \in \mathcal{T}} \widetilde{\lambda}_t \min \left\{ 1, \widetilde{\lambda}_t^{-1} \sigma_{\widetilde{\Sigma}_{t-1}}(\boldsymbol{x}_t; \mathbf{X}_{t-1}) \right\} \tag{32}$$

$$\leq \frac{1}{|\mathcal{T}|} \sqrt{\left( \sum_{t \in \mathcal{T}} \widetilde{\lambda}_t^2 \right) \sum_{t \in \mathcal{T}} \min \left\{ 1, \widetilde{\lambda}_t^{-2} \sigma_{\widetilde{\Sigma}_{t-1}}^2(\boldsymbol{x}_t; \mathbf{X}_{t-1}) \right\}} \tag{33}$$

$$\leq \frac{2}{|\mathcal{T}|} \sqrt{\left( \sum_{t \in [T]} \widetilde{\lambda}_t^2 \right) \sum_{t \in \mathcal{T}} \frac{1}{2} \ln \left( 1 + \widetilde{\lambda}_t^{-2} \sigma_{\widetilde{\Sigma}_{t-1}}^2(\boldsymbol{x}_t; \mathbf{X}_{t-1}) \right)} \tag{34}$$

$$\leq \frac{2}{|\mathcal{T}|} \sqrt{\left( \sum_{t \in [T]} \widetilde{\lambda}_t^2 \right) \sum_{t \in [T]} \frac{1}{2} \ln \left( 1 + \widetilde{\lambda}_t^{-2} \sigma_{\widetilde{\Sigma}_{t-1}}^2(\boldsymbol{x}_t; \mathbf{X}_{t-1}) \right)} \tag{35}$$

$$\leq \frac{2}{|\mathcal{T}|} \sqrt{\left( \sum_{t \in [T]} \widetilde{\lambda}_t^2 \right) \gamma_T(\widetilde{\Sigma}_T)}. \tag{36}$$

Furthermore, from Lemma C.1 and the condition of $T$, we have

$$|\mathcal{T}| = T - |\mathcal{T}^c| \tag{37}$$

$$\geq T - 4\gamma_T(\widetilde{\Sigma}_T) \tag{38}$$

$$\geq T/2. \tag{39}$$

Combining the above inequality with Eq. (36), we obtain the desired result. □

### C.2. Proof of Corollary 3.2

*Proof.* We describe the proof for each statement separately.

**Statement 1.** From the assumption, $\forall T \in \mathbb{N}_+, \overline{\lambda}_T^2 \geq C \exp \left( -T^{\frac{1}{d+1}} \ln^{-\alpha}(1+T) \right)$ holds for some constant $C > 0$. Furthermore, since $k = k_{\mathrm{SE}}$, there exist constant $\overline{C} > 0$ such that $\gamma_T(\overline{\lambda}_T^2 \boldsymbol{I}_T) \leq \overline{C} \ln^{d+1}(T/\overline{\lambda}_T^2)$. Here, $\overline{C}$ depends on $\ell$ and $d$. Then,

$$\gamma_T(\overline{\lambda}_T^2 \boldsymbol{I}_T) \leq \overline{C} \ln^{d+1} \left( TC^{-1} \exp \left( T^{\frac{1}{d+1}} \ln^{-\alpha}(1+T) \right) \right) \tag{40}$$

$$\leq \overline{C} \ln^{d+1} \left( \exp \left( \widetilde{C} T^{\frac{1}{d+1}} \ln^{-\alpha}(1+T) \right) \right) \tag{41}$$

$$= T \overline{C} \widetilde{C}^{d+1} \left( \ln(1+T) \right)^{-\alpha(d+1)}, \tag{42}$$

where $\widetilde{C} > 0$ is a constant that satisfies $TC^{-1} \exp \left( T^{\frac{1}{d+1}} \ln^{-\alpha}(1+T) \right) \leq \exp \left( \widetilde{C} T^{\frac{1}{d+1}} \ln^{-\alpha}(1+T) \right)$ for all $T \in \mathbb{N}_+$. Here, note that $\widetilde{C}$ may depend on $d$, $\alpha$, and $C$. Since $\overline{C} \widetilde{C}^{d+1} \left( \ln(1+T) \right)^{-\alpha(d+1)} \to 0$ as $T \to 0$, there exists a constant $\overline{T}$ such that $T/2 \geq 3T\overline{C} \widetilde{C}^{d+1} \left( \ln(1+T) \right)^{-\alpha(d+1)}$ holds for all $T \geq \overline{T}$. Furthermore, such constant $\overline{T}$ satisfies $\forall T \geq \overline{T}, T/2 \geq 3\gamma_T(\overline{\lambda}_T^2 \boldsymbol{I}_T)$ from Eq. (42). Therefore, for any $T \geq \overline{T}$, Eq. (7) holds from Lemma 3.1 with $\widetilde{\lambda}_T^2 = \overline{\lambda}_T^2$.

**Statement 2.** From the assumption, $\forall T \in \mathbb{N}_+, \overline{\lambda}_T^2 \geq C T^{-\frac{2\nu}{d}} \ln^{\frac{2\nu(1+\alpha)}{d}}(1+T)$ holds for some constant $C > 0$. Furthermore, since $k = k_{\mathrm{Matérn}}$ with $\nu > 1/2$, there exist constant $\overline{C} > 0$ such that $\gamma_T(\overline{\lambda}_T^2 \boldsymbol{I}_T) \leq \overline{C}(T/\overline{\lambda}_T^2)^{\frac{d}{2\nu+d}} \ln^{\frac{2\nu}{2\nu+d}}(1+$

$T/\overline{\lambda}_T^2$). Here, $\overline{C}$ depends on $\ell$, $\nu$, and $d$. Then,

$$\gamma_T(\overline{\lambda}_T^2 \boldsymbol{I}_T) \leq \overline{C} C^{-\frac{d}{2\nu+d}} T \left( \ln^{-\frac{2\nu}{2\nu+d}(1+\alpha)}(1+T) \right) \ln^{\frac{2\nu}{2\nu+d}} \left( 1 + C^{-1} T^{\frac{2\nu+d}{d}} \ln^{-\frac{2\nu(1+\alpha)}{d}}(1+T) \right) \tag{43}$$

$$\leq \overline{C} C^{-\frac{d}{2\nu+d}} T \left( \ln^{-\frac{2\nu}{2\nu+d}(1+\alpha)}(1+T) \right) \ln^{\frac{2\nu}{2\nu+d}} \left( \widehat{C}(1+T)^{\frac{2\nu+d}{d}} \right) \tag{44}$$

$$\leq \overline{C} C^{-\frac{d}{2\nu+d}} \widetilde{C} T \ln^{-\frac{2\nu\alpha}{2\nu+d}}(1+T), \tag{45}$$

where $\widehat{C} > 0$ is a constant that satisfies $1 + C^{-1} T^{\frac{2\nu+d}{d}} \ln^{-\frac{2\nu(1+\alpha)}{d}}(1+T) \leq \widehat{C}(1+T)^{\frac{2\nu+d}{d}}$ for all $T \in \mathbb{N}_+$. Furthermore, $\widetilde{C} > 0$ is a constant that satisfies $\ln^{\frac{2\nu}{2\nu+d}}\left(\widehat{C}(1+T)^{\frac{2\nu+d}{d}}\right) \leq \widetilde{C} \ln^{\frac{2\nu}{2\nu+d}}(1+T)$ for all $T \in \mathbb{N}_+$. Here, note that $\widehat{C}$ and $\widetilde{C}$ may depend on $d$, $\alpha$, and $\nu$, but are the constants under the condition that $d$, $\alpha$, and $\nu$ are fixed. Since $\overline{C} C^{-\frac{d}{2\nu+d}} \widetilde{C} \ln^{-\frac{2\nu\alpha}{2\nu+d}}(1+T) \to 0$ as $T \to 0$, there exists a constant $\overline{T}$ such that $T/2 \geq 3\overline{C} C^{-\frac{d}{2\nu+d}} \widetilde{C} T \ln^{-\frac{2\nu\alpha}{2\nu+d}}(1+T)$ holds for all $T \geq \overline{T}$. Furthermore, such constant $\overline{T}$ satisfies $\forall T \geq \overline{T}, T/2 \geq 3\gamma_T(\overline{\lambda}_T^2 \boldsymbol{I}_T)$ from Eq. (45). Therefore, for any $T \geq \overline{T}$, Eq. (7) holds from Lemma 3.1 with $\widetilde{\lambda}_T^2 = \overline{\lambda}_T^2$.

**Statements 3 and 4.** For statement 3, we set $\widetilde{\lambda}_T^2 = C \exp\left(-T^{\frac{1}{d+1}} \ln^{-\alpha}(1+T)\right)$. Then, following the same argument as the proof of statement 1 in Corollary 3.2, we confirm that $\gamma_T(\widetilde{\lambda}_T^2 \boldsymbol{I}_T) = o(T)$, and the quantity $\min\{T \in \mathbb{N}_+ \mid \forall t \geq T, t/2 \geq 3\gamma_t(\widetilde{\lambda}_t^2 \boldsymbol{I}_t)\}$ is bounded from above by some finite constant $\overline{T}$ that depends on $d$, $\alpha$, and $C$. Furthermore, from $\gamma_T(\widetilde{\lambda}_T^2 \boldsymbol{I}_T) = o(T)$ and the definition of $\widetilde{\lambda}_T$, we can evaluate the order of the r.h.s. in Eq. (4) as

$$\frac{4}{T}\sqrt{\widetilde{\lambda}_T^2 T \gamma_T(\widetilde{\lambda}_T^2 \boldsymbol{I}_T)} \leq O\left(\sqrt{\exp\left(-T^{\frac{1}{d+1}}\ln^{-\alpha} T\right)}\right). \tag{46}$$

We also obtain the result for statement 4 by setting $\widetilde{\lambda}_T^2$ as $\widetilde{\lambda}_T^2 = C T^{-\frac{2\nu}{d}}(\ln T)^{\frac{2\nu(1+\alpha)}{d}}$ and calculating the explicit value of r.h.s. in Eq. (4). $\qquad \square$

## D. Proof in Section 4

### D.1. Proof of Theorem 4.1

**Lemma D.1** (Deterministic confidence bound for noiseless setting, Lemma 11 in (Lyu et al., 2019) or Proposition 1 in (Vakili et al., 2021a)). *Suppose Assumptions 2.1, 2.2 with $\forall t \in \mathbb{N}_+, \rho_t = 0$ hold. Then, for any sequence[3] $(\boldsymbol{x}_t)_{t \in \mathbb{N}_+}$ on $\mathcal{X}$, the following event holds:*

$$\forall t \in \mathbb{N}_+, \ \forall \boldsymbol{x} \in \mathcal{X}, \ |f(\boldsymbol{x}) - \mu_{\lambda^2 \boldsymbol{I}_t}(\boldsymbol{x}; \mathbf{X}_t, \boldsymbol{f}_t)| \leq B\sigma_{\lambda^2 \boldsymbol{I}_t}(\boldsymbol{x}; \mathbf{X}_t), \tag{47}$$

*where $\lambda = 0$, $\mathbf{X}_t = (\boldsymbol{x}_1, \ldots, \boldsymbol{x}_t)$, and $\boldsymbol{f}_t = (f(\boldsymbol{x}_1), \ldots, f(\boldsymbol{x}_t))^\top$.*

The proof of Theorem 4.1 follows by using Lemma D.1 and Lemma 3.2 in the standard PE analysis.

*Proof of Theorem 4.1.* Fix any $\alpha > 0$ and set $\overline{T}$ as the constant defined in statements 3 or 4 of Corollary 3.2. Furthermore, let $\overline{i} \in \mathbb{N}_+$ be the first batch index such that $N_{\overline{i}} \geq \overline{T}$ holds. In both kernels, the cumulative regret before the start of batch $\overline{i}+1$ is bounded from above by $\max\{8B\overline{T}, 2BN_1\}$ due to $\|f\|_\infty \leq \|f\|_{\mathcal{H}_k} \leq B$, $N_{\overline{i}} < 2\overline{T}$, and $\sum_{i=1}^{\overline{i}-1} N_i \leq N_{\overline{i}}$ if $N_1 < \overline{T}$.

---

[3] Strictly speaking, the input $(\boldsymbol{x}_t)_{t \in \mathbb{N}_+}$ is required to have no duplication to guarantee the existence of the inverse gram matrix. In all of our algorithms, such events only occur when the algorithm finds the maximizer $\boldsymbol{x}^*$, which leads to subsequent instantaneous regrets of 0, and our upper bounds trivially hold. Therefore, we suppose that such events do not occur in our proof.

Next, for any $i$-th batch with $i \geq \bar{i} + 1$, we have

$$\sum_{j=1}^{N_i} f(\boldsymbol{x}^*) - f(\boldsymbol{x}_j^{(i)}) \leq \sum_{j=1}^{N_i} \mathrm{ucb}_{i-1}(\boldsymbol{x}^*) - \mathrm{lcb}_{i-1}(\boldsymbol{x}_j^{(i)}) \tag{48}$$

$$= \sum_{j=1}^{N_i} \mathrm{lcb}_{i-1}(\boldsymbol{x}^*) - \mathrm{ucb}_{i-1}(\boldsymbol{x}_j^{(i)}) + 2\beta^{1/2} \sigma_{\lambda^2 \boldsymbol{I}_{N_{i-1}}}(\boldsymbol{x}_j^{(i)}; \mathbf{X}_{N_{i-1}}^{(i-1)}) + 2\beta^{1/2} \sigma_{\lambda^2 \boldsymbol{I}_{N_{i-1}}}(\boldsymbol{x}^*; \mathbf{X}_{N_{i-1}}^{(i-1)}) \tag{49}$$

$$\leq \sum_{j=1}^{N_i} \mathrm{lcb}_{i-1}(\boldsymbol{x}^*) - \max_{\boldsymbol{x} \in \mathcal{X}_{i-1}} \mathrm{lcb}_{i-1}(\boldsymbol{x}) + 4B \max_{\boldsymbol{x} \in \mathcal{X}_{i-1}} \sigma_{\lambda^2 \boldsymbol{I}_{N_{i-1}}}(\boldsymbol{x}; \mathbf{X}_{N_{i-1}}^{(i-1)}) \tag{50}$$

$$\leq 4BN_i \max_{\boldsymbol{x} \in \mathcal{X}_{i-1}} \sigma_{\lambda^2 \boldsymbol{I}_{N_{i-1}}}(\boldsymbol{x}; \mathbf{X}_{N_{i-1}}^{(i-1)}), \tag{51}$$

where Eq. (48) follows from the definition of $\beta^{1/2}$ and Lemma D.1. Furthermore, Eq. (50) follows from $\boldsymbol{x}_j^{(i)}, \boldsymbol{x}^* \in \mathcal{X}_{i-1}$.

**For SE kernel.** From statement 3 in Corollary 3.2[4], we have

$$4BN_i \max_{\boldsymbol{x} \in \mathcal{X}_{i-1}} \sigma_{\lambda^2 \boldsymbol{I}_{N_{i-1}}}(\boldsymbol{x}; \mathbf{X}_{N_{i-1}}^{(i)}) \leq 8BN_{i-1} \sqrt{C_1 \exp\left(-N_{i-1}^{\frac{1}{d+1}} \ln^{-\alpha} N_{i-1}\right)} \tag{52}$$

$$\leq 8BC_2. \tag{53}$$

In the above inequalities, $C_1, C_2 \in (0, \infty)$ are the constant that may depend on $d$, $\ell$, and $\alpha$. The existence of $C_2$ is guaranteed by $t\sqrt{\exp\left(-t^{\frac{1}{d+1}} \ln^{-\alpha} t\right)} \to 0$. Since the total number of batches is bounded from above by $1 + \log_2 T$, we have

$$R_T \leq \max\{8B\overline{T}, 2BN_1\} + 8BC_2 \log_2 T = O(\ln T). \tag{54}$$

**For Matérn kernel.** From statement 4 in Corollary 3.2, we have

$$4BN_i \max_{\boldsymbol{x} \in \mathcal{X}_{i-1}} \sigma_{\lambda^2 \boldsymbol{I}_{N_{i-1}}}(\boldsymbol{x}; \mathbf{X}_{N_{i-1}}^{(i)}) \leq 8BC_1 N_{i-1}^{\frac{d-\nu}{d}} \ln^{\frac{\nu}{d}(1+\alpha)} N_{i-1}, \tag{55}$$

with some constant $C_1 \in (0, \infty)$ that depends on $d$, $\nu$, $\ell$, and $\alpha$. When $d > \nu$, we have $8BC_1 N_{i-1}^{\frac{d-\nu}{d}} \ln^{\frac{\nu}{d}(1+\alpha)} N_{i-1} \leq 8BC_1 T^{\frac{d-\nu}{d}} \ln^{\frac{\nu}{d}(1+\alpha)} T$; therefore,

$$R_T \leq \max\{8B\overline{T}, 2BN_1\} + 8BC_1 T^{\frac{d-\nu}{d}} \left(\ln^{\frac{\nu}{d}(1+\alpha)} T\right) (\log_2 T) = \widetilde{O}\left(T^{\frac{d-\nu}{d}}\right). \tag{56}$$

When $d = \nu$, we have $8BC_1 N_{i-1}^{\frac{d-\nu}{d}} \ln^{\frac{\nu}{d}(1+\alpha)} N_{i-1} \leq 8BC_1 \ln^{1+\alpha} T$; hence,

$$R_T \leq \max\{8B\overline{T}, 2BN_1\} + 8BC_1 \left(\ln^{1+\alpha} T\right) (\log_2 T) = O\left(\ln^{2+\alpha} T\right). \tag{57}$$

Finally, when $d < \nu$, we have $8BC_1 N_{i-1}^{\frac{d-\nu}{d}} \ln^{\frac{\nu}{d}(1+\alpha)} N_{i-1} \leq 8BC_2$ for some constant $C_2 \in (0, \infty)$; therefore,

$$R_T \leq \max\{8B\overline{T}, 2BN_1\} + 8BC_2 (\log_2 T) = O\left(\ln T\right). \tag{58}$$

$$\square$$

---

[4]The application of Lemma 3.1 requires the compactness of the potential maximizers, which is verified by the continuity of $\mathrm{ucb}_i(\cdot)$ under $k_{\mathrm{SE}}$ and $k_{\mathrm{Matérn}}$.

### D.2. Proof of Theorem 4.2

*Proof.* If $T \geq \overline{T}$, from Lemma D.1, we have

$$f(\boldsymbol{x}^*) - f(\widehat{\boldsymbol{x}}_T) \leq \mu_{\lambda^2 \boldsymbol{I}_T}(\boldsymbol{x}^*; \mathbf{X}_T, \boldsymbol{f}_T) + B\sigma_{\lambda^2 \boldsymbol{I}_T}(\boldsymbol{x}^*; \mathbf{X}_T) - \mu_{\lambda^2 \boldsymbol{I}_T}(\widehat{\boldsymbol{x}}_T; \mathbf{X}_T, \boldsymbol{f}_T) + B\sigma_{\lambda^2 \boldsymbol{I}_T}(\widehat{\boldsymbol{x}}_T; \mathbf{X}_T) \quad (59)$$

$$\leq 2B \max_{\boldsymbol{x} \in \mathcal{X}} \sigma_{\lambda^2 \boldsymbol{I}_T}(\boldsymbol{x}; \mathbf{X}_T), \quad (60)$$

where the last line follows from the definition of $\widehat{\boldsymbol{x}}_T$.

**For SE kernel.** From statement 3 in Corollary 3.2 and $\|f\|_\infty \leq B$, we have

$$r_T \leq \begin{cases} 2B & \text{if } T < \overline{T}, \\ 2BC_1 \exp\left(-\frac{1}{2} T^{\frac{1}{d+1}} \ln^{-\alpha} T\right) & \text{if } T \geq \overline{T}, \end{cases} \quad (61)$$

Here, $C_1 \in (0,1)$ is the implied constant in Corollary 3.2. From the above inequality, $\forall T \in N_+ \setminus \{1\}, r_T \leq BC_2 \exp\left(-\frac{1}{2} T^{\frac{1}{d+1}} \ln^{-\alpha} T\right)$ holds for sufficiently large constant $C_2 \in (0, \infty)$, which depends on $C_1, \overline{T}, \alpha$, and $d$. Note that $\overline{T}$ and $C_1$ are the constant that only depends $\alpha, d, \ell$, and $\nu$. This implies $r_T = O\left(\exp\left(-\frac{1}{2} T^{\frac{1}{d+1}} \ln^{-\alpha} T\right)\right)$.

**For Matérn kernel.** From statement 4 in Corollary 3.2, we have

$$r_T \leq \begin{cases} 2B & \text{if } T < \overline{T}, \\ 2BC_1 T^{-\frac{\nu}{d}} \ln^{\frac{\nu}{d}(1+\alpha)} T & \text{if } T \geq \overline{T}, \end{cases} \quad (62)$$

Here, $C_1 \in (0,1)$ is the implied constant in Corollary 3.2. From the above inequality, $\forall T \in N_+ \setminus \{1\}, r_T \leq BC_2 T^{-\frac{\nu}{d}} \ln^{\frac{\nu}{d}(1+\alpha)} T$ holds for sufficiently large constant $C_2 \in (0, \infty)$, which depends on $C_1, \overline{T}, \alpha, d$, and $\nu$. This implies $r_T = \widetilde{O}\left(T^{-\frac{\nu}{d}}\right)$. $\qquad \square$

## E. Proof in Section 5

### E.1. Proof of Theorem 5.3

**Lemma E.1** (Non-adaptive confidence bound for noisy setting, Theorem 1 in (Vakili et al., 2021a)). *Fix any $T \in \mathbb{N}_+$, $\delta \in (0,1)$, $\lambda^2 > 0$, and suppose Assumptions 2.1, 2.2 with $\forall t \in \mathbb{N}_+, \rho_t = \rho \geq 0$. Furthermore, assume $\mathcal{X}$ is finite. Then, if the input sequence $(\boldsymbol{x}_t)_{t \in [T]}$ is independent of the noise sequence $(\epsilon_t)_{t \in [T]}$, the following event holds with probability at least $1 - \delta$:*

$$\forall \boldsymbol{x} \in \mathcal{X}, \ |f(\boldsymbol{x}) - \mu_{\lambda^2 \boldsymbol{I}_t}(\boldsymbol{x}; \mathbf{X}_T, \boldsymbol{y}_T)| \leq \left(B + \frac{\rho}{\lambda} \sqrt{2 \ln \frac{2|\mathcal{X}|}{\delta}}\right) \sigma_{\lambda^2 \boldsymbol{I}_T}(\boldsymbol{x}; \mathbf{X}_T), \quad (63)$$

*where $\mathbf{X}_T = (\boldsymbol{x}_1, \dots, \boldsymbol{x}_T)$ and $\boldsymbol{y}_T = (y_1, \dots, y_T)^\top$.*

*Proof of Theorem 5.3.* From Lemma E.1 and the union bound, the following event holds with probability at least $1 - \delta$:

$$\forall i \in [Q_T], \forall \boldsymbol{x} \in \mathcal{X}, \text{lcb}_i(\boldsymbol{x}) \leq f(\boldsymbol{x}) \leq \text{ucb}_i(\boldsymbol{x}), \quad (64)$$

where $Q_T \in \mathbb{N}_+$ denotes the total batch size of PE. Note that $Q_T \leq 1 + \log_2 T$ holds. Hereafter, we assume the above event holds.

First, the cumulative regret at the first batch is bounded from above by $2BN_1$. Next, for any $i$-th batch with $i \geq 2$, we have

$$\sum_{j=1}^{N_i} f(\boldsymbol{x}^*) - f(\boldsymbol{x}_j^{(i)}) \leq \sum_{j=1}^{N_i} \mathrm{ucb}_{i-1}(\boldsymbol{x}^*) - \mathrm{lcb}_{i-1}(\boldsymbol{x}_j^{(i)}) \tag{65}$$

$$= \sum_{j=1}^{N_i} \mathrm{lcb}_{i-1}(\boldsymbol{x}^*) - \mathrm{ucb}_{i-1}(\boldsymbol{x}_j^{(i)}) + 2\beta^{1/2}\sigma_{\lambda^2\boldsymbol{I}_{N_{i-1}}}(\boldsymbol{x}_j^{(i)}; \mathbf{X}_{N_{i-1}}^{(i-1)}) + 2\beta^{1/2}\sigma_{\lambda^2\boldsymbol{I}_{N_{i-1}}}(\boldsymbol{x}^*; \mathbf{X}_{N_{i-1}}^{(i-1)}) \tag{66}$$

$$\leq \sum_{j=1}^{N_i} \mathrm{lcb}_{i-1}(\boldsymbol{x}^*) - \max_{\boldsymbol{x}\in\mathcal{X}_{i-1}}\mathrm{lcb}_{i-1}(\boldsymbol{x}) + 4B\max_{\boldsymbol{x}\in\mathcal{X}_{i-1}}\sigma_{\lambda^2\boldsymbol{I}_{N_{i-1}}}(\boldsymbol{x}; \mathbf{X}_{N_{i-1}}^{(i-1)}) \tag{67}$$

$$\leq 4N_i\beta^{1/2}\max_{\boldsymbol{x}\in\mathcal{X}_{i-1}}\sigma_{\lambda^2\boldsymbol{I}_{N_{i-1}}}(\boldsymbol{x}; \mathbf{X}_{N_{i-1}}^{(i-1)}). \tag{68}$$

From the definition of $\beta^{1/2}$ and $\lambda^2 = C/B^2$ for some constant $C > 0$, the above inequality implies

$$\sum_{j=1}^{N_i} f(\boldsymbol{x}^*) - f(\boldsymbol{x}_j^{(i)}) \leq 4N_iB\left(1 + C^{-1/2}\rho\sqrt{2\ln\frac{2|\mathcal{X}|(1+\log_2 T)}{\delta}}\right)\max_{\boldsymbol{x}\in\mathcal{X}_{i-1}}\sigma_{\lambda^2\boldsymbol{I}_{N_{i-1}}}(\boldsymbol{x}; \mathbf{X}_{N_{i-1}}^{(i-1)}). \tag{69}$$

Here, let us define $\mathcal{T}$ and $\mathcal{T}^c$ as $\mathcal{T} = \{j \in [N_{i-1}] \mid \lambda^{-1}\sigma_{\lambda^2\boldsymbol{I}_{j-1}}(\boldsymbol{x}_j; \mathbf{X}_{j-1}^{(i-1)}) \leq 1\}$ and $\mathcal{T}^c = [N_{i-1}] \setminus \mathcal{T}$, respectively. From elliptical potential count lemma (Lemma 3.3), we have

$$|\mathcal{T}^c| \leq \min\left\{3\overline{\gamma}\left(3\gamma_{N_{i-1}}(\lambda^2\boldsymbol{I}_{N_{i-1}}), \lambda^2\right), 3\gamma_{N_{i-1}}(\lambda^2\boldsymbol{I}_{N_{i-1}})\right\}. \tag{70}$$

Furthermore, from the definition of $\mathcal{T}$, we have the following inequality as with the Eqs. (12)–(16):

$$\sum_{j\in\mathcal{T}}\sigma_{\lambda^2\boldsymbol{I}_{j-1}}(\boldsymbol{x}_j; \mathbf{X}_{j-1}^{(i-1)}) \leq \sqrt{\lambda^2 N_{i-1}\gamma_{N_{i-1}}(\lambda^2\boldsymbol{I}_{N_{i-1}})}. \tag{71}$$

Then, regarding the maximum of the posterior standard deviation, we have the following from the above inequalities:

$$\max_{\boldsymbol{x}\in\mathcal{X}_{i-1}}\sigma_{\lambda^2\boldsymbol{I}_{N_{i-1}}}(\boldsymbol{x}; \mathbf{X}_{N_{i-1}}^{(i-1)}) \tag{72}$$

$$\leq \frac{1}{N_{i-1}}\sum_{j\in[N_{i-1}]}\sigma_{\lambda^2\boldsymbol{I}_{j-1}}(\boldsymbol{x}_j; \mathbf{X}_{j-1}^{(i-1)}) \tag{73}$$

$$\leq \frac{1}{N_{i-1}}\left[|\mathcal{T}^c| + \sum_{j\in\mathcal{T}}\sigma_{\lambda^2\boldsymbol{I}_{j-1}}(\boldsymbol{x}_j; \mathbf{X}_{j-1}^{(i-1)})\right] \tag{74}$$

$$\leq \frac{1}{N_{i-1}}\left[\min\left\{3\overline{\gamma}\left(3\gamma_{N_{i-1}}(\lambda^2\boldsymbol{I}_{N_{i-1}}), \lambda^2\right), 3\gamma_{N_{i-1}}(\lambda^2\boldsymbol{I}_{N_{i-1}})\right\} + 2\sqrt{\lambda^2 N_{i-1}\gamma_{N_{i-1}}(\lambda^2\boldsymbol{I}_{N_{i-1}})}\right] \tag{75}$$

$$\leq \frac{1}{N_{i-1}}\left[\min\left\{3\overline{\gamma}\left(3\gamma_T(\lambda^2\boldsymbol{I}_T), \lambda^2\right), 3\gamma_T(\lambda^2\boldsymbol{I}_T)\right\} + 2\sqrt{\lambda^2 T\gamma_T(\lambda^2\boldsymbol{I}_T)}\right], \tag{76}$$

where the first inequality follows from the definition of the MVR-selection rule, and the second inequality follows from $\sigma_{\lambda^2\boldsymbol{I}_{j-1}}(\boldsymbol{x}_j; \mathbf{X}_{j-1}^{(i-1)}) \leq k(\boldsymbol{x}_j, \boldsymbol{x}_j) \leq 1$. Combining the above inequality with Eq. (68) and $Q_T \leq (1 + \log_2 T)$, we have

$$R_T \leq 2BN_1 + 8(1 + \log_2 T)\beta^{1/2}\left[\min\left\{3\overline{\gamma}\left(3\gamma_T(\lambda^2\boldsymbol{I}_T), \lambda^2\right), 3\gamma_T(\lambda^2\boldsymbol{I}_T)\right\} + 2\sqrt{\lambda^2 T\gamma_T(\lambda^2\boldsymbol{I}_T)}\right]. \tag{77}$$

**For SE kernel.** Note that $\lambda^2 = \Theta(1/B^2)$, $B = O(\sqrt{T})$, and $\overline{\gamma}(t, \lambda^2) = O(\ln^{d+1}(t/\lambda^2))$. Since $3\gamma_T(\lambda^2\boldsymbol{I}_T) = O(\ln^{d+1}(TB^2))$, we obtain $\min\left\{3\overline{\gamma}\left(3\gamma_T(\lambda^2\boldsymbol{I}_T), \lambda^2\right), 3\gamma_T(\lambda^2\boldsymbol{I}_T)\right\} = O(\ln^{d+1}(TB^2))$. Furthermore, from $\sqrt{\lambda^2 T\gamma_T(\lambda^2\boldsymbol{I}_T)} = O\left(\frac{\sqrt{T}}{B}\sqrt{\ln^{d+1}(TB^2)}\right)$, $\min\left\{3\overline{\gamma}\left(3\gamma_T(\lambda^2\boldsymbol{I}_T), \lambda^2\right), 3\gamma_T(\lambda^2\boldsymbol{I}_T)\right\} + 2\sqrt{\lambda^2 T\gamma_T(\lambda^2\boldsymbol{I}_T)} =$

$O\left(\frac{\sqrt{T}}{B}\sqrt{\ln^{d+1}(TB^2)}\right)$ when $B = O(\sqrt{T})$. Hence, by noting $\beta^{1/2} = \Theta\left(B\sqrt{\ln\frac{|\mathcal{X}|}{\delta}}\right)$ and $B = O(\sqrt{T})$, we have

$$R_T \leq O\left(\max\left\{B, (\ln T)\sqrt{T\left(\ln^{d+1}(TB^2)\right)\left(\ln\frac{|\mathcal{X}|}{\delta}\right)}\right\}\right) \tag{78}$$

$$= O\left((\ln T)\sqrt{T\left(\ln^{d+1}(TB^2)\right)\left(\ln\frac{|\mathcal{X}|}{\delta}\right)}\right). \tag{79}$$

**For Matérn kernel.** Note that $\lambda^2 = \Theta(1/B^2)$, $B = O\left(T^{\frac{2\nu^2+3\nu d}{4d^2+4\nu^2+6\nu d}}\right)$, and $\overline{\gamma}(t, \lambda^2) = \widetilde{O}\left((t/\lambda^2)^{\frac{d}{2\nu+d}}\right)$. Furthermore, we can see that

$$\frac{2\nu^2 + 3\nu d}{4d^2 + 4\nu^2 + 6\nu d} \leq \frac{\nu}{d} \Leftrightarrow 2\nu^2 d + 3\nu d^2 \leq 4\nu d^2 + 4\nu^3 + 6\nu^2 d$$
$$\Leftrightarrow 0 \leq \nu d^2 + 4\nu^3 + 4\nu^2 d$$
$$\Leftrightarrow 0 \leq \nu(d + 2\nu)^2.$$

Thus, from $B = O\left(T^{\frac{2\nu^2+3\nu d}{4d^2+4\nu^2+6\nu d}}\right)$, we can obtain $B = O\left(T^{\frac{\nu}{d}}\right)$. Then, $3\gamma_T(\lambda^2 \boldsymbol{I}_T) = \widetilde{O}\left((TB^2)^{\frac{d}{2\nu+d}}\right)$, $\sqrt{\lambda^2 T \gamma_T(\lambda^2 \boldsymbol{I}_T)} = \widetilde{O}\left(T^{\frac{\nu+d}{2\nu+d}}B^{-\frac{2\nu}{2\nu+d}}\right)$, and $3\overline{\gamma}\left(3\gamma_T(\lambda^2\boldsymbol{I}_T), \lambda^2\right) = \widetilde{O}\left(T^{\left(\frac{d}{2\nu+d}\right)^2}(B^2)^{\frac{d(2\nu+2d)}{(2\nu+d)^2}}\right)$. Therefore, for the second term of Eq. (77), we see that

$$8(1 + \log_2 T)\beta^{1/2}\left[\min\left\{3\overline{\gamma}\left(3\gamma_T(\lambda^2\boldsymbol{I}_T), \lambda^2\right), 3\gamma_T(\lambda^2\boldsymbol{I}_T)\right\} + 2\sqrt{\lambda^2 T \gamma_T(\lambda^2\boldsymbol{I}_T)}\right] \tag{80}$$

$$= \widetilde{O}\left(\beta^{1/2}\left[\min\left\{T^{\left(\frac{d}{2\nu+d}\right)^2}(B^2)^{\frac{d(2\nu+2d)}{(2\nu+d)^2}}, (TB^2)^{\frac{d}{2\nu+d}}\right\} + T^{\frac{\nu+d}{2\nu+d}}B^{-\frac{2\nu}{2\nu+d}}\right]\right). \tag{81}$$

Note that, if $B = \Theta\left(T^{\frac{\nu}{d}}\right)$, then $T^{\left(\frac{d}{2\nu+d}\right)^2}(B^2)^{\frac{d(2\nu+2d)}{(2\nu+d)^2}} = \Theta(T)$ and $(TB^2)^{\frac{d}{2\nu+d}} = \Theta(T)$. Therefore, if $B = O\left(T^{\frac{\nu}{d}}\right)$, $T^{\left(\frac{d}{2\nu+d}\right)^2}(B^2)^{\frac{d(2\nu+2d)}{(2\nu+d)^2}} = O\left((TB^2)^{\frac{d}{2\nu+d}}\right)$ since $\frac{d}{2\nu+d} < \frac{d(2\nu+2d)}{(2\nu+d)^2}$. Thus, Eq. (81) is $\widetilde{O}\left(\beta^{1/2}\left[T^{\left(\frac{d}{2\nu+d}\right)^2}(B^2)^{\frac{d(2\nu+2d)}{(2\nu+d)^2}} + T^{\frac{\nu+d}{2\nu+d}}B^{-\frac{2\nu}{2\nu+d}}\right]\right)$. Then, we can see that

$$\frac{T^{\left(\frac{d}{2\nu+d}\right)^2}(B^2)^{\frac{d(2\nu+2d)}{(2\nu+d)^2}}}{T^{\frac{\nu+d}{2\nu+d}}B^{-\frac{2\nu}{2\nu+d}}} = T^{\frac{d^2-2\nu^2-3\nu d-d^2}{(2\nu+d)^2}}B^{\frac{4d\nu+4d^2+4\nu^2+2\nu d}{(2\nu+d)^2}} \tag{82}$$

$$= T^{\frac{-2\nu^2-3\nu d}{(2\nu+d)^2}}B^{\frac{4d^2+4\nu^2+6\nu d}{(2\nu+d)^2}}. \tag{83}$$

Therefore, if $B = O\left(T^{\frac{2\nu^2+3\nu d}{4d^2+4\nu^2+6\nu d}}\right)$, then $T^{\left(\frac{d}{2\nu+d}\right)^2}(B^2)^{\frac{d(2\nu+2d)}{(2\nu+d)^2}} = O\left(T^{\frac{\nu+d}{2\nu+d}}B^{-\frac{2\nu}{2\nu+d}}\right)$. Thus, from the assumption $B = O\left(T^{\frac{2\nu^2+3\nu d}{4d^2+4\nu^2+6\nu d}}\right)$, $T^{\left(\frac{d}{2\nu+d}\right)^2}(B^2)^{\frac{d(2\nu+2d)}{(2\nu+d)^2}} = O\left(T^{\frac{\nu+d}{2\nu+d}}B^{-\frac{2\nu}{2\nu+d}}\right)$. Hence, Eq. (81) is $\widetilde{O}\left(\beta^{1/2}T^{\frac{\nu+d}{2\nu+d}}B^{-\frac{2\nu}{2\nu+d}}\right) = \widetilde{O}\left(T^{\frac{\nu+d}{2\nu+d}}B^{\frac{d}{2\nu+d}}\right)$ because of $\beta^{1/2} = \Theta\left(B\sqrt{\ln\frac{|\mathcal{X}|}{\delta}}\right)$. Therefore, we have

$$R_T = \widetilde{O}\left(\max\left\{B, T^{\frac{\nu+d}{2\nu+d}}B^{\frac{d}{2\nu+d}}\right\}\right). \tag{84}$$

Furthermore, noting that $B = \Theta\left(T^{\frac{\nu+d}{2\nu}}\right) \Leftrightarrow B = \Theta\left(T^{\frac{\nu+d}{2\nu+d}}B^{\frac{d}{2\nu+d}}\right)$, since $\frac{\nu+d}{2\nu} = \frac{1}{2} + \frac{d}{2\nu} > \frac{2\nu^2+3\nu d}{4d^2+4\nu^2+6\nu d}$, we see $B = O\left(T^{\frac{\nu+d}{2\nu+d}}B^{\frac{d}{2\nu+d}}\right)$. Consequently, we have

$$R_T = \widetilde{O}\left(T^{\frac{\nu+d}{2\nu+d}}B^{\frac{d}{2\nu+d}}\right). \tag{85}$$

$\square$

### E.2. Proof of Theorem 5.1

*Proof.* From Lemma E.1, we have the following with probability at least $1 - \delta$:

$$f(\boldsymbol{x}^*) - f(\widehat{\boldsymbol{x}}_T) \leq \mu_{\lambda^2 \boldsymbol{I}_T}(\boldsymbol{x}^*; \mathbf{X}_T, \boldsymbol{y}_T) + \beta^{1/2}\sigma_{\lambda^2 \boldsymbol{I}_T}(\boldsymbol{x}^*; \mathbf{X}_T) - \mu_{\lambda^2 \boldsymbol{I}_T}(\widehat{\boldsymbol{x}}_T; \mathbf{X}_T, \boldsymbol{y}_T) + \beta^{1/2}\sigma_{\lambda^2 \boldsymbol{I}_T}(\widehat{\boldsymbol{x}}_T; \mathbf{X}_T) \quad (86)$$

$$\leq 2\beta^{1/2} \max_{\boldsymbol{x} \in \mathcal{X}} \sigma_{\lambda^2 \boldsymbol{I}_T}(\boldsymbol{x}; \mathbf{X}_T) \quad (87)$$

$$= 2B\left(1 + \rho\sqrt{2\ln\frac{2|\mathcal{X}|}{\delta}}\right) \max_{\boldsymbol{x} \in \mathcal{X}} \sigma_{\lambda^2 \boldsymbol{I}_T}(\boldsymbol{x}; \mathbf{X}_T). \quad (88)$$

Here, when $k = k_{\mathrm{SE}}$, note that the condition $B = O(\exp(T^{\frac{1}{d+1}}\ln^{-\alpha}(1+T)))$ implies $\lambda^2 = O(\exp(-\frac{1}{2}T^{\frac{1}{d+1}}\ln^{-\alpha}(1+T)))$. Therefore, in the SE kernel, from statement 1 in Corollary 3.2, we have the following inequality for any $T \geq \overline{T}$.

$$r_T \leq 8\left(1 + \rho\sqrt{2\ln\frac{2|\mathcal{X}|}{\delta}}\right)\sqrt{\frac{\overline{C}\ln^{d+1}(TB^2)}{T}} = O\left(\sqrt{\frac{\ln^{d+1}(TB^2)}{T}}\right), \quad (89)$$

where $\overline{C} > 0$ is the implied constant of the upper bound of MIG. As for the Matérn kernel, leveraging statement 2 in Corollary 3.2 by noting the condition of $B$, we have

$$r_T \leq \frac{8}{T}\left(1 + \rho\sqrt{2\ln\frac{2|\mathcal{X}|}{\delta}}\right)\sqrt{\overline{C}B^{\frac{2d}{2\nu+d}}T^{\frac{2(\nu+d)}{2\nu+d}}\ln^{\frac{2\nu}{2\nu+d}}(TB^2)} = \widetilde{O}\left(B^{\frac{d}{2\nu+d}}T^{-\frac{\nu}{2\nu+d}}\right), \quad (90)$$

for any $T \geq \overline{T}$. $\qquad\square$

## F. Proof in Section 6

### F.1. Proof of Theorem 6.3

**Lemma F.1** (Non-adaptive confidence bound for non-stationary variance setting, extension of Theorem 1 in (Vakili et al., 2021a)). *Fix any $T \in \mathbb{N}_+$, $\delta \in (0, 1)$, any non-negative sequence $(\lambda_t)_{t \in \mathbb{N}_+}$, and suppose Assumptions 2.1 and 2.2 with $\rho_t \leq \lambda_t$. Furthermore, assume $\mathcal{X}$ is finite. Then, if the input sequence $(\boldsymbol{x}_t)_{t \in [T]}$ is independent of the noise sequence $(\epsilon_t)_{t \in [T]}$, the following event holds with probability at least $1 - \delta$:*

$$\forall \boldsymbol{x} \in \mathcal{X}, \; |f(\boldsymbol{x}) - \mu_{\Sigma_T}(\boldsymbol{x}; \mathbf{X}_T, \boldsymbol{y}_T)| \leq \left(B + \sqrt{2\ln\frac{2|\mathcal{X}|}{\delta}}\right)\sigma_{\Sigma_T}(\boldsymbol{x}; \mathbf{X}_T), \quad (91)$$

*where $\mathbf{X}_T = (\boldsymbol{x}_1, \ldots, \boldsymbol{x}_T)$, $\boldsymbol{y}_T = (y_1, \ldots, y_T)^\top$, and $\Sigma_T = \mathrm{diag}(\lambda_1^2, \ldots, \lambda_T^2)$.*

*Proof.* The proof almost directly follows by replacing the original Proposition 1 in (Vakili et al., 2021a). Let $\boldsymbol{Z}_T(\boldsymbol{x})$ be $\boldsymbol{Z}_T(\boldsymbol{x})^\top = \boldsymbol{k}(\boldsymbol{x}, \mathbf{X}_T)^\top(\mathbf{K}(\mathbf{X}_T, \mathbf{X}_T) + \Sigma_T)^{-1}$. Then, following the proof of Proposition 1 in (Vakili et al., 2021a), we obtain its extension as

$$\sigma_{\Sigma_T}^2(\boldsymbol{x}; \mathbf{X}_T) = \sup_{f: \|f\|_{\mathcal{H}_k} \leq 1} \left(f(\boldsymbol{x}) - \boldsymbol{Z}_T(\boldsymbol{x})^\top \boldsymbol{f}_T\right)^2 + \|\boldsymbol{Z}_T(\boldsymbol{x})\|_{\Sigma_T}^2, \quad (92)$$

where $\boldsymbol{f}_T = (f(\boldsymbol{x}_1), \ldots, f(\boldsymbol{x}_T))^\top$ and $\|\boldsymbol{Z}_T(\boldsymbol{x})\|_{\Sigma_T} = \boldsymbol{Z}_T(\boldsymbol{x})^\top \Sigma_T \boldsymbol{Z}_T(\boldsymbol{x})$. Next, we consider replacing Proposition 1 with Eq. (92) in the original proof of Theorem 1 in (Vakili et al., 2021a). As with the original proof, we decompose $|f(\boldsymbol{x}) - \mu_{\Sigma_T}(\boldsymbol{x}; \mathbf{X}_T, \boldsymbol{y}_T)|$ into two terms:

$$|f(\boldsymbol{x}) - \mu_{\Sigma_T}(\boldsymbol{x}; \mathbf{X}_T, \boldsymbol{y}_T)| \leq |f(\boldsymbol{x}) - \boldsymbol{Z}_T(\boldsymbol{x})^\top \boldsymbol{f}_T| + |\boldsymbol{Z}_T(\boldsymbol{x})^\top \boldsymbol{\epsilon}_T|, \quad (93)$$

where $\boldsymbol{\epsilon}_T = (\epsilon_1, \ldots, \epsilon_T)^\top$. The first term of r.h.s. is bounded from above as

$$|f(\boldsymbol{x}) - \boldsymbol{Z}_T(\boldsymbol{x})^\top \boldsymbol{f}_T| = B\left|\frac{f(\boldsymbol{x})}{B} - \boldsymbol{Z}_T(\boldsymbol{x})^\top\left(\frac{\boldsymbol{f}_T}{B}\right)\right| \leq B\sigma_{\Sigma_T}(\boldsymbol{x}; \mathbf{X}_T). \quad (94)$$

The last inequality follows from Eq. (92) since $\|f(\cdot)/B\|_{\mathcal{H}_k} \leq 1$ holds by Assumption 2.1. Regarding the second term, $\boldsymbol{Z}_T(\boldsymbol{x})^\top \boldsymbol{\epsilon}_T$ is the sub-Gaussian random variable from the independence assumption between $(\boldsymbol{x}_t)_{t \in [T]}$ and $(\epsilon_t)_{t \in [T]}$, and its concentration inequality is obtained by evaluating the upper bound of the moment generating function. By following the proof of Theorem 1 in (Vakili et al., 2021a), we have

$$\mathbb{E}\left[\exp(\boldsymbol{Z}_T(\boldsymbol{x})^\top \boldsymbol{\epsilon}_T)\right] \leq \exp\left(\frac{\|\boldsymbol{Z}_T(\boldsymbol{x})\|_{\mathrm{diag}(\rho_1^2,\ldots,\rho_t^2)}^2}{2}\right) \tag{95}$$

$$\leq \exp\left(\frac{\|\boldsymbol{Z}_T(\boldsymbol{x})\|_{\Sigma_T}^2}{2}\right) \tag{96}$$

$$\leq \exp\left(\frac{\sigma_{\Sigma_T}^2(\boldsymbol{x}; \mathbf{X}_T)}{2}\right), \tag{97}$$

where the second inequality follows from the assumption $\rho_t \leq \lambda_t$, and the third inequality follows from Eq. (92). The above upper bound for the moment-generating function implies the following concentration inequality for any $\boldsymbol{x} \in \mathcal{X}$ from Chernoff-Hoeffding inequality:

$$\mathbb{P}\left(|\boldsymbol{Z}_T(\boldsymbol{x})^\top \boldsymbol{\epsilon}_T| \leq \sqrt{2\ln\frac{2}{\delta}}\sigma_{\Sigma_T}(\boldsymbol{x}; \mathbf{X}_T)\right) \geq 1 - \delta. \tag{98}$$

Finally, we obtain the desired result by combining Eq. (93) with Eqs. (94), (98), and the union bound. $\qquad\square$

*Proof of Theorem 6.3.* From Lemma F.1 and the union bound, the confidence bound is valid with probability at least $1 - \delta$. Hereafter, From the assumption $V_T = \Omega(1)$, we have the constant $C > 0$ such that $V_T \geq C$. Below, we prove each statement of the theorem separately based on the setting of the kernel.

**For $k = k_{\mathrm{SE}}$.** Let us set $\overline{T}$ as $\overline{T} = \min\{T \in \mathbb{N}_+ \mid \forall t \geq T, t/2 \geq 4\gamma_t(Ct^{-1}\boldsymbol{I}_t)\}$. Note that the constant $\overline{T}$ is well-defined, since $\gamma_t(Ct^{-1}\boldsymbol{I}_t) = O(\ln^{d+1} t^2) = o(t)$. Furthermore, let $\bar{i} \in \mathbb{N}_+$ be the first batch index such that $N_{\bar{i}} \geq \overline{T}$ holds. Then, the cumulative regret before the start of $(\bar{i}+1)$-th batch is bounded from above by $\max\{8B\overline{T}, 2BN_1\}$. Next, we consider the cumulative regret after the start of $(\bar{i}+1)$-th batch. We apply statement 2 of Lemma 3.1 with the following arguments:

1. Set $\widetilde{\lambda}_j^{(i)}$ as $\left(\widetilde{\lambda}_j^{(i)}\right)^2 = \max\left\{\left(\lambda_j^{(i)}\right)^2, \frac{V_T}{N_i}\right\}$. Then, we have $\left(\widetilde{\lambda}_j^{(i)}\right)^2 \geq V_T/N_i \geq C/N_i$. Furthermore, we set $\widetilde{\Sigma}_{N_i}^{(i)} = \mathrm{diag}\left(\left(\widetilde{\lambda}_1^{(i)}\right)^2, \ldots, \left(\widetilde{\lambda}_{N_i}^{(i)}\right)^2\right)$.

2. From the above lower bound of $\widetilde{\lambda}_j^{(i)}$, we have $\gamma_{N_i}\left(\widetilde{\Sigma}_{N_i}^{(i)}\right) \leq \gamma_{N_i}(CN_i^{-1}\boldsymbol{I}_{N_i})$, which implies $N_i/2 \geq 4\gamma_{N_i}(\widetilde{\Sigma}_{N_i}^{(i)})$ if $N_i \geq \overline{T}$. Therefore, applying statement 2 of Lemma 3.1, we have

$$\max_{\boldsymbol{x} \in \mathcal{X}_i} \sigma_{\Sigma_{N_i}^{(i)}}(\boldsymbol{x}; \mathbf{X}_{N_i}^{(i)}) \leq \frac{4}{N_i}\sqrt{\left(\sum_{j=1}^{N_i}\left(\widetilde{\lambda}_j^{(i)}\right)^2\right)\gamma_{N_i}\left(\widetilde{\Sigma}_{N_i}^{(i)}\right)} \quad \text{for all } i \geq \bar{i}. \tag{99}$$

The equation inside the square root in r.h.s. of Eq. (99) can be bounded from above further as

$$\left(\sum_{j=1}^{N_i}\left(\widetilde{\lambda}_j^{(i)}\right)^2\right)\gamma_{N_i}\left(\widetilde{\Sigma}_{N_i}^{(i)}\right) \leq \left[\sum_{j=1}^{N_i}\left(\rho_j^{(i)}\right)^2 + \sum_{j=1}^{N_i}\frac{V_T}{N_i}\right]\gamma_{N_i}\left(\widetilde{\Sigma}_{N_i}^{(i)}\right) \tag{100}$$

$$\leq 2V_T\gamma_{N_i}\left(\frac{V_T}{N_i}\boldsymbol{I}_{N_i}\right) \tag{101}$$

$$\leq 2V_T\gamma_T\left(\frac{V_T}{T}\boldsymbol{I}_T\right) \tag{102}$$

$$\leq O\left(V_T\ln^{d+1}\frac{T^2}{V_T}\right). \tag{103}$$

Since $\rho_j^{(i)} = \lambda_j^{(i)}$, the total cumulative regret is bounded from above with probability at least $1 - \delta$ as

$$R_T = \sum_{i \leq \bar{i}} \sum_{j=1}^{N_i} f(\boldsymbol{x}^*) - f(\boldsymbol{x}_j^{(i)}) + \sum_{i > \bar{i}} \sum_{j=1}^{N_i} f(\boldsymbol{x}^*) - f(\boldsymbol{x}_j^{(i)}) \tag{104}$$

$$\leq \max\{8B\overline{T}, 2BN_1\} + 4\beta^{1/2} \sum_{i > \bar{i}} N_i \max_{\boldsymbol{x} \in \mathcal{X}_{i-1}} \sigma_{\Sigma_{N_{i-1}}^{(i-1)}}(\boldsymbol{x}; \mathbf{X}_{N_{i-1}}^{(i-1)}) \tag{105}$$

$$\leq \max\{8B\overline{T}, 2BN_1\} + 32\beta^{1/2} \sum_{i > \bar{i}} \sqrt{\left(\sum_{j=1}^{N_{i-1}} \left(\widetilde{\lambda}_j^{(i-1)}\right)^2\right) \gamma_{N_{i-1}}\left(\widetilde{\Sigma}_{N_{i-1}}^{(i-1)}\right)} \tag{106}$$

$$\leq \max\{8B\overline{T}, 2BN_1\} + O\left(\beta^{1/2}(1 + \log_2 T)\sqrt{V_T \ln^{d+1} \frac{T^2}{V_T}}\right) \tag{107}$$

$$\leq O\left((\ln T)\sqrt{V_T \left(\ln^{d+1} \frac{T^2}{V_T}\right)\left(\ln \frac{|\mathcal{X}|}{\delta}\right)}\right), \tag{108}$$

where the first inequality follows from the standard analysis of the PE (e.g., Eqs. (65)–(68)) with Lemma F.1. Furthermore, the second inequality uses the fact that the total number of batches is at most $1 + \log_2 T$.

**For $k = k_{\text{Matérn}}$.** The proof is almost the same as that of the SE kernel, while we need to introduce the additional lower bound to set $\widetilde{\lambda}_j^{(i)}$. Let us set $\overline{T}$ as $\overline{T} = \min\{T \in \mathbb{N}_+ \mid \forall t \geq T, t/2 \geq 4\gamma_t(\overline{\lambda}_t^2 \boldsymbol{I}_t)\}$, where we set $\overline{\lambda}_t^2 = t^{-\frac{2\nu}{d}} \ln^{\frac{2\nu(1+\alpha)}{d}}(1 + t)$ here, where $\alpha > 0$ is any fixed constant. Next, set $\widetilde{\lambda}_j^{(i)}$ as $\left(\widetilde{\lambda}_j^{(i)}\right)^2 = \max\left\{\left(\lambda_j^{(i)}\right)^2, \frac{V_T}{N_i}, \overline{\lambda}_{N_i}^2\right\}$. Then, as with the proof for the case $k = k_{\text{SE}}$, the following statement holds from statement 2 of Lemma 3.1:

$$\max_{\boldsymbol{x} \in \mathcal{X}_i} \sigma_{\Sigma_{N_i}^{(i)}}(\boldsymbol{x}; \mathbf{X}_{N_i}^{(i)}) \leq \frac{4}{N_i} \sqrt{\left(\sum_{j=1}^{N_i} \left(\widetilde{\lambda}_j^{(i)}\right)^2\right) \gamma_{N_i}\left(\widetilde{\Sigma}_{N_i}^{(i)}\right)} \text{ for all } i \geq \bar{i}, \tag{109}$$

where $\bar{i} \in \mathbb{N}_+$ is the first batch index such that $N_{\bar{i}} \geq \overline{T}$ holds. Furthermore,

$$\left(\sum_{j=1}^{N_i} \left(\widetilde{\lambda}_j^{(i)}\right)^2\right) \gamma_{N_i}\left(\widetilde{\Sigma}_{N_i}^{(i)}\right) \leq \left[\sum_{j=1}^{N_i} \left(\rho_j^{(i)}\right)^2 + \sum_{j=1}^{N_i} \frac{V_T}{N_i} + \sum_{j=1}^{N_i} \overline{\lambda}_{N_i}^2\right] \gamma_{N_i}\left(\widetilde{\Sigma}_{N_i}^{(i)}\right) \tag{110}$$

$$\leq \left(2V_T + N_i \overline{\lambda}_{N_i}^2\right) \gamma_{N_i}\left(\widetilde{\Sigma}_{N_i}^{(i)}\right). \tag{111}$$

In the above inequality, we can see that $\gamma_{N_i}\left(\widetilde{\Sigma}_{N_i}^{(i)}\right) \leq \gamma_{N_i}\left(\frac{V_T}{N_i} \boldsymbol{I}_{N_i}\right) \leq \gamma_T\left(\frac{V_T}{T} \boldsymbol{I}_T\right)$. Furthermore, if $V_T/N_i \geq \overline{\lambda}_{N_i}^2$, then, $2V_T + N_i \overline{\lambda}_{N_i}^2 \leq 3V_T$, which implies

$$\sqrt{\left(\sum_{j=1}^{N_i} \left(\widetilde{\lambda}_j^{(i)}\right)^2\right) \gamma_{N_i}\left(\widetilde{\Sigma}_{N_i}^{(i)}\right)} \leq \sqrt{3V_T \gamma_T\left(\frac{V_T}{T} \boldsymbol{I}_T\right)} = \widetilde{O}\left(V_T^{\frac{\nu}{2\nu+d}} T^{\frac{d}{2\nu+d}}\right). \tag{112}$$

On the other hand, $\gamma_{N_i}\left(\widetilde{\Sigma}_{N_i}^{(i)}\right) \leq \gamma_{N_i}\left(\overline{\lambda}_{N_i}^2 \boldsymbol{I}_{N_i}\right)$ also holds. If $V_T/N_i \leq \overline{\lambda}_{N_i}^2$, then, $2V_T + N_i \overline{\lambda}_{N_i}^2 \leq 3N_i \overline{\lambda}_{N_i}^2$, which implies

$$\sqrt{\left(\sum_{j=1}^{N_i} \left(\widetilde{\lambda}_j^{(i)}\right)^2\right) \gamma_{N_i}\left(\widetilde{\Sigma}_{N_i}^{(i)}\right)} \leq \sqrt{3N_i \overline{\lambda}_{N_i}^2 \gamma_{N_i}\left(\overline{\lambda}_{N_i}^2 \boldsymbol{I}_{N_i}\right)} = \widetilde{O}\left(N_i^{\frac{d-\nu}{d}}\right) \leq \begin{cases} \widetilde{O}(1) & \text{if } d \leq \nu, \\ \widetilde{O}\left(T^{\frac{d-\nu}{d}}\right) & \text{if } d > \nu. \end{cases} \tag{113}$$

Therefore, since $T^{\frac{d}{2\nu+d}} \geq T^{\frac{d-\nu}{d}}$ if $d \leq 2\nu$ and $V_T = \Omega(1)$, we have

$$\sqrt{\left(\sum_{j=1}^{N_i}\left(\widetilde{\lambda}_j^{(i)}\right)^2\right)\gamma_{N_i}\left(\widetilde{\Sigma}_{N_i}^{(i)}\right)} = \begin{cases} \widetilde{O}\left(V_T^{\frac{\nu}{2\nu+d}}T^{\frac{d}{2\nu+d}}\right) & \text{if } d \leq 2\nu, \\ \widetilde{O}\left(\max\left\{V_T^{\frac{\nu}{2\nu+d}}T^{\frac{d}{2\nu+d}}, T^{\frac{d-\nu}{d}}\right\}\right) & \text{if } d > 2\nu. \end{cases} \tag{114}$$

Hence, as with Eq. (104) of the proof for $k = k_{\mathrm{SE}}$, with probability at least $1 - \delta$, we have

$$R_T \leq 8B\overline{T} + 16\beta^{1/2}(1 + \log_2 T)\sqrt{\left(\sum_{j=1}^{N_i}\left(\widetilde{\lambda}_j^{(i)}\right)^2\right)\gamma_{N_i}\left(\widetilde{\Sigma}_{N_i}^{(i)}\right)} \tag{115}$$

$$= \begin{cases} \widetilde{O}\left(V_T^{\frac{\nu}{2\nu+d}}T^{\frac{d}{2\nu+d}}\right) & \text{if } d \leq 2\nu, \\ \widetilde{O}\left(\max\left\{V_T^{\frac{\nu}{2\nu+d}}T^{\frac{d}{2\nu+d}}, T^{\frac{d-\nu}{d}}\right\}\right) & \text{if } d > 2\nu. \end{cases} \tag{116}$$

Here, $V_T = O(T^{\frac{d-2\nu}{d}})$ implies $V_T^{\frac{\nu}{2\nu+d}}T^{\frac{d}{2\nu+d}} = O(T^{\frac{d-\nu}{d}})$, and $V_T = \Omega(T^{\frac{d-2\nu}{d}})$ implies vise versa; therefore, by combining the condition $V_T = \Omega(1)$, we have

$$R_T = \begin{cases} \widetilde{O}\left(V_T^{\frac{\nu}{2\nu+d}}T^{\frac{d}{2\nu+d}}\right) & \text{if } d \leq 2\nu, \\ \widetilde{O}\left(V_T^{\frac{\nu}{2\nu+d}}T^{\frac{d}{2\nu+d}}\right) & \text{if } d > 2\nu \text{ and } V_T = \Omega\left(T^{\frac{d-2\nu}{d}}\right), \\ \widetilde{O}\left(T^{\frac{d-\nu}{d}}\right) & \text{if } d > 2\nu \text{ and } V_T = O\left(T^{\frac{d-2\nu}{d}}\right). \end{cases} \tag{117}$$

$\square$

### F.2. Proof of Theorem 6.4

*Proof.* Set $\beta^{1/2} = \left(B + \sqrt{2\ln\frac{2|\mathcal{X}|}{\delta}}\right)$. From Lemma F.1, with probability at least $1 - \delta$, we have

$$f(\boldsymbol{x}^*) - f(\widehat{\boldsymbol{x}}_T) \leq \mu_{\Sigma_T}(\boldsymbol{x}^*; \mathbf{X}_T, \boldsymbol{y}_T) + \beta^{1/2}\sigma_{\Sigma_T}(\boldsymbol{x}^*; \mathbf{X}_T) - \mu_{\Sigma_T}(\widehat{\boldsymbol{x}}_T; \mathbf{X}_T, \boldsymbol{y}_T) + \beta^{1/2}\sigma_{\Sigma_T}(\widehat{\boldsymbol{x}}_T; \mathbf{X}_T) \tag{118}$$

$$\leq 2\beta^{1/2}\max_{\boldsymbol{x}\in\mathcal{X}}\sigma_{\Sigma_T}(\boldsymbol{x}; \mathbf{X}_T). \tag{119}$$

We first consider the case where $k = k_{\mathrm{SE}}$. Let us respectively define $\overline{T}, \widetilde{\lambda}_t, \widetilde{\Sigma}_T$ as $\overline{T} = \min\{T \in \mathbb{N}_+ \mid \forall t \geq T, t/2 \geq 4\gamma_t(Ct^{-1}\boldsymbol{I}_t)\}$, $\widetilde{\lambda}_t^2 = \max\{\lambda_t^2, V_T/T\}$, and $\widetilde{\Sigma}_T = \mathrm{diag}(\widetilde{\lambda}_1^2, \ldots, \widetilde{\lambda}_T^2)$. Here, $C > 0$ is the constant such that $V_T \geq C$. Note that the existence of $C$ is guaranteed by the assumption $V_T = \Omega(1)$. Then, as with the arguments of the proof of Theorem 6.3,

$$\max_{\boldsymbol{x}\in\mathcal{X}}\sigma_{\Sigma_T}(\boldsymbol{x}; \mathbf{X}_T) \leq \frac{4}{T}\sqrt{\left(\sum_{t=1}^{T}\widetilde{\lambda}_t^2\right)\gamma_T\left(\widetilde{\Sigma}_T\right)} \quad \text{for all } \mathrm{T} \geq \overline{\mathrm{T}}. \tag{120}$$

From the definition of $\widetilde{\lambda}_t$, the above inequality implies, for all $T \geq \overline{T}$,

$$f(\boldsymbol{x}^*) - f(\widehat{\boldsymbol{x}}_T) \leq \frac{8\beta^{1/2}}{T}\sqrt{\left(\sum_{t=1}^{T}\widetilde{\lambda}_t^2\right)\gamma_T\left(\widetilde{\Sigma}_T\right)} \tag{121}$$

$$\leq \frac{8\beta^{1/2}}{T}\sqrt{\left(\sum_{t=1}^{T}\rho_t^2 + \sum_{t=1}^{T}\frac{V_T}{T}\right)\gamma_T\left(\widetilde{\Sigma}_T\right)} \tag{122}$$

$$\leq \frac{8\beta^{1/2}}{T}\sqrt{2V_T\gamma_T\left(\frac{V_T}{T}\boldsymbol{I}_T\right)} \tag{123}$$

$$= O\left(\sqrt{\frac{V_T}{T^2}\left(\ln^{d+1}\frac{T^2}{V_T}\right)\left(\ln\frac{|\mathcal{X}|}{\delta}\right)}\right). \tag{124}$$

Next, when $k = k_{\text{Matérn}}$, we set $\overline{T}, \widetilde{\lambda}_t, \widetilde{\Sigma}_T$ as $\overline{T} = \min\{T \in \mathbb{N}_+ \mid \forall t \geq T, t/2 \geq 4\gamma_t(\overline{\lambda}_t^2 \boldsymbol{I}_t)\}$, $\widetilde{\lambda}_t^2 = \max\{\lambda_t^2, V_T/T, \overline{\lambda}_T^2\}$, and $\widetilde{\Sigma}_T = \text{diag}(\widetilde{\lambda}_1^2, \ldots, \widetilde{\lambda}_T^2)$, respectively. Here, $\overline{\lambda}_t^2 = t^{-\frac{2\nu}{d}} \ln^{\frac{2\nu(1+\alpha)}{d}}(1+t)$, where $\alpha > 0$ is any fixed constant. Then, as with the arguments of the proof of Theorem 6.3, statement (120) also holds for $k = k_{\text{Matérn}}$. Therefore,

$$f(\boldsymbol{x}^*) - f(\widehat{\boldsymbol{x}}_T) \leq \frac{8\beta^{1/2}}{T} \sqrt{\left(\sum_{t=1}^T \widetilde{\lambda}_t^2\right) \gamma_T\left(\widetilde{\Sigma}_T\right)} \tag{125}$$

$$\leq \frac{8\beta^{1/2}}{T} \sqrt{\left(2V_T + T\overline{\lambda}_T^2\right) \gamma_T\left(\widetilde{\Sigma}_T\right)}. \tag{126}$$

As with the proof of Theorem 6.3, considering the two cases: $V_T/T \geq \overline{\lambda}_T^2$ or not, we obtain

$$\sqrt{\left(2V_T + T\overline{\lambda}_T^2\right) \gamma_T\left(\widetilde{\Sigma}_T\right)} = \begin{cases} \widetilde{O}\left(V_T^{\frac{\nu}{2\nu+d}} T^{\frac{d}{2\nu+d}}\right) & \text{if } \frac{V_T}{T} \geq \overline{\lambda}_T^2, \\ \widetilde{O}\left(T^{\frac{d-\nu}{d}}\right) & \text{if } \frac{V_T}{T} < \overline{\lambda}_T^2. \end{cases} \tag{127}$$

Therefore, we have

$$f(\boldsymbol{x}^*) - f(\widehat{\boldsymbol{x}}_T) = \widetilde{O}\left(\max\left\{T^{-\frac{\nu}{d}}, V_T^{\frac{\nu}{2\nu+d}} T^{-\frac{2\nu}{2\nu+d}}\right\}\right). \tag{128}$$

Finally, since $V_T = O(T^{\frac{d-2\nu}{d}})$ implies $V_T^{\frac{\nu}{2\nu+d}} T^{-\frac{2\nu}{2\nu+d}} = O(T^{-\frac{\nu}{d}})$, and $V_T = \Omega(1)$, we have

$$f(\boldsymbol{x}^*) - f(\widehat{\boldsymbol{x}}_T) = \begin{cases} \widetilde{O}\left(V_T^{\frac{\nu}{2\nu+d}} T^{-\frac{2\nu}{2\nu+d}}\right) & \text{if } d \leq 2\nu, \\ \widetilde{O}\left(V_T^{\frac{\nu}{2\nu+d}} T^{-\frac{2\nu}{2\nu+d}}\right) & \text{if } d > 2\nu \text{ and } V_T = \Omega\left(T^{\frac{d-2\nu}{d}}\right), \\ \widetilde{O}\left(T^{-\frac{\nu}{d}}\right) & \text{if } d > 2\nu \text{ and } V_T = O\left(T^{\frac{d-2\nu}{d}}\right). \end{cases} \tag{129}$$

$\square$

## G. Pseudo Code of PE and MVR

Algorithms 1 and 2 show the pseudo-code of PE and MVR, respectively. In the PE algorithm, we denote $\boldsymbol{x}_j^{(i)}, y_j^{(i)}$, and $\epsilon_j^{(i)}$ as the selected query point, observed output, and the observation noise at step $j$ on $i$-th batch, respectively.

## H. Pseudo Code of VA-PE and VA-MVR

Algorithms 3 and 4 show the pseudo-code of VA-PE and VA-MVR described in Section 6, respectively.

## I. VA-GP-UCB Algorithm

We consider the GP-UCB-based algorithm as the extension of the variance-aware UCB-style algorithms (Zhou et al., 2021; Zhang et al., 2021; Zhou & Gu, 2022) in linear bandits.

**Algorithm.** Algorithm 5 shows the pseudo-code of VA-GP-UCB algorithm. Overall, the algorithm construction is almost the same as the GP-UCB algorithm with a heteroscedastic GP model. The only difference is the application of the lower threshold $\zeta > 0$ for the variance parameter (Line 7 in Algorithm 5). Intuitively, such a lower threshold has a role in preventing an explosion of the MIG in the first term of the elliptical potential count lemma (Lemma C.1) in the analysis. Note that such a lower threshold is also leveraged in the existing variance-aware UCB-style algorithms (Zhou et al., 2021; Zhang et al., 2021; Zhou & Gu, 2022). We believe this algorithm construction is the most natural kernelized extension of the methods proposed in (Zhou et al., 2021; Zhang et al., 2021; Zhou & Gu, 2022).

**Theoretical analysis for VA-GP-UCB.**

**Lemma I.1** (Adaptive confidence bound in non-stationary variance, e.g., Lemma 7 in (Kirschner & Krause, 2018)). *Fix any strictly positive sequence $(\lambda_t)_{t \in \mathbb{N}_+}$ and define $\Sigma_t$ as $\Sigma_t = \text{diag}(\lambda_1^2, \ldots, \lambda_t^2)$. Suppose Assumptions 2.1 and 2.2 holds with*

---

**Algorithm 1** Phased Elimination (PE)

---

**Require:** Confidence width parameter $\beta^{1/2} > 0$, initial batch size $N_1$, noise variance parameter $\lambda^2 \geq 0$.

1: Initialize the potential maximizer $\mathcal{X}_1 \leftarrow \mathcal{X}$.

2: **for** $i = 1, 2, \ldots$ **do**

3:     $\mathbf{X}_0^{(i)} = \emptyset$.

4:     **for** $j = 1, \ldots, N_i$ **do**

5:         $\boldsymbol{x}_j^{(i)} \leftarrow \arg\max_{\boldsymbol{x} \in \mathcal{X}_i} \sigma_{\lambda^2 \boldsymbol{I}_{j-1}}(\boldsymbol{x}; \mathbf{X}_{j-1}^{(i)})$.

6:         Observe $y_j^{(i)} = f\left(\boldsymbol{x}_j^{(i)}\right) + \epsilon_j^{(i)}$.

7:         $\mathbf{X}_j^{(i)} \leftarrow [\boldsymbol{x}_m^{(i)}]_{m \in [j]}$.

8:     **end for**

9:     $\boldsymbol{y}^{(i)} \leftarrow \left[y_j^{(i)}\right]_{k \in [N_i]}$.

10:     Calculate $\text{lcb}_i(\cdot)$ and $\text{ucb}_i(\cdot)$ as

$$\text{lcb}_i(\boldsymbol{x}) = \mu_{\lambda^2 \boldsymbol{I}_{N_i}}(\boldsymbol{x}; \mathbf{X}_{N_i}^{(i)}, \boldsymbol{y}^{(i)}) - \beta^{1/2} \sigma_{\lambda^2 \boldsymbol{I}_{N_i}}(\boldsymbol{x}; \mathbf{X}_{N_i}^{(i)}),$$

$$\text{ucb}_i(\boldsymbol{x}) = \mu_{\lambda^2 \boldsymbol{I}_{N_i}}(\boldsymbol{x}; \mathbf{X}_{N_i}^{(i)}, \boldsymbol{y}^{(i)}) + \beta^{1/2} \sigma_{\lambda^2 \boldsymbol{I}_{N_i}}(\boldsymbol{x}; \mathbf{X}_{N_i}^{(i)}).$$

11:     $\mathcal{X}_{i+1} \leftarrow \left\{\boldsymbol{x} \in \mathcal{X}_i \;\middle|\; \text{ucb}_i(\boldsymbol{x}) \geq \max_{\widetilde{\boldsymbol{x}} \in \mathcal{X}_i} \text{lcb}_i(\widetilde{\boldsymbol{x}})\right\}$.

12:     Update the batch size $N_{i+1} \leftarrow 2N_i$.

13: **end for**

---

---

**Algorithm 2** Maximum Variance Reduction (MVR)

---

**Require:** Noise variance parameter $\lambda^2 \geq 0$.

1: $\mathbf{X}_0 = \emptyset$.

2: **for** $t = 1, 2, \ldots, T$ **do**

3:     $\boldsymbol{x}_t \leftarrow \arg\max_{\boldsymbol{x} \in \mathcal{X}} \sigma_{\lambda^2 \boldsymbol{I}_{t-1}}(\boldsymbol{x}; \mathbf{X}_{t-1})$.

4:     Observe $y_t$ and construct data $\mathbf{X}_t := [\boldsymbol{x}_j]_{j \in [t]}$.

5: **end for**

6: Return the estimated solution $\widehat{\boldsymbol{x}}_T := \arg\max_{\boldsymbol{x} \in \mathcal{X}} \mu_{\lambda^2 \boldsymbol{I}_T}(\boldsymbol{x}; \mathbf{X}_T; \boldsymbol{y}_T)$, where $\boldsymbol{y}_T = [y_t]_{t \in [T]}$.

---

$\rho_t \leq \lambda_t$. Then, for any algorithm, the following event holds with probability at least $1 - \delta$:

$$\forall t \in \mathbb{N}_+, \ \forall \boldsymbol{x} \in \mathcal{X}, \ |\mu_{\Sigma_{t-1}}(\boldsymbol{x}; \mathbf{X}_{t-1}, \boldsymbol{y}_{t-1}) - f(\boldsymbol{x})| \leq \left(B + \sqrt{2\gamma_t(\Sigma_t) + 2\ln\frac{1}{\delta}}\right) \sigma_{\Sigma_{t-1}}(\boldsymbol{x}; \mathbf{X}_{t-1}). \tag{130}$$

**Lemma I.2** (General regret bound of VA-GP-UCB). *Fix any $\zeta > 0$ and $\delta \in (0, 1)$. Suppose Assumptions 2.1 and 2.2 hold. Then, when running Algorithm 5 with $\beta_t^{1/2} = B + \sqrt{2\gamma_t(\Sigma_t) + 2\ln\frac{1}{\delta}}$, the following two statements hold with probability at least $1 - \delta$:*

- *The cumulative regret $R_T$ of VA-GP-UCB satisfies*

$$R_T \leq 2B \min\left\{4\overline{\gamma}\left(4\gamma_T(\Sigma_T), \zeta^2\right), 4\gamma_T(\Sigma_T)\right\} + 4\beta_T^{1/2}\sqrt{(V_T + \zeta^2 T)\gamma_T(\Sigma_T)}. \tag{131}$$

- *The simple regret $r_T$ of VA-GP-UCB satisfies*

$$r_T \leq \frac{2B}{T} \min\left\{4\overline{\gamma}\left(4\gamma_T(\Sigma_T), \zeta^2\right), 4\gamma_T(\Sigma_T)\right\} + \frac{4\beta_T^{1/2}}{T}\sqrt{(V_T + \zeta^2 T)\gamma_T(\Sigma_T)} \tag{132}$$

*In the above upper bound, the estimated solution $\widehat{\boldsymbol{x}}_T$ is defined as $\widehat{\boldsymbol{x}}_T = \boldsymbol{x}_{\widetilde{t}}$ with $\widetilde{t} \in \arg\max_{t \in [T]} \text{lcb}_t(\boldsymbol{x}_t)$. Here, $\text{lcb}_t(\boldsymbol{x}_t)$ is defined as $\text{lcb}_t(\boldsymbol{x}_t) = \mu_{\Sigma_{t-1}}(\boldsymbol{x}_t; \mathbf{X}_{t-1}, \boldsymbol{y}_{t-1}) - \beta_t^{1/2} \sigma_{\Sigma_{t-1}}(\boldsymbol{x}_t; \mathbf{X}_{t-1})$.*

---

**Algorithm 3** Variance-aware phased elimination (VA-PE)

---

**Require:** Confidence width parameter $\beta^{1/2} > 0$, finite input set $\mathcal{X}$, initial batch size $N_1 \in \mathbb{N}_+$.
1: Initialize the potential maximizer $\mathcal{X}_1 \leftarrow \mathcal{X}$.
2: **for** $i = 1, 2, \dots$ **do**
3:     $\mathbf{X}_0^{(i)} = \emptyset, \Sigma_0^{(i)} = \emptyset$.
4:     **for** $j = 1, \dots, N_i$ **do**
5:        $\boldsymbol{x}_j^{(i)} \leftarrow \arg\max_{\boldsymbol{x} \in \mathcal{X}_i} \sigma_{\Sigma_{j-1}^{(i)}}(\boldsymbol{x}; \mathbf{X}_{j-1}^{(i)})$.
6:        Observe $y_j^{(i)} = f\left(\boldsymbol{x}_j^{(i)}\right) + \epsilon_j^{(i)}$.
7:        Obtain the variance proxy $\left(\rho_j^{(i)}\right)^2$.
8:        $\left(\lambda_j^{(i)}\right)^2 \leftarrow \left(\rho_j^{(i)}\right)^2$.
9:        $\mathbf{X}_j^{(i)} \leftarrow [\boldsymbol{x}_m^{(i)}]_{m \in [j]}, \Sigma_j^{(i)} \leftarrow \mathrm{diag}\left(\left(\lambda_1^{(i)}\right)^2, \dots, \left(\lambda_j^{(i)}\right)^2\right)$
10:     **end for**
11:     $\boldsymbol{y}^{(i)} \leftarrow \left[y_j^{(i)}\right]_{k \in [N_i]}$.
12:     Calculate $\mathrm{lcb}_i(\cdot)$ and $\mathrm{ucb}_i(\cdot)$ as

$$\mathrm{lcb}_i(\boldsymbol{x}) = \mu_{\Sigma_{N_i}^{(i)}}\left(\boldsymbol{x}; \mathbf{X}_{N_i}^{(i)}, \boldsymbol{y}^{(i)}\right) - \beta^{1/2} \sigma_{\Sigma_{N_i}^{(i)}}\left(\boldsymbol{x}; \mathbf{X}_{N_i}^{(i)}\right),$$

$$\mathrm{ucb}_i(\boldsymbol{x}) = \mu_{\Sigma_{N_i}^{(i)}}\left(\boldsymbol{x}; \mathbf{X}_{N_i}^{(i)}, \boldsymbol{y}^{(i)}\right) + \beta^{1/2} \sigma_{\Sigma_{N_i}^{(i)}}\left(\boldsymbol{x}; \mathbf{X}_{N_i}^{(i)}\right).$$

13:     $\mathcal{X}_{i+1} \leftarrow \left\{\boldsymbol{x} \in \mathcal{X}_i \,\middle|\, \mathrm{ucb}_i(\boldsymbol{x}) \geq \max_{\widetilde{\boldsymbol{x}} \in \mathcal{X}_i} \mathrm{lcb}_i(\widetilde{\boldsymbol{x}})\right\}$.
14:     Update the batch size $N_{i+1} \leftarrow 2N_i$.
15: **end for**

---

**Algorithm 4** Variance-aware maximum variance reduction (VA-MVR)

---

**Require:** Finite input set $\mathcal{X}$.
1: $\mathbf{X}_0 = \emptyset, \Sigma_0 = \emptyset$.
2: **for** $t = 1, 2, \dots, T$ **do**
3:     $\boldsymbol{x}_t \leftarrow \arg\max_{\boldsymbol{x} \in \mathcal{X}} \sigma_{\Sigma_{t-1}}(\boldsymbol{x}; \mathbf{X}_{t-1})$.
4:     Observe $y_t$, and obtain the variance proxy $\rho_t^2$.
5:     Construct data $\mathbf{X}_t := [\boldsymbol{x}_j]_{j \in [t]}$.
6:     $\lambda_t^2 \leftarrow \rho_t^2, \Sigma_t \leftarrow \mathrm{diag}(\lambda_1^2, \dots, \lambda_t^2)$.
7: **end for**
8: Return the estimated solution $\widehat{\boldsymbol{x}}_T := \arg\max_{\boldsymbol{x} \in \mathcal{X}} \mu_{\Sigma_T}(\boldsymbol{x}; \mathbf{X}_T; \boldsymbol{y}_T)$, where $\boldsymbol{y}_T = [y_t]_{t \in [T]}$.

---

*Proof.* From the construction of $\lambda_t^2$ in Algorithm 5 and the definition of $\beta^{1/2}$, the event (130) holds with probability at least $1 - \delta$. Hereafter, we suppose the event (130) holds. Furthermore, we set $\mathcal{T} = \{t \in [T] \mid \lambda_t^{-1} \sigma_{\Sigma_{t-1}}(\boldsymbol{x}_t; \mathbf{X}_{t-1}) \leq 1\}$ and $\mathcal{T}^c = \{t \in [T] \mid \lambda_t^{-1} \sigma_{\Sigma_{t-1}}(\boldsymbol{x}_t; \mathbf{X}_{t-1}) > 1\}$.

**Cumulative regret upper bound.** We decompose $R_T$ as $R_T = \sum_{t \in \mathcal{T}} f(\boldsymbol{x}^*) - f(\boldsymbol{x}_t) + \sum_{t \in \mathcal{T}^c} f(\boldsymbol{x}^*) - f(\boldsymbol{x}_t)$, and consider the upper bound of each term separately. First, the first term satisfies

$$\sum_{t \in \mathcal{T}} f(\boldsymbol{x}^*) - f(\boldsymbol{x}_t) \leq 2 \sum_{t \in \mathcal{T}} \beta_t^{1/2} \sigma_{\Sigma_{t-1}}(\boldsymbol{x}; \mathbf{X}_{t-1}) \tag{133}$$

$$\leq 2\beta_T^{1/2} \sum_{t \in \mathcal{T}} \sigma_{\Sigma_{t-1}}(\boldsymbol{x}; \mathbf{X}_{t-1}) \tag{134}$$

$$\leq 4\beta_T^{1/2} \sqrt{\left( \sum_{t \in [T]} \lambda_t^2 \right) \gamma_T(\Sigma_T)} \tag{135}$$

$$\leq 4\beta_T^{1/2} \sqrt{(V_T + \zeta^2 T) \gamma_T(\Sigma_T)}, \tag{136}$$

where the first inequality follows from the event (130) and the UCB-selection rule of $\boldsymbol{x}_t$, and the third inequality follows from the same arguments as Eqs. (32)–(36). Regarding the second term, from the extension of the elliptical potential count lemma (Lemma C.1), we have

$$\sum_{t \in \mathcal{T}^c} f(\boldsymbol{x}^*) - f(\boldsymbol{x}_t) \leq 2B|\mathcal{T}^c| \tag{137}$$

$$\leq 2B \min \left\{ 4\bar{\gamma} \left( 4\gamma_T(\Sigma_T), \zeta^2 \right), 4\gamma_T(\Sigma_T) \right\}. \tag{138}$$

From Eqs. (135) and (138), we obtain the desired result.

**Simple regret upper bound.** From the definition of $\widehat{\boldsymbol{x}}_T$, for any $t \in \mathcal{T}$, we have

$$f(\boldsymbol{x}^*) - f(\widehat{\boldsymbol{x}}_T) \leq \text{ucb}_t(\boldsymbol{x}_t) - \text{lcb}_{\widetilde{t}}(\boldsymbol{x}_{\widetilde{t}}) \tag{139}$$

$$\leq \text{ucb}_t(\boldsymbol{x}_t) - \text{lcb}_t(\boldsymbol{x}_t) \tag{140}$$

$$\leq 2\beta_T^{1/2} \sigma_{\Sigma_{t-1}}(\boldsymbol{x}; \mathbf{X}_{t-1}). \tag{141}$$

Furthermore, for any $t \in \mathcal{T}^c$, we have $f(\boldsymbol{x}^*) - f(\widehat{\boldsymbol{x}}_T) \leq 2B$. By taking the average of the above inequalities, we have

$$f(\boldsymbol{x}^*) - f(\widehat{\boldsymbol{x}}_T) \leq \frac{1}{T} \left[ \sum_{t \in \mathcal{T}} 2\beta_T^{1/2} \sigma_{\Sigma_{t-1}}(\boldsymbol{x}; \mathbf{X}_{t-1}) + \sum_{t \in \mathcal{T}^c} 2B \right] \tag{142}$$

$$\leq \frac{4\beta_T^{1/2}}{T} \sqrt{(V_T + \zeta^2 T) \gamma_T(\Sigma_T)} + \frac{2B}{T} \min \left\{ 4\bar{\gamma} \left( 4\gamma_T(\Sigma_T), \zeta^2 \right), 4\gamma_T(\Sigma_T) \right\}. \tag{143}$$

$\square$

**Theorem I.3** (Regret bound of VA-GP-UCB for $k = k_{\text{SE}}$ and $k = k_{\text{Matérn}}$). *Let us assume the same setting as that of Lemma I.2. Furthermore, suppose $V_T = \Omega(1)$. Then, the following statements hold with probability at least $1 - \delta$,*

- *If $k = k_{\text{SE}}$, by setting $\zeta^2$ as $\zeta^2 = 1/T$, we have*

$$R_T = O\left( \sqrt{V_T} \ln^{d+1} T \right) \text{ and } r_T = O\left( \sqrt{\frac{V_T}{T^2}} \ln^{d+1} T \right). \tag{144}$$

- *If $k = k_{\text{Matérn}}$, by setting $\zeta^2$ as $\zeta^2 = 1/T$, we have*

$$R_T = \widetilde{O}\left( T^{\frac{2d}{2\nu+d}} \sqrt{V_T} \right) \text{ and } r_T = \widetilde{O}\left( T^{-\frac{2\nu-d}{2\nu+d}} \sqrt{V_T} \right). \tag{145}$$

*Proof.* When $k = k_{\text{SE}}$, $\gamma_T(\Sigma_T) = O\left( \ln^{d+1} \frac{T}{\zeta^2} \right) = O\left( \ln^{d+1} T^2 \right)$ and $4\beta_T^{1/2} \sqrt{(V_T + \zeta^2 T) \gamma_T(\Sigma_T)} = O\left( \gamma_T(\Sigma_T)\sqrt{V_T} \right) = O\left( \sqrt{V_T} \ln^{d+1} T^2 \right)$. By noting $V_T = \Omega(1)$, we obtain

$$R_T = O\left( \sqrt{V_T} \ln^{d+1} T \right) \tag{146}$$

---

**Algorithm 5** Variance-aware Gaussian process upper confidence bound (VA-GP-UCB)

**Require:** Confidence width parameter $\beta_t^{1/2} > 0$, lower threshold $\zeta > 0$ of variance parameters.
1: $\mathbf{X}_0 = \emptyset$, $\boldsymbol{y}_0 = \emptyset$, $\Sigma_0 = \emptyset$.
2: **for** $t = 1, \ldots, T$ **do**
3:     Compute $\mathrm{ucb}_t(\boldsymbol{x}) \coloneqq \mu_{\Sigma_{t-1}}(\boldsymbol{x}; \mathbf{X}_{t-1}, \boldsymbol{y}_{t-1}) + \beta^{1/2} \sigma_{\Sigma_{t-1}}(\boldsymbol{x}; \mathbf{X}_{t-1})$.
4:     $\boldsymbol{x}_t \leftarrow \arg\max_{\boldsymbol{x} \in \mathcal{X}} \mathrm{ucb}_t(\boldsymbol{x})$.
5:     Observe $y_t = f(\boldsymbol{x}_t) + \epsilon_t$.
6:     Obtain the variance proxy $\rho_t^2 \geq 0$.
7:     $\lambda_t^2 \leftarrow \max\{\rho_t^2, \zeta^2\}$.
8:     $\mathbf{X}_t \leftarrow [\boldsymbol{x}_m]_{m \in [t]}$, $\Sigma_t \leftarrow \mathrm{diag}(\lambda_1^2, \ldots, \lambda_t^2)$.
9: **end for**

---

from Lemma I.2.

When $k = k_{\mathrm{Matérn}}$, $\gamma_T(\Sigma_T) \leq \gamma_T(\zeta^2 \boldsymbol{I}_T) = \widetilde{O}\left(\left(\frac{T}{\zeta^2}\right)^{\frac{d}{2\nu+d}}\right) = \widetilde{O}\left(T^{\frac{2d}{2\nu+d}}\right)$, $\overline{\gamma}\left(4\gamma_T(\Sigma_T), \zeta^2\right) = \widetilde{O}\left(T^{\left(\frac{d}{2\nu+d}\right)^2}(\zeta^2)^{-\frac{d}{2\nu+d} - \left(\frac{d}{2\nu+d}\right)^2}\right) = \widetilde{O}\left(T^{\left[2\left(\frac{d}{2\nu+d}\right)^2 + \frac{d}{2\nu+d}\right]}\right)$, and $4\beta_T^{1/2}\sqrt{(V_T + \zeta^2 T)\gamma_T(\Sigma_T)} = O\left(\gamma_T(\Sigma_T)\sqrt{V_T}\right) = \widetilde{O}\left(T^{\frac{2d}{2\nu+d}}\sqrt{V_T}\right)$. By noting $V_T = \Omega(1)$, we can conclude the following from Lemma I.2:

$$R_T = \widetilde{O}\left(T^{\frac{2d}{2\nu+d}}\sqrt{V_T}\right). \tag{147}$$

Finally, by comparing the upper bound of $r_T$ and $R_T$ in Lemma I.2, we can also confirm that the simple regret bounds are followed by multiplying $1/T$ to the above cumulative regret upper bounds. □

*Remark* I.4. In Theorem I.3, we suppose that the learner does not have prior knowledge about $V_T$ before running the algorithm. On the other hand, if we assume the learner knows $V_T$, the regret upper bound in Theorem I.3 can be smaller by setting the lower threshold $\zeta^2$ depending on $V_T$. Specifically, when $k = k_{\mathrm{Matérn}}$, the setting $\zeta^2 = V_T/T$ improves the polynomial dependence of $V_T$.

## J. Potential Applications for Non-Stationary Variance Setting in GP-Bandits

In this section, we discuss potential applications for the non-stationary variance setting in GP-bandits.

- **Reinforcement Learning:** The existing variance dependent algorithms in linear bandits (Zhou et al., 2021; Zhang et al., 2021; Zhou & Gu, 2022; Zhao et al., 2023) motivates the non-stationary variance setting as one of the online reinforcement learning problems. Specifically, Zhou et al. (2021) consider the regret minimization problem under a specific Markov decision process (MDP), which is called linear mixture MDP, and subsumes the linear bandit problem when the length of the horizon is 1. In the decision-making under linear mixture MDP, Zhao et al. (2023) show the algorithm with variance-dependent regret guarantees. If we assume the linear mixture MDP whose feature map of the transition probability is infinite-dimensional and induced by some kernel function, the kernelized extension of the setting of the existing works (Zhou et al., 2021; Zhang et al., 2021; Zhao et al., 2023) is naturally derived. Extensions in such a reinforcement learning setting are an interesting direction for our future research.

- **Experimental Design:** In scientific experiments, the observation noise level of the result of the experiment may vary over time. Specifically, the observation noise level can increase over extended periods due to factors such as environmental fluctuations, material degradation, or systematic drifts in measurement instruments. For example, such factors in measurement accuracy have been observed in studies of chemical analysis (Hickstein et al., 2018).

- **Stationary Setting with Heteroscedastic Variance:** Kirschner & Krause (2018) consider the heteroscedastic variance setting where the variance proxy $\rho^2(\boldsymbol{x}_t)$ may depend on the selected input $\boldsymbol{x}_t$. If we apply our algorithm to a heteroscedastic setting, the resulting regret of our non-stationary variance algorithm is quantified by the cumulative

variance proxy $V_T = \sum_{t=1}^{T} \rho^2(\boldsymbol{x}_t)$ of the selected inputs[5]. The further precise quantification of the increasing speed of $V_T$ in this setting requires an additional structural assumption about $\rho^2(\cdot)$. For example, we expect $V_T$ is increasing sublinearly with our algorithm design of $\boldsymbol{x}_t$ if there exists a unique maximizer $\boldsymbol{x}^*$, and $\rho^2(\boldsymbol{x}) \to \rho^2(\boldsymbol{x}^*)$ (as $\boldsymbol{x} \to \boldsymbol{x}^*$) and $\rho^2(\boldsymbol{x}^*) = 0$ holds. We believe this research direction is an interesting application of our analysis to the heteroscedastic setting.

---

[5] Our VA-GP-UCB algorithm can be applied in the same conditionally sub-Gaussian assumption as used in (Kirschner & Krause, 2018) (Eq. (1) in (Kirschner & Krause, 2018)). On the other hand, VA-PE and VA-MVR algorithms require a more restricted independence assumption that the noise sequence $(\epsilon_t)$ is conditionally independent given the MVR input sequence.

