# OpenReview forum: "Improved Regret Analysis in Gaussian Process Bandits: Optimality for Noiseless Reward, RKHS norm, and Non-Stationary Variance"
_ICML.cc/2025/Conference — ICML 2025 oral_

### Official Review · Reviewer_MinQ · 2025-03-12

**Overall Recommendation:** 4

**Summary:**

This paper provides refined regret analyses of maximum variance reduction (MVR) type of algorithms in Gaussian Process (GP) bandits. It first establishes a general upper bound on the maximum posterior variance for MVR algorithm (Lemma 3.1), and applies it to obtain upper bounds on the cumulative regret and the simple regret in the noiseless setting (Section 4), the bounded RKHS norm setting (Section 5), and the time-varying noise variance setting (Section 6). Across all three settings, the resulting regret bounds improve the existing upper bounds previously established in the literature.

**Claims And Evidence:**

All claims are made through theoretical analysis with solid proofs.

Although I am not doubting the results, I believe the statements Corollary 3.2 should be made more explicit regarding the hidden constants. It was stated that those hidden constants have dependence on $\mathcal{X}$, but it is unclear what the dependence is, even in their proofs in Appendix C.2. For example, is the constant finite even if $\mathcal{X}$ is continuous or unbounded? I understand the authors' main goal is to analyze the dependence of regret on $T$, $B$, or $V_T$, but the hidden constants should be stated explicitly somewhere in the paper.

**Essential References Not Discussed:**

None

**Experimental Designs Or Analyses:**

This paper does not include numerical experiments.

**Methods And Evaluation Criteria:**

This paper suggests a novel analysis, not a novel algorithm. The MVR algorithm and its variant called PE algorithm are investigated in this paper, which would be fairly straightforward and sensible algorithms for GP bandits. As performance metrics, this paper investigated the cumulative regret and the simple regret, which are also standard in the literature.

**Other Comments Or Suggestions:**

- It will also be great if the paper includes numerical experiments showing that MVR algorithms with suggested parameter choice actually perform very well.

- I am not so sure whether "heteroscedastic" is the right term to describe the non-stationary noise variance setting. I would believe "heteroscedastic" describes the situation where the noise variance depends on the query point. If the algorithm adaptively chooses the query points utilizing the heteroscedastic noise variance, it may violate Assumption 2.2 saying that the noise sequence should be mutually independent. I believe the best description would be just "time-varying noise variance".

**Other Strengths And Weaknesses:**

Although the main contribution of this paper is theoretical analysis, I believe that it also has practical implications on the implementation of MVR algorithms -- the analysis provides a guideline on how to choose the algorithm parameters $\beta$ or $\lambda$ given the target time horizon $T$ or RKHS norm $B$. However, the analysis ends up revealing the optimal asymptotic dependence of those algorithm parameters on the problem constants. It will be great if explicit suggestions are given.

**Questions For Authors:**

In Section 5, the paper analyzes the algorithms with $\lambda$ chosen to be different from the actual noise variance $\rho$, unlike the other sections. Is this something necessary?

**Relation To Broader Scientific Literature:**

The main contribution of this paper is that the improved regret upper bounds were established for GP bandits. At the core of their analysis, Lemma 3.1 plays an important role, suggesting improved posterior variance upper bounds for MVR. Although these results only apply to MVR algorithms, it is notable that the resulting regret bounds improve or match the state-of-the-art results including the other algorithms. I believe that this work will become one of key papers in the GP bandit area.

I am not so sure about this paper’s contribution beyond GP bandits. As mentioned in the paper, this work is partly related to heteroscedastic linear bandits, but I do not see a clear implication.

**Theoretical Claims:**

I had a look into the proof of Lemma 3.1, Corollary 3.2, Theorem 4.1 and Theorem 4.2. I couldn’t find a critical error, although the proof of Corollary 3.2 lacks details.

---

> ### Author Rebuttal · Authors · 2025-03-31
>
> Thank you for taking the time to review our paper and for the overall positive feedback. We will carefully incorporate the reviewer's feedback in the revision. Below is our answer to the reviewer's question.
>
> **In Section 5, the paper analyzes the algorithms with $\lambda$ chosen to be different from the actual noise variance, unlike the other sections. Is this something necessary?**
>
> The setting $\lambda = \Theta(B^{-2})$ is necessary to obtain improved dependence against the RKHS norm upper bound $B$.
> Roughly speaking, our result suggests that we should set smaller (larger) noise variance parameters if the underlying function could be complex (simple). This is intuitively reasonable from the frequentist point of view since the noise variance parameter of GP corresponds to the regularization parameter of the kernel ridge estimate. Namely, it is reasonable to set a suitable regularization parameter depending on the underlying function's complexity to obtain a good kernel ridge estimator and corresponding GP.

---

### Official Review · Reviewer_wREb · 2025-03-12

**Overall Recommendation:** 4

**Summary:**

The paper studies the classic problem of Bayesian optimization under the frequentist setting (where the target function lies in the RKHS of a known kernel). It derives a novel bound on the maximum variance after $T$ observations (Lemma 3.1). This bound has several consequences and applies to various Bayesian optimization problems. In the noiseless setting, it establishes near-optimal cumulative regret guarantees, addressing a COLT open problem. In the noisy setting, it improves regret bounds with respect to the RKHS norm of the target function. The results are also extended to the heteroscedastic noise setting.

**Claims And Evidence:**

The proofs are provided and seem correct to me.

**Essential References Not Discussed:**

The bounds on information gain given in Section 2 seem from the following paper but the reference is not provided.

On information gain and regret bounds in gaussian process bandits, Vakili, S. and Khezeli, K. and Picheny, V., AISTATS 2021.

**Ethical Review Concerns:**

Not applicable!

**Experimental Designs Or Analyses:**

Not applicable!

**Methods And Evaluation Criteria:**

The paper is of theoretical nature.

**Other Comments Or Suggestions:**

Typos in the introduction: "fileds", "Establised". Check for more typos!

**Other Strengths And Weaknesses:**

The paper is clearly written, with rigorous mathematics, and addresses regret bounds in BO under the frequentist setting, improving upon existing results.

The literature on BO with heteroscedastic noise, such as Makarova et al. (2021), does not seem to be well discussed.

**Questions For Authors:**

1. Are there additional technical challenges that should be emphasized, or do the results follow directly from combining Lemma 3.1 with the existing analysis for BO algorithms MVR and PE?

2. Can the authors specifically state the bounds on information gain that they use?

3. Is Footnote 1 on page 5 directly related to the results of this paper? Does Lemma 3.1 have implications for other algorithms such as GP-UCB? Can similar bounds to those in Lemma 3.1 be established for the sum of variances rather than the maximum variance?

4. Can the authors provide more technical details on the comparison with Flynn and Reeb (2024)? While Lemma 3.3 appears to be a crucial step in both papers, what makes the results of this paper stronger? Is the improvement related to bounding maximum variance instead of the sum of variances?

**Relation To Broader Scientific Literature:**

The paper improves regret bounds in a well-studied problem and addresses a COLT open problem.

**Theoretical Claims:**

I checked the proof of Lemma 3.1 in detail. Other results are sound.

---

> ### Author Rebuttal · Authors · 2025-03-31
>
> Thank you for taking the time to review our paper and for the overall positive feedback. Below are our answers to the reviewer's questions.
>
> **1. Are there additional technical challenges that should be emphasized, or ...**
>
> We clarify the technical challenges and novelty of the results in Sections 4-6 below.
> - The results in Section 4 directly follow Lemma 3.1 with the existing analysis; therefore, once we obtain Lemma 3.1, there is no technical challenge.
> - The novelty of the results in Section 5 is the observation that the combination of the proper setting of $\lambda^2 = \Theta(B^{-2})$ and Lemma 3.1 results in the optimal dependence of RKHS norm $B$ in regret. However, the proof itself is straightforward and simple once we find the proper scaling of noise variance setting $\lambda^2 = \Theta(B^{-2})$.
> - The analysis in Section 6 contains non-trivial technical challenges in the proof, while we omit the discussion in the paper due to space limitations. Specifically, the time-varying variance proxies make the naive application of statement 2 in Lemma 3.1 difficult. One naive way is to set $\tilde{\lambda}_t^2$ as the true variance proxy $\rho_t^2$; however, this strategy breaks the condition related to maximum information gain in Lemma 3.1 when $\rho_t^2$ becomes too small. Another idea is to introduce the lower threshold against
> variance proxy so that the condition lemma 3.1 always holds, but this strategy does not lead to the nearly-optimal dependence in the regret since the upper bound of the information gain is quantified only by the minimum of the elements in the noise variance parameter matrix. Roughly speaking, the difficult problem is that the existing theory of the information gain cannot capture the total noise level. For example, if the cumulative variance proxy is low, the information gain upper bound becomes very high if one of the variance proxies is very low. To overcome the above issues, we set $\tilde{\lambda}_t^2$ based on carefully chosen thresholds depending on the cumulative variance $V_T$ (Line 1221 for SE kernel, Line 1268 for Matern kernel). We believe that our paper's solution to the above-described issues is highly novel.
>
>
> **2. Can the authors specifically state the bounds on information gain that they use?**
>
> We use the result of Vakili et al. (2021) (Corollary 1). As the reviewer kindly pointed out, we will add the citation in the revision.
>
> **3. Can similar bounds to those in Lemma 3.1 be established for the sum of variances rather than the maximum variance?**
>
> To our current understanding, we cannot generalize our result to the sum of the posterior variance or the maximum variance in the algorithms other than MVR, because the effect of $\mathcal{T}^c$ seems not negligible (See also next question's answer in detail).
> The only thing we can obtain is an algorithm-independent "minimum" posterior variance upper bound as Eq. (4). Please see our related rebuttal of the #2 answer for Reviewer 7Bi6.
>
>
> **4. Can the authors provide more technical details on the comparison with Flynn and Reeb (2024)?**
>
> The reason why we can obtain a stronger result than that in Flynn and Reeb (2024) is that the MVR algorithm does not have to care about the regret incurred from the time subset $\mathcal{T}^c$, which is defined in Lemma 3.3.
> Flynn and Reeb (2024) uses elliptical potential count lemma for the analysis of GP-UCB; however, the resulting cumulative regret upper bound suffers from approximately $O(\min\\{\gamma_{\gamma_T(\lambda^2)}(\lambda^2), \gamma_T(\lambda^2)\\} + \sqrt{\lambda^2 T \gamma_T(\lambda^2)})$ regret. While the second term is the favorable dependence against noise variance parameter (as with our upper bound in Eq.(4)), the existence of the first term, which comes from $\mathcal{T}^c$ and the elliptical potential count lemma (Lemma 3.3), becomes problematic.
> Specifically, if we desire optimal regret in our paper's settings (such as the noiseless setting), we have to decrease the noise variance parameter rapidly so that the second term $\sqrt{\lambda^2 T \gamma_T(\lambda^2)}$ matches the lower bound.
> This can be done by setting the decreasing noise variance parameter $\lambda^2$ such that the maximum information gain $\gamma_T(\lambda^2)$ increases almost linearly with $T$; however, in this case, the resulting regret becomes far away from the desired result due to the effect of the first term $O(\min\\{\gamma_{\gamma_T(\lambda^2)}(\lambda^2), \gamma_T(\lambda^2)\\})$.
> On the other hand, as described in the proof sketch of Lemma 3.1, we can obtain the maximum posterior variance bound of MVR without the effect $|\mathcal{T}^c|$ by bounding the maximum variance with the average only on the favorable subset $\mathcal{T} = [T]\setminus \mathcal{T}^c$ (Line 226-230 in the right column). Therefore, we can obtain stronger results than those in Flynn and Reeb (2024) by bypassing the above-described problematic behavior of their analysis with MVR.

---

### Official Review · Reviewer_ATja · 2025-03-13

**Overall Recommendation:** 4

**Summary:**

The paper develops a novel bound for the posterior variance of the Gaussian process. Such bounds are used to obtain a tighter regret bound of noise-free simple/cumulative regret bounds of Bayesian optimization algorithms. Furthermore, these bounds facilitates establishing novel regret bounds for MVR/PE algorithms in both stationary/nonstationary noise setting.

**Claims And Evidence:**

The claims made in this paper seem to be valid and convincing.

**Essential References Not Discussed:**

The manuscript contains a broad range of relevant literature, including the most up-to-date ones.

**Experimental Designs Or Analyses:**

No numerical experiments.

**Methods And Evaluation Criteria:**

No numerical experiments.

**Other Comments Or Suggestions:**

1. While I was able to follow the manuscript after spending quite a bit of time, I initially had a hard time following their notations to distinguish stationary/non-stationary bound. in Lemma 3.1, it would help readers if authors add a remark in clarifying the notations and the time index $t$ and $T$.

2. The paper briefly describes the two metrics, simple regret and cumulative regret, in Section 2. It is worthwhile to distinguish the role/relationship of these regrets. For instance, simple regret represents the convergence rate, while the cumulative regret serves as a metric to assess the holistic performance of the algorithm, not just the final output.

**Other Strengths And Weaknesses:**

No significant weakness.

The manuscript contains several contributions in advancing the theory of Bayesian optimization. 1) By establishing the upper bound of the posterior variance, 2) it establishes competitive regret bounds for noise-free Bayesian optimization, 3) tightened PE/MVR regret bound under both stationary/nonstationary noise. I believe these contributions are non-trivial and, accordingly, the manuscript deserves to be accepted.

**Questions For Authors:**

Is the condition $\mathcal{X} = [0,1]^d$ necessary for corollary 6.1 and 6.2? If I recall correctly, Bull's result only assumes $\mathcal{X}$ to be a compact subset (perhaps satisfying the interior cone condition) of the Euclidean space.

**Relation To Broader Scientific Literature:**

This paper provides a novel bound on posterior variance of Gaussian process, which can be utilized many downstreaming tasks involving Gaussian process.

**Theoretical Claims:**

I went through most of the appendix to check the validity of the manuscript. It seems to be a solid work.

---

> ### Author Rebuttal · Authors · 2025-03-31
>
> Thank you for taking the time to review our paper and for the overall positive feedback. We will carefully incorporate your suggestions about the clarity into the revision.  Below is our answer to the reviewer's question.
>
> **Is the condition $\mathcal{X} = [0, 1]^d$ necessary for corollary 6.1 and 6.2? If I recall correctly, Bull's result only assumes to be a compact subset (perhaps satisfying the interior cone condition) of the Euclidean space.**
>
> Although Bull's lower bound for the noiseless setting is applicable to any sufficiently regularized compact subset, the lower bound for the noisy setting in Scarlett et al. (2017) explicitly assumes $\mathcal{X} = [0, 1]^d$.
> We added this condition to be consistent with the setting in Scarlett et al. (2017). We also believe that the result of Scarlett et al. (2017) can be generalized to any compact subset whose packing number behaves as that of $[0, 1]^d$; however, since we are not aware of any formal proof in existing work, we decided to add the condition $\mathcal{X} = [0, 1]^d$ explicitly.
>
> - Scarlett, Jonathan, Ilija Bogunovic, and Volkan Cevher. "Lower bounds on regret for noisy Gaussian process bandit optimization." Conference on Learning Theory. PMLR, 2017.

---

### Official Review · Reviewer_7Bi6 · 2025-03-14

**Overall Recommendation:** 4

**Summary:**

This paper presents improved theoretical guarantees for Gaussian Process (GP) bandit algorithms, with a particular focus on reducing regret under three key scenarios: the noiseless setting, dependence on the RKHS norm of the underlying reward function, and non-stationary noise variance. The main contribution is a new, tighter upper bound on the posterior variance of GP models used in MVR and PE algorithms. This result refines existing analyses by considering a subset of time indices in which the variance reduction is well-controlled, rather than uniformly averaging over all steps. As a result, the paper shows nearly optimal cumulative and simple regret bounds for standard kernels in various settings. The paper also provides regret bounds under time-varying noise, filling a gap in existing literature. Overall, the analysis is clean, and the technical results are novel and broadly applicable.

## update after rebuttal
The authors have provided clear and satisfactory responses to most of the questions raised in the initial review. As a result, I will maintain my original score.

**Claims And Evidence:**

The theoretical claims in the paper are generally well-supported. The main claim regarding the improved posterior variance upper bound (Lemma 3.1) is carefully proven, with clear intuition and appropriate use of mutual information and elliptical potential arguments. The paper also provides detailed corollaries (Corollary 3.2) that concretely instantiate the regret implications under SE and Matérn kernels. The claims about achieving optimal dependence on the RKHS norm are also backed by theorems with well-structured proofs and appropriate comparison to prior bounds. However, the paper does not discuss the polynomial kernel case, which was treated in prior work (e.g., Srinivas et al., 2010). Including this would help verify consistency with linear bandit results and strengthen the generality of the framework.

**Essential References Not Discussed:**

References are sufficient enough.

**Experimental Designs Or Analyses:**

There are no experiments in this paper, which is acceptable given its theoretical nature.

**Methods And Evaluation Criteria:**

The methods are theoretically grounded and appropriate for the goal of tightening regret bounds under the GP bandit setting.

**Other Comments Or Suggestions:**

- Consider adding a "Notation Table" near the end of Section 2 for ease of reference.
- Discuss the behavior under polynomial kernels, as it connects to linear bandits and prior results.

**Other Strengths And Weaknesses:**

- Theoretical novelty in posterior variance bounding technique.
- Clear improvement over existing results in noiseless and RKHS-dependent settings.
- Addresses an underexplored but practically relevant scenario of heteroscedastic noise.

**Questions For Authors:**

1. Have you considered evaluating your bounds under the polynomial kernel? Given its finite-dimensional nature, the information gain is logarithmic, and your results should match linear bandit regret bounds (I guess). This would be a valuable consistency check.

2. Is it possible to extend your special time-index argument to randomized selection methods such as GP-TS? Could this lead to improved analyses for such algorithms?
- Chowdhury, Sayak Ray, and Aditya Gopalan. "On kernelized multi-armed bandits." International Conference on Machine Learning. PMLR, 2017.

3. To ensure I understood the key technical improvement correctly: Is the main reason for the improved regret bound in Lemma 3.1 the fact that the authors consider a carefully chosen subset of time indices $\mathcal{T}=\\{\frac{T}{2} \geq 3 \gamma_T(\tilde{\lambda}_T^2 I_T)\\}$, instead of averaging over all $[T]$? If so, this selective averaging strategy appears crucial, could the authors clarify how general this idea is, and whether it can be extended to other types of GP bandit algorithms beyond MVR?

**Relation To Broader Scientific Literature:**

The work builds upon classical results in GP bandits and recent advances in MVR and PE. It contributes to tightening regret bounds with respect to noise and RKHS norm, which is important in both theoretical and practical settings.

**Theoretical Claims:**

I checked the derivation of Lemma 3.1 and its application to the noiseless setting and RKHS-norm-dependent regret bounds. The idea of using a selected index subset $\mathcal{T}$ for tighter bounding is novel and clean. The use of the elliptical potential count lemma and its adaptation are sound. The regret bounds in Theorems 4.1, 4.2, and 5.1 seem correct under the stated assumptions. I did not identify any obvious mistakes.

---

> ### Author Rebuttal · Authors · 2025-03-31
>
> Thank you for taking the time to review our paper and for the overall positive feedback. We will carefully incorporate your comments into the revision. Below are the answers to your questions.
>
> **1. Have you considered evaluating your bounds under the polynomial kernel? Given its finite-dimensional nature, the information gain is logarithmic, and your results should match linear bandit regret bounds (I guess). This would be a valuable consistency check.**
>
> Thank you for your suggestion.
> As for the non-stationary variance setting, in Lines 402-424, we discuss the connection between the existing linear bandit method and
> our work. However, so far, we are not aware of the existing linear bandit literature that focuses on the noiseless setting, or the RKHS norm upper bound (L2-norm upper bound of linear parameter). Therefore, we will add the discussions in the revision after carefully investigating the related works in linear bandits.
>
>
> **2. Is it possible to extend your special time-index argument to randomized selection methods such as GP-TS? Could this lead to improved analyses for such algorithms?**
>
> At least, maximum posterior variance upper bounds (e.g., Eq. (4)) are only applicable for the MVR algorithm.
> We believe that the extension to other well-known algorithms is important for future work.
> On the other hand, we can also obtain a more limited result: an algorithm-independent "minimum" posterior variance upper bound as Eq. (4).
> This can be easily confirmed by replacing the L.H.S. of the inequality in Lines 229-231 with the minimum posterior variance.
> This upper bound is meaningless for the analysis of the cumulative regret; however, it is useful for analyzing the simple regret of other algorithms or stopping time analysis for other problem settings (e.g., level-set estimation (Gotovos et al., 2013)).
>
> - Gotovos, Alkis, et al. "Active learning for level set estimation." Proceedings of the Twenty-Third international joint conference on Artificial Intelligence. 2013.
>
> **3. To ensure I understood the key technical improvement correctly: Is the main reason for the improved regret bound in Lemma 3.1 the fact that the authors consider a carefully chosen subset of time indices. This selective averaging strategy appears crucial, could the authors clarify how general this idea is, and whether it can be extended to other types of GP bandit algorithms beyond MVR?**
>
> Since the subset selection $\mathcal{T}$ itself is conducted by the elliptical potential count lemma (Lemma 3.3), which is an algorithm-independent lemma, at least Eq. (9) always holds for any algorithm. On the other hand, dealing with the regret incurred from the remaining index set $\mathcal{T}^c = [T] \setminus \mathcal{T}$ is difficult for algorithms other than MVR. So far, the only possible claim in algorithms other than MVR is the upper bound given on Lines 237-242 in the left column. As discussed in Lines 234-250 in the left column, although this gives a certain extent of improved dependence on the noise variance parameter for any algorithm, the decreasing speed of the noise variance is more restricted and is not sufficient to claim near-optimality in our analysis.

---

### Decision · Program_Chairs · 2025-05-01

**Decision:**

Accept (oral)

**Comment:**

This paper studies the Gaussian Process bandits problem in noiseless, varying RKHS norm, and non-stationary variance settings. Due to the refined maximum posterior variance analysis, this paper provides improved bounds in these three settings. All reviewers like the technical novelties and contributions of this paper, therefore, I'm happy to recommend this paper for outstanding paper award and oral presentation.